# Genetic imputation of kidney transcriptome, proteome and multi-omics illuminates new blood pressure and hypertension targets

Genetic mechanisms of blood pressure (BP) regulation remain poorly defined. Using kidney-specific epigenomic annotations and 3D genome information we generated and validated gene expression prediction models for the purpose of transcriptome-wide association studies in 700 human kidneys. We identified 889 kidney genes associated with BP of which 399 were prioritised as contributors to BP regulation. Imputation of kidney proteome and microRNAome uncovered 97 renal proteins and 11 miRNAs associated with BP. Integration with plasma proteomics and metabolomics illuminated circulating levels of myo-inositol, 4-guanidinobutanoate and angiotensinogen as downstream effectors of several kidney BP genes (*SLC5A11*, *AGMAT*, *AGT*, respectively). We showed that genetically determined reduction in renal expression may mimic the effects of rare loss-of-function variants on kidney mRNA/protein and lead to an increase in BP (e.g., *ENPEP*). We demonstrated a strong correlation (r = 0.81) in expression of protein-coding genes between cells harvested from urine and the kidney highlighting a diagnostic potential of urinary cell transcriptomics. We uncovered adenylyl cyclase activators as a repurposing opportunity for hypertension and illustrated examples of BP-elevating effects of anticancer drugs (e.g. tubulin polymerisation inhibitors). Collectively, our studies provide new biological insights into genetic regulation of BP with potential to drive clinical translation in hypertension.

Persistently raised blood pressure (BP) (hypertension) is single most important attributable risk factor for death globally[1-3]. BP is a heritable trait[4-6]; previous studies uncovered ultra-rare[7], low-frequency[8-11] and common[12-23] genetic variants associated with BP and/or hypertension. The mechanisms underpinning the role of ultra-rare variants in the development of low/high BP is well-defined – almost all of them result in alterations of sodium/water reabsorption in the kidney through effect on target genes in the distal nephron and collecting duct[7,24]. Several common genetic variants associated with BP in genome-wide association studies (GWAS) also act through alterations of epigenetic and transcriptional programmes operating in the kidney[25,26] – an organ of "over-riding dominance" in the pathogenesis of hypertension[27-30]. Indeed, our recent studies uncovered the identity of some of their

target genes and established causal connections between DNA-methylation, expression and splicing of these genes and BP through causal inference analyses[25]. However, for a majority of GWAS loci associated with BP the effector genes have not been yet identified. Thus, new strategies inclusive of previously unexplored "omics" layers are needed to accelerate the BP gene discovery and fully characterise the downstream molecular mechanisms and clinically measurable consequences of these signals.

Here, using a collection of up to 700 human kidneys[25,31-34] and new computational algorithms embedded in three-dimensional (3D) configuration of the genome and kidney epigenome, we uncover 6490 kidney genes with genetically imputable expression (~30.3% of kidney transcriptome). We perform BP transcriptome-wide association

✉e-mail: maciej.tomaszewski@manchester.ac.uk

studies (TWAS), Mendelian randomisation and fine-mapping of causal gene sets (FOCUS) to prioritise kidney effector genes for BP regulation. Through genetic imputation of the kidney microRNAome and proteome we uncover the identity of renal miRNAs and proteins associated with BP. Our computational drug repositioning analysis demonstrates BP effects of the existing non-cardiovascular medications and highlight new drug repositioning opportunities for hypertension. We also analyse urinary cell transcriptome to provide insights into diagnostic tractability of BP kidney genes. Finally, we triangulate outputs from plasma proteomics and metabolomics with kidney TWAS to yield new insights into pathways of blood pressure regulation.

## Results

### Prioritisation of human cell-types/tissues for relevance to blood pressure through transcriptome-wide association studies

TWAS can not only shed light on the biological importance of cell-types/tissues to a trait/disease but also uncover new and unexpected tissue-disease associations, e.g. that of intestines to psychiatric disorders[35] or monocytes to Parkinson's disease[36].

We applied the elastic net method[37] to predict the genetically regulated component of gene expression across 49 human cell types and tissues from numerically identical sets of individuals in Genotype Tissue Expression (GTEx). We used an equal number of samples ($n = 65$) across the panel of 49 tissues to maximise the comparability of BP TWAS discovery rates across these tissues accepting that this would be at the expense of the individual

discovery rates in tissues that have been down-sampled. After quality control and a nested cross-validation we used S-PrediXcan to estimate the mediating effects of gene expression levels in various GTEx tissues on systolic BP (SBP) and diastolic BP (DBP). In brief, we used the eQTL data for 25,332 human genes across 49 GTEx tissues and summary statistics for 7,088,121 and 7,160,657 SNPs from the GWAS meta-analysis of SBP and DBP conducted in UK Biobank and the International Consortium for Blood Pressure (ICBP)[22] (~750,000 individuals). We then calculated the overall SBP and DBP TWAS scores for each of the human cell-types/tissues ranking them for their relevance to genetic regulation of SBP and DBP, (Figs. 1A, 2A, S1–2 and Supplementary Data 1–2). Cultured fibroblasts and kidney cortex showed the strongest overall association with SBP and DBP, respectively (Figs. 2A, S1–2 and Supplementary Data 1–2).

Only five cell types/tissues – kidney, cultured fibroblasts, adrenal gland, cells EBV-transformed lymphocytes and thyroid ranked within top ten tissues for both SBP and DBP (Figs. 2A, S1 and Supplementary Data 1–2). Both vasculature (aorta, tibial artery) and central nervous system (several areas of brain including hippocampus, cortex, cerebellar hemisphere, frontal cortex, cerebellum, and caudate basal ganglia) showed a strong representation within the tissues of the highest relevance to BP (Figs. 2A, S1 and Supplementary Data 1–2). The uncovered association between lymphocytes, spleen and minor salivary gland supports the emerging role of immune system in BP regulation and hypertension[38,39] while the connection between the brain

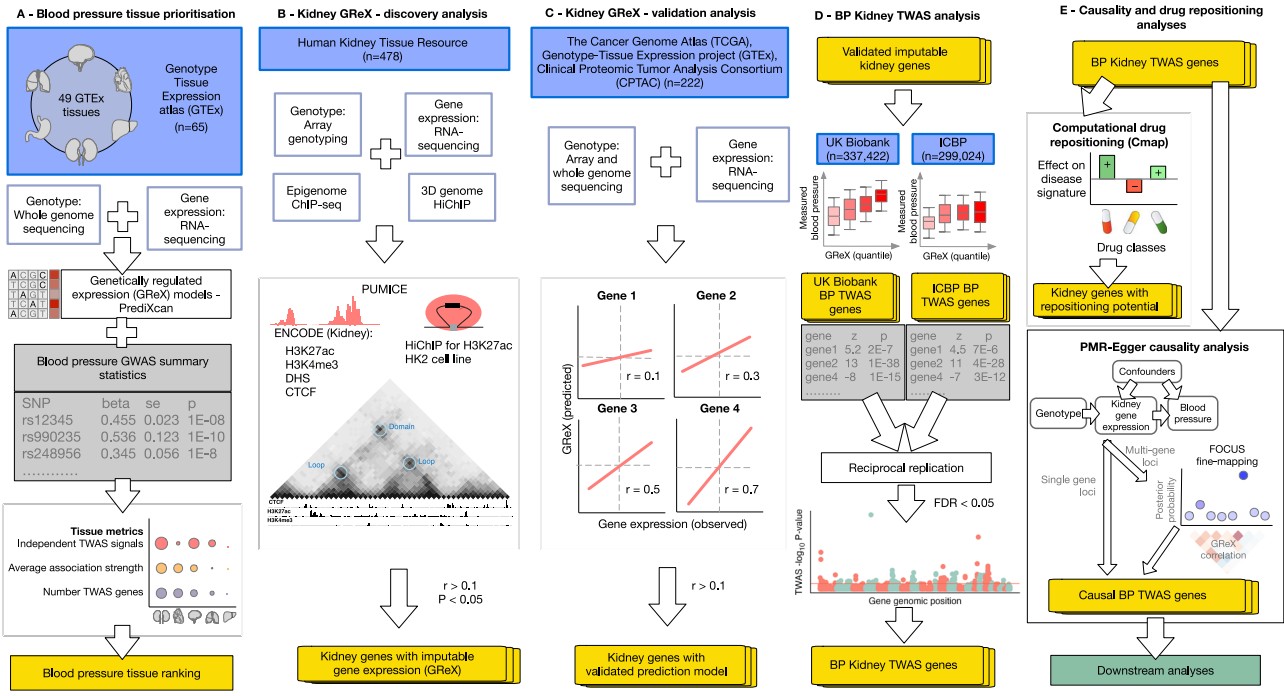

**Fig. 1 | Transcriptome-wide association studies, kidney and blood pressure – schematic representation of input data sources (with sample size), analytical processes and output data. A** Blood pressure tissue prioritisation. **B** Kidney GReX derived by Prediction Using Models Informed by Chromatin conformations and Epigenomics (PUMICE) algorithm – discovery analysis. **C** Kidney GReX – validation analysis. **D** BP kidney TWAS analysis. **E** Causality and drug repositioning analyses. The input data sources are coloured in blue, schematic intermediate results are coloured in grey, the primary outputs are coloured in yellow, downstream single-gene analyses are marked in green. GReX – genetically regulated expression, GWAS – genome-wide association study, TWAS – transcriptome-wide association studies, Beta – effect size estimate, SE – standard error of beta, ChIP-seq – chromatin immunoprecipitation sequencing, HiChIP – chromosome conformation capture by sequencing and immunoprecipitation, ENCODE – encyclopaedia of DNA

elements consortium, H3K27me3 – histone 3, lysine residue 27, tri-methylation, H3K4me3 – histone 3, lysine residue 4, tri-methylation, DHS – DNase I hypersensitive sites, H3K27ac – histone 3, lysine residue 27, acetylation, HK2 cell line – human kidney 2 cell line, BP – blood pressure, ICBP – International Consortium for Blood Pressure, FDR – false discovery rate, PMR – probabilistic Mendelian randomisation, FOCUS – fine-mapping of causal gene sets. Parts of the figure were drawn by using pictures from Servier Medical Art and some of these pictures were modified. Servier Medical Art by Servier is licensed under a Creative Commons Attribution 3.0 Unported License (https://creativecommons.org/licenses/by/3.0/. Further parts of the figure were drawn by using pictures from Marcel Tisch (https://twitter.com/MarcelTisch) and are licensed under a Creative Commons CC0 License (https://creativecommons.org/publicdomain/zero/1.0/).

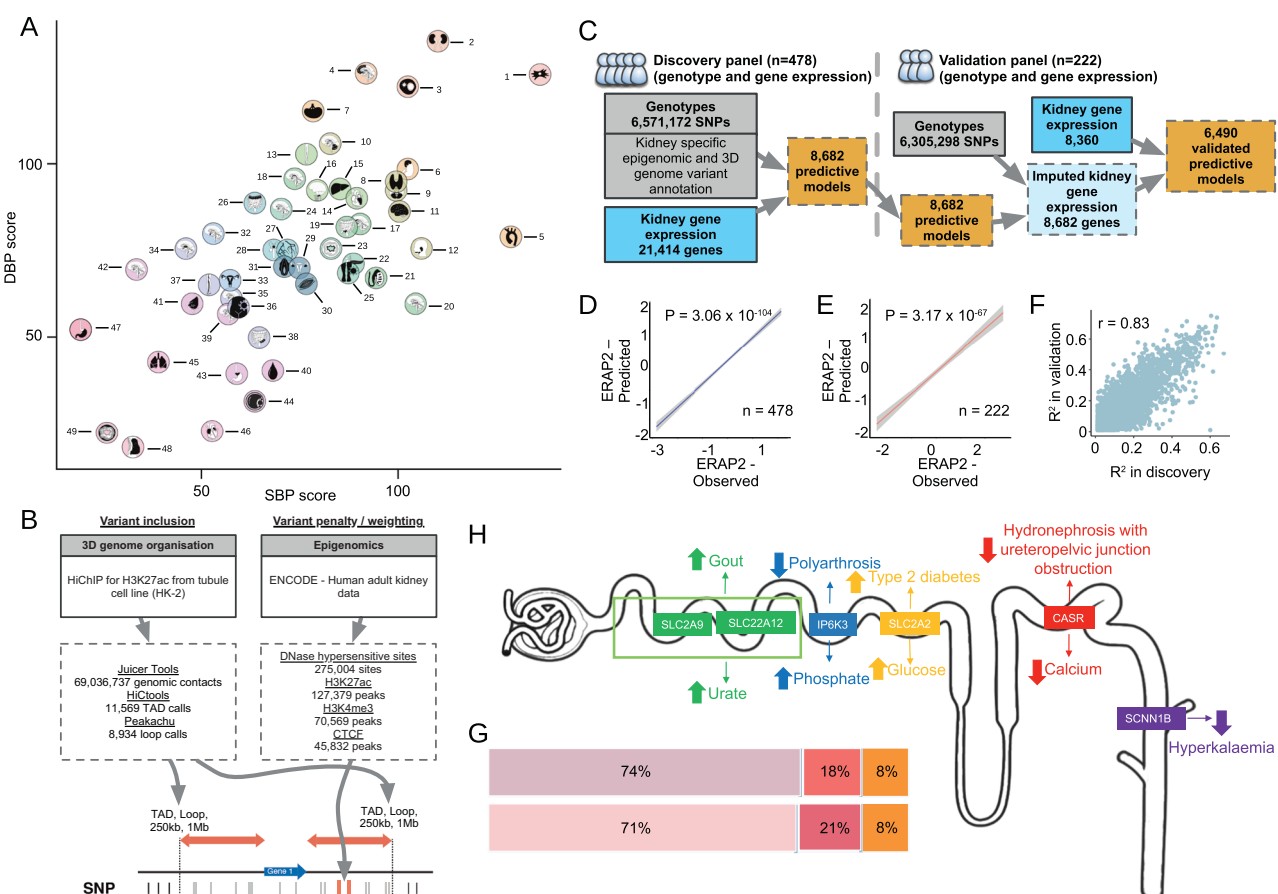

**Fig. 2 | Blood pressure TWAS – from prioritisation of human tissues to development of enhanced gene expression prediction kidney model.**
**A** Representation of 49 human tissues and cell-types from the GTEx project ranked by magnitude of association with systolic (SBP) and diastolic blood pressure (DBP). Higher score represents stronger association with blood pressure. The ranking of each tissue is labelled from 1 to 49 derived by the sum of SBP and DBP scores. The colour scale is based on the ranking (from highest to lowest) from orange, green, blue, purple to pink. Each tissue or cell-type highlighted in black. Further information is shown in Figs. S1–2. Created by Idoya Lahortiga. **B** Kidney-informed enhanced prediction of gene expression. Numbers of features (peaks, 3D structures) are shown for each kidney data layer. TAD – The variant inclusion window [delimited by a topologically associating domain (TAD), chromatin loop domain (Loop), 1 Mb fixed window or 250 kb fixed window] and variant specific weighting strategy are optimised to maximise gene discovery in kidney transcriptome-wide association study (TWAS) using 3D genome and epigenomic data directly measured in the kidney. CTCF – CCCTC-binding factor. HiChIP – chromosome conformation capture with chromatin immunoprecipitation, genomic contact – two regions of chromatin in close physical proximity in the HK-2 cell line. **C** TWAS

model prediction workflow. Data are coloured by type: genotype – grey, gene expression – blue, predictive model – orange. **D**, **E** Correlation between the predicted genetically regulated expression (GReX) and observed expression of *ERAP2* in HKTR (blue) and in NIH resources (red). The best-fitting line with 95% confidence interval (highlighted in grey) is represented, P – nominal *P*-value is calculated from two-sided Pearson correlation. **F** Predictive performance ($r^2$) of gene expression models from discovery resource (HKTR, $n = 478$) vs validation resource (NIH, $n = 222$). r – Pearson correlation coefficient. **G** Percentage of imputable genes (top) and all expressed genes (bottom) in biotypes. Data are coloured by biotype: protein coding gene – pink, long non-coding RNA – red, others – orange. **H.** Examples of imputable kidney genes of relevance to cellular transport of solutes. Associations between GReX of each gene and significantly associated quantitative and disease traits are represented by large arrows showing the directionality of change in the trait (upwards – higher risk or increased blood levels, downwards – lower risk or lower blood levels). Example genes are coloured according to their associated traits and are placed in the region of the nephron with highest expression. Partially created with BioRender.com.

regions and BP is increasingly recognised not only in the pathogenesis of hypertension but also hypertension-mediated cognitive decline[40].

Collectively, our data uncovered new and prioritised existing tissue contributors to BP regulation. We have also provided an additional line of evidence for the role of the kidney as a key mediator of genetic effects on BP regulation and the development of hypertension.

### Development and validation of the prediction models for a genetic component of kidney gene expression – enhanced performance through integration of information from kidney epigenomics and 3D configuration of the genome

We applied Prediction Using Models Informed by Chromatin conformations and Epigenomics (PUMICE)[41], an algorithm which integrates 3D chromatin organisation data and epigenetic annotations, to generate prediction models for genetically regulated gene expression

(GReX) in the human kidney (Fig. 1B). To obtain 3D kidney genome data, we first generated a high-resolution transcription-centred chromatin map using established pipelines from H3K27ac Hi-ChIP in HK2 cell line[42] (Fig. 2B). In total, we obtained 69,036,737 genomic contacts and identified 11,569 Topologically Associating Domains (TADs) and 8934 chromatin loops (Fig. 2B). For epigenomics data, we retrieved information for human adult kidney from ENCODE using 4 different tracks, including H3K27ac (127,379 peaks), H3K4me3 (70,569 peaks), DNase hypersensitivity sites (275,004 peaks) and CTCF (45,832 peaks; Fig. 2B).

By integrating these data with RNA-sequencing-derived transcriptome of 478 kidneys available in the discovery resource using PUMICE (Fig. 1B), we generated 8682 significant gene expression prediction models (Fig. 2C). This is ~24% increase in comparison to 7011 models generated by fitting "traditional" elastic-net linear models on

the same number of kidneys by PrediXcan (Supplementary Data 3). We, then, validated these models in an independent dataset of 222 kidneys collected as our validation resource (Fig. 1C). We found that 6490 kidney PUMICE-derived models remained significant (in comparison to 5647 PrediXcan models; Fig. 2C and Supplementary Data 3–4). *ERAP2* (endoplasmic reticulum aminopeptidase 2) gene is an example of kidney gene with an excellent correlation between the predicted GReX and its observed expression in both our discovery and validation resource (Fig. 2D, E). Overall, there was a very strong correlation (r = 0.83, *P*-value = 1.1 × 10$^{-1647}$) in the predictive performance of PUMICE-derived models between our discovery and validation kidney resources (Fig. 2F). We saw no significant differences in proportions of imputable kidney genes across three main gene biotypes (Fig. 2G).

Collectively, we enhanced the discovery of kidney genes with genetically imputable expression through integration of kidney epigenomic and 3D genomic annotations and validated approximately 75% of the discovered models in an independent collection of the same tissue type.

## The predicted kidney gene expression of metabolic transporters/receptors recapitulates their biological functions and expected contributions to human disease

In addition to validation of the computational robustness for the predictive models, we sought to examine if their imputed gene expression is associated with biochemical readouts of their molecular activity in the kidney. We reasoned that the generated GReX models for the key receptors/transporters involved in the renal handling of urate, glucose, phosphate and calcium should capture the expected contributions of these genes to the circulating levels of these solutes. Amongst kidney genes with robust, independently validated GReX we selected: solute carrier family 2 member 9 gene (*SLC2A9*), solute carrier family 22 member 12 gene (*SLC22A12*), solute carrier family 2 member 2 gene (*SLC2A2*), inositol hexakisphosphate kinase 3 gene (*IP6K3*), calcium sensing receptor gene (*CASR*) as the examples for regulators of urate, glucose, phosphate and calcium transport in tubular epithelium (respectively) (Fig. 2H). We demonstrated that the genetically imputed kidney expression for these genes showed the expected associations with the relevant blood biochemistry phenotypes in UK Biobank; e.g. increased renal expression of *SLC22A12* and *SLC2A9* ([encoding apical URAT1 and basolateral GLUT9] (respectively) – the main transporters responsible for reabsorption of urate in proximal tubule)[43] was associated with increased serum levels of urate in 321,210 individuals from UK Biobank (Fig. 2H and Supplementary Data 5). Moreover, predicted renal expressions of these genes show directionally consistent associations with diseases arising from increased/decreased changes of these phenotypes in blood, e.g., increased predicted kidney expression of *SLC22A12* and *SLC2A9* was associated with increased odds of gout in UK Biobank (Fig. 2H and Supplementary Data 5).

In the absence of information on serum sodium and potassium levels in UK Biobank we could not test their associations with *SCNN1B* (encoding the imputable beta subunit of epithelial sodium channel – ENaC). The latter operates as a key regulator of sodium/potassium handling in the aldosterone-sensitive portion of the distal nephron[44], is targeted by amiloride[45] and represents one of the most consistent gene expression signatures of hypertension-mediated effect on kidney disease[46]. However, we noted that increased level of *SCNN1B* expression was associated with numerically lower and nominally significant risk for hyperkalaemia in UK Biobank consistent with the lowering effect of upregulated ENaCs on circulating levels of potassium (Fig. 2H and Supplementary Data 5).

Taken together, these examples illustrate the biological robustness of the models for genetically predicted expression of kidney genes developed for the purpose of TWAS.

## Kidney transcriptome-wide association studies and computational drug repositioning analysis uncover new genes associated with blood pressure and provide new therapeutic insights

Using 6490 independently validated GReX models we conducted a two-stage reciprocal replication BP kidney TWAS with SBP, DBP and pulse pressure (PP) summary statistics from 337,422 UK Biobank individuals and 299,024 individuals from ICBP (Figs. 1D and S3). In brief, we used each BP summary statistic resource as a discovery population and followed up the significant signals (i.e., kidney genes associated with at least one BP trait) in the other cohort (for the same BP trait) (Fig. S3). Through this reciprocal replication we uncovered a total of 889 unique kidney genes showing statistically significant associations (FDR < 0.05) with at least one BP trait in both datasets (Supplementary Data 6 and Fig. S3). We then mapped these genes onto the existing 429 independent BP GWAS loci (Supplementary Data 6) and found at least one kidney gene within 258 of these loci (60.1%; Fig. S3). Altogether, 772 BP TWAS genes were mappable to the existing BP GWAS loci (Fig. S4). One-hundred and seventeen of these genes mapped outside the known BP GWAS loci (Supplementary Data 6 and Fig. S4) illustrating the potential of TWAS to uncover new associations in the chromosomal regions "missed by GWAS"[47]. 78.7% and 57.9% of kidney genes uncovered by our TWAS were not amongst BP genes from any human tissues examined in TWAS by Giri et al[23]. or multi-tissue panel TWAS examined more recently by Wu et al[48]. (Supplementary Data 7). There was a modest (18%) degree of overlap between kidney genes associated with BP in our TWAS and the genes associated with CKD-defining traits in TWAS conducted by Schlosser et al[49]. (Supplementary Data 7). Amongst those that overlap were several notable genes linked already to both BP and kidney health/disease including interferon regulatory factor 5 gene (*IRF5*)[24,25], *N*-Acetyltransferase 8B gene (*NAT8B*)[32,50] and Dipeptidase 1 gene (*DPEP1*)[32,51].

We then examined the output from our BP kidney TWAS via Connectivity Map (CMap)[52,53] – a library of gene expression changes induced by a panel of 1309 different FDA-approved drugs and small chemical compounds. We sought to identify drugs/compounds both inducing and reversing changes in gene expression associated with BP – pharmaceuticals with reversed direction of the effects on gene expression (to that of BP) can be interpreted as potential repurposing/repositioning options for hypertension (Fig. 1E).

Based on the reversal of the BP-related changes in the transcriptome, adenylyl cyclase activators were identified as a group of medications with a potential to lower BP (Supplementary Data 8). This is in line with their effect on adenylate cyclase and an elevation of intracellular cyclic AMP (cAMP)[54] leading to vascular smooth muscle relaxation and subsequent vasodilation[55]. cAMP-dependent effect of forskolin [natural root extract from *Coleus barbatus* (Blue Spur Flower)] showed a potential to protect podocytes from injury[56] and already emerged as a new drug repurposing opportunity for kidney diseases[57]. Forskolin is available as a diet supplement[58] and was associated with several health benefits before[59].

Our data support previously reported side effect of topoisomerase inhibitors (commonly used to treat acute myeloid leukaemia) on BP (hypotension)[60]. We also demonstrated both known (e.g., glucocorticoids) and less recognised (e.g., tubulin polymerisation inhibitors) potential of several groups of therapeutics to increase BP based on the direction of their effects on gene expression (i.e., synchronous with that of BP) (Supplementary Data 8). The data on tubulin polymerisation inhibitors are in line with one of our most recent studies showing how docetaxel induced endothelial dysfunction and hypertension[61].

Collectively, our studies identified almost 900 genes whose predicted kidney expression show directionally consistent associations with SBP, DBP and/or PP across two independent cohorts providing a robust input for further downstream analyses. We also identified a new potential pharmaceutical repositioning opportunity for hypertension

and highlighted examples of BP elevating effects of the existing medications with indications for conditions other than hypertension.

## Mendelian randomisation and fine-mapping of causal gene sets uncovers putatively causal independent associations between kidney genes and blood pressure

We then sought to substantiate the evidence of causal effects of 889 BP genes identified in our BP kidney TWAS on BP traits (Figs. 1E and S3–4). We integrated the summary statistics information from *cis*-eQTL analysis [generated using 478 kidneys from the Human Kidney Tissue Resource (HKTR)] and GWAS summary statistics for SBP, DBP and PP (generated using ~750,000 individuals from both UK Biobank and ICBP) in two-sample Mendelian randomisation (MR) designed for TWAS applications[62]. We determined that 663 kidney genes from our TWAS analysis showed robust evidence of potentially causal effect on at least one BP trait (FDR < 0.05; Fig. S4).

We noticed that 510 of these genes cluster with at least one other gene within the same locus (Supplementary Data 9) – we identified a total of 133 loci with two or more BP kidney genes showing potentially causal associations with BP (Supplementary Data 9). To account for a potential effect of linkage disequilibrium-driven correlations between genes in these regions on our findings we applied FOCUS-based analysis[63]. Within 35 of these regions, FOCUS pointed to a single BP kidney gene. For example, within a locus on chromosome 17, the initial set of ten genes showing causal effects on DBP, FOCUS prioritised N-Myristoyltransferase 1 gene (*NMT1*) whose primary function is a co-/post-translational modification of proteins through the addition of a fatty acid (myristate) to the N-terminal glycine residue[64,65] (Supplementary Data 10).

Altogether, our MR studies followed by fine-mapping of causal gene sets prioritised 399 kidney genes showing a potentially causal effect on BP (Figs. 3 and S4). Nearly half (182, 45.6%) of them mapped as a single gene onto specific BP GWAS/TWAS locus (Supplementary Data 9 and Fig. S4) and 29 of them had causal effects on all three BP traits (Fig. 3).

We assigned each of these 399 genes into one of 16 biological master themes based on their known function and relevance to human health and disease (Supplementary Data 11). We observed a very strong footprint of human metabolism (Fig. 3 and Supplementary Data 11) among BP kidney TWAS genes. This included biochemical pathways responsible for metabolic processing of amino acids [e.g. agmatinase gene (*AGMAT*) and serine racemase gene (*SRR*)], carbohydrates [e.g. solute carrier family 5 member 11 gene (*SLC5A11*), solute carrier family 2 (facilitated glucose transporter), member 4 gene (*SLC2A4*) and starch binding domain 1 gene (*STBD1*)] lipids [(e.g. Acyl-CoA Thioesterase 8 gene (*ACOT8*), ELOVL Fatty Acid Elongase 7 gene (*ELOVL7*) and fatty acid desaturase 1 (*FADS1*)], vitamins [e.g. Folate receptor alpha gene (*FOLR1*)] and oxidative phosphorylation [(e.g. Inner Membrane Mitochondrial Protein gene (*IMMT*) and Nicotinamide Nucleotide Transhydrogenase gene (*NNT*)] in line with a theory of metabolic roots of hypertension[66] and increasingly appreciated role of the kidney as a key regulatory organ of human metabolism beyond its contributions to fluid-ion homeostasis[67].

A few of these genes have an established role in the physiological maintenance of sodium-water homeostasis and our data demonstrate their contributions to BP elevation. For example, increased renal expression of aquaporin 1 gene (*AQP1*) and aquaporin 4 gene (*AQP4*) – water-selective channels operating in the proximal tubule/thin descending limb of Henle/descending vasa recta and principal cells of the collecting duct (respectively)[68] showed causal effects on increased SBP and DBP (Fig. 3). This is most consistent with the increase in both constitutive (e.g. via *AQP1*)[68] and arginine-vasopressin (AVP)-regulated[69] (e.g. via *AQP4*) reabsorption of water and their role in hypertension proposed in experimental models[70,71]. Together with the previous findings on aquaporin 11 gene (*AQP11*)[25], this also means that

one third (3/9) of renal aquaporins[68] are mediators of the genetically determined predisposition to elevated BP.

Overall, our TWAS-driven studies identified 7.5-fold greater number of kidney genes whose expression shows a potentially causal effect on BP than our earlier MR analyses (*n* = 53)[25]. This is also over 2-fold increase in the discovery over the number of genes identified by integration of BP GWAS and three different types of kidney omics[25].

## Triangulation of outputs from plasma proteomics and metabolomics with kidney transcriptome-wide association studies yields new insights into pathways of blood pressure regulation

Genetic analyses of plasma proteomics and metabolomics are increasingly used to gain insights into the pathogenesis of complex traits including chronic kidney disease (CKD) and hypertension[51,72]. We sought to explore how such layers of omics may help in functional interpretation of findings from kidney TWAS using *SLC5A11*, agmatinase gene (*AGMAT*) and angiotensinogen gene (*AGT*) as examples (Fig. 4).

We found that reduced renal expression of *SLC5A11* is causally associated with increase in PP (Fig. 4A, B). This gene encodes sodium/glucose cotransporter 6 (SGLT6, SMIT2). Given a well-established role of this family of transporters [e.g. solute carrier family 5 member 1 gene (*SLC5A1*, SGLT1) and solute carrier family 5 member 2 gene (*SLC5A2*, SGLT2)] in glucose homeostasis[73], we firstly examined the association between renal expression of *SLC5A11* and levels of glucose and HbA1C using data from 337,350 unrelated European individuals from UK Biobank. Having found no association with either (Supplementary Data 12), we then sought to determine whether the key target substrate of SLC5A11 [(i.e. myo-inositol – a cyclic carbohydrate of importance to signal transduction and osmoregulation)[74]] may act as a mediator of its association with BP. Using data from 14,296 individuals from two cohorts (INTERVAL and EPIC-Norfolk)[75] we confirmed that renal *SLC5A11* was indeed associated with circulating levels of myo-inositol (Fig. 4C). This is in line with the key role of the kidney in myo-inositol metabolism[74]. We then uncovered an inverse association between myo-inositol and PP (Fig. 4C). Through further mediation analysis, we determined that ~48.4% of effect of *SLC5A11* renal expression on PP is mediated by serum levels of myo-inositol. Indeed, myo-inositol depletion was linked before to several metabolic disorders such as insulin resistance and polycystic ovary syndrome[74]; its increased urinary excretion was highlighted as a marker of CKD progression[76]. These findings show that a reduction in circulating concentrations of myo-inositol may contribute to increased BP at least in part because of genetically determined drop in expression of its key renal cotransporter. Our data also suggest that increasing levels of myo-inositol (e.g. through enhancing *SLC5A11*-dependent reabsorption in the proximal tubule[77]) may be beneficial to BP control.

Our BP kidney TWAS also demonstrated that genetically determined increase in renal expression of *AGMAT* was causally associated with increased SBP (Fig. 4D, E). *AGMAT* is responsible for enzymatic conversion of agmatine to putrescine downstream from arginine on the alternative pathway for polyamine biosynthesis[78] (Fig. 4D). It shows a strong enrichment in the kidney (Fig. S5) and a cell-type specific enrichment in proximal tubule (https://www.proteinatlas.org/ENSG00000116771-AGMAT/single+cell+type). While we could not examine how kidney *AGMAT* expression correlates with putrescine levels, we showed that genetically determined kidney expression of *AGMAT* mRNA is associated with circulating blood urea nitrogen (BUN) levels – a by-product of its enzymatic activity (i.e. agmatine + H2O = putrescine + urea) in two independent populations (Fig. S6). Using data of 14,296 individuals from two cohorts (INTERVAL and EPIC-Norfolk)[75] we also confirmed the association between renal *AGMAT* and plasma levels of 4-guanidinobutanoate (known also as gamma-guanidinobutyric acid or gamma-guanidinobutanoate) – a metabolite

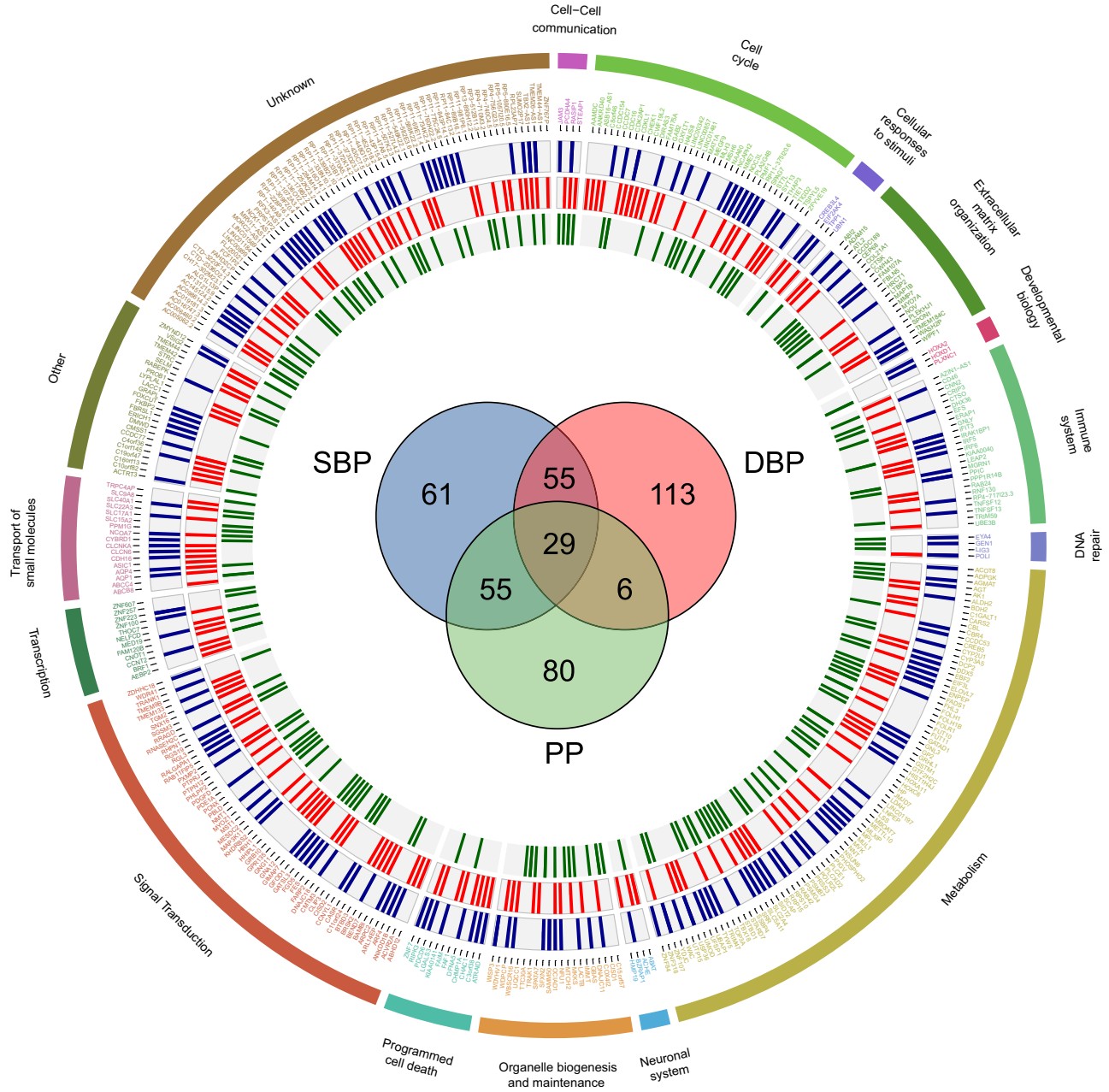

**Fig. 3 | Circular representation of information on 399 putatively causal genes for blood pressure and the degree of their shared association with SBP (blue), DBP (red) and PP (green).** Genes are grouped by their biological theme (shown as coloured regions). From outermost to inner most data circle: associations with systolic blood pressure (SBP) coloured in blue, associations with diastolic blood pressure (DBP) coloured in red and associations with pulse pressure (PP) coloured in green.

whose salivary levels were linked to *AGMAT* before[79] (Fig. 4F). We also found an association between BP and circulating levels of 4-guanidinobutanoate (Fig. 4F). The latter was previously proposed to act in a pro-inflammatory manner[80,81]. Finally, through further mediation analysis we determined that approximately 12.3% of effect of renal *AGMAT* expression on SBP is mediated by increased levels of 4-guanidinobutanoate in plasma (Fig. 4F). These data demonstrate the importance of renal catabolism of arginine in BP regulation and uncover an arginine-derived compound as a new metabolic readout of high BP.

Our TWAS also uncovered a consistently strong causal association between increased renal expression of angiotensinogen gene (*AGT*) and BP (Fig. 4G, H). Expression of angiotensinogen mRNA has been thought to influence BP through intra-renal RAS activity (Fig. 4G) and we sought to quantify the extent to which the effect we detected is indeed independent of systemic (circulating) angiotensinogen. Using data from 10,708 individuals from the Fenland study[82] with available plasma levels of angiotensinogen protein we first confirmed the causal association between the plasma angiotensinogen and BP in the expected direction (Fig. 4I). Further

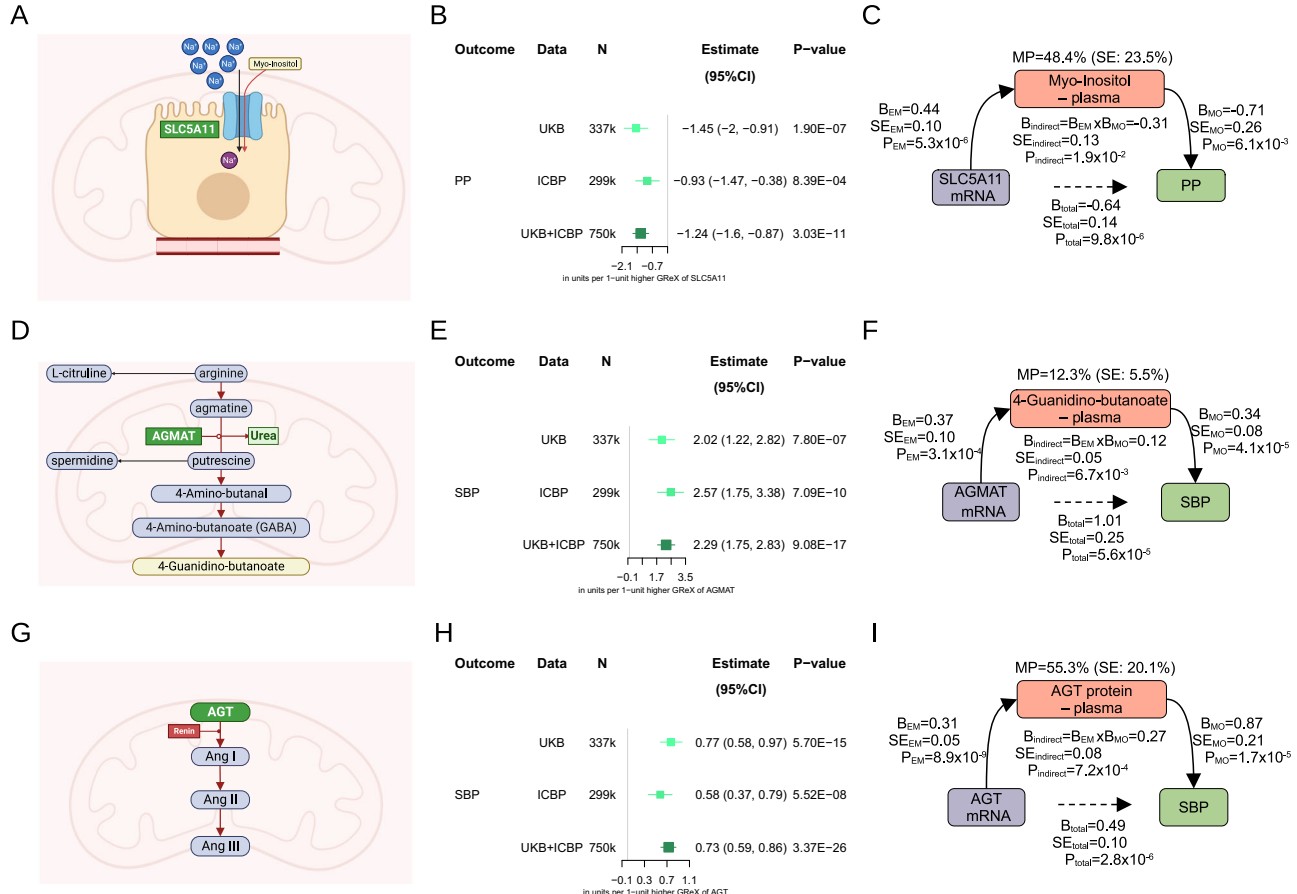

**Fig. 4 | Genetically predicted expression of kidney genes and their biochemical readouts – integrative multi-omics analysis. A** Localisation of the sodium/myo-inositol cotransporter 2 (SMIT2), encoded by the *SLC5A11*, in the apical pole of renal proximal tubular cells. SMIT2 facilitates the co-transport of Na+ and myo-inositol across the cell membrane. Created with BioRender.com. **B** Effects of genetically regulated expression (GReX) of kidney *SLC5A11* on pulse pressure (PP) from transcriptome-wide association study (TWAS). Nominal *P*-value is calculated from two-sided *Z*-score test. UKB – UK Biobank, ICBP – International Consortium for Blood Pressure, Estimate – change of PP in units per one-unit higher GReX of *SLC5A11*, CI – confidence interval. **C** Representation of 58.4% effect of *SLC5A11* mRNA expression on PP mediated by plasma levels myo-inositol. Nominal *P*-value is calculated from two-sided *Z*-score test. MP – mediation proportion, B – estimated effect, SE – standard error of the estimated effect, EM – from exposure to mediator, EO – from exposure to outcome, MO – from mediator to outcome, indirect – indirect effect from exposure to outcome, total – total effect from exposure to outcome. **D** The polyamine pathway of arginine catabolism. Agmatinase, encoded by the *AGMAT*, catalyzes the reaction between agmatine and putrescine, resulting in the production of urea. Created with BioRender.com. **E** Effects of kidney *AGMAT* (GReX) on systolic blood pressure (SBP) from TWAS. Nominal *P*-value is calculated from two-sided *Z*-score test. Estimate – change of SBP in units per one-unit higher GReX of *AGMAT*. **F** Representation of 11.7% effect of *AGMAT* mRNA expression on SBP mediated by plasma levels of 4-guanidinobutanoate. Nominal *P*-value is calculated from two-sided *Z*-score test. **G** Renin catalyzes the reaction from *ANG* (angiotensinogen) to Angiotensin I (AngI), which is subsequently converted to AngII by Angiotensin-Converting Enzyme (ACE). Created with BioRender.com. **H** Effects of kidney *AGT* mRNA (GReX) on SBP from TWAS. Nominal *P*-value is calculated from two-sided Z-score test. Estimate – change of SBP in units per one-unit higher GReX of *AGT*. **I** Representation of 52.7% effect from *AGT* mRNA expression on SBP mediated by circulating plasma protein levels of angiotensinogen. Nominal *P*-value is calculated from two-sided Z-score test. In 4B, 4E and 4H, squares are positioned by the estimated effects with horizontal error bars illustrating 95% confidence intervals of the estimated effects.

mediation analysis demonstrated that approximately 44.7% of the effect of kidney *AGT* mRNA on BP was independent of circulating plasma angiotensinogen (Fig. 4I) and possibly reflective of the local activity of angiotensinogen in the kidney[83]. Interestingly, a significant proportion of renal angiotensinogen mRNA effect on BP appeared to be mediated by plasma levels of angiotensinogen (Fig. 4I). This may suggest either a shared genetic regulation of angiotensinogen mRNA expression between the kidney and the key tissue(s) from where it is released into circulation (i.e. liver)[83] or/and a largely unrecognised systemic effect of renal angiotensinogen on BP.

Collectively, these studies exemplify how integration of genomics and kidney transcriptomics with data from other "omics" (e.g., proteomics and metabolomics) can provide insights into molecular mechanisms underpinning the findings from TWAS, identify the downstream effectors of kidney TWAS genes and highlight new biochemical readouts of elevated BP.

## Genetically regulated expression of miRNAs in the kidney is associated with blood pressure

Several miRNAs have been proposed to contribute to BP regulation and the development of hypertension mostly through gene expression-phenotype correlation studies[84,85]. To systematically examine whether kidney miRNAs are associated with BP we first created a repository of 339 miRNA expression profiles with matching genotype information in our discovery resource (HKTR). We uncovered 1459 kidney miRNAs, a majority of which are encoded by intronic sequences (Fig. 5A). As expected, miRNAs accounted for a much smaller proportion of kidney genes when compared to protein-coding genes and long non-coding RNAs (Fig. 5B). Fewer kidney miRNAs than

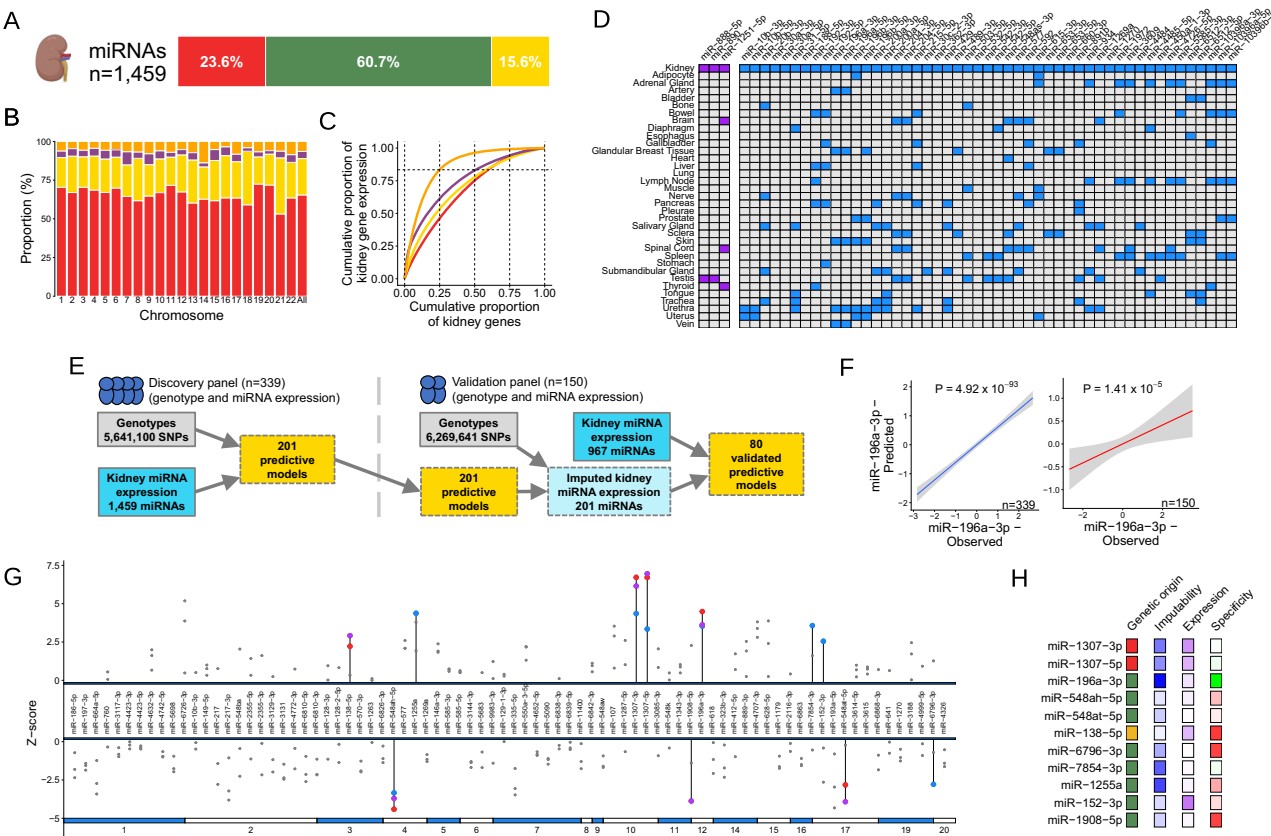

**Fig. 5 | Kidney miRNAs and blood pressure. A** The number of miRNAs expressed in the human kidney and the distribution of their genetic origins. Red – exonic, green – intronic, yellow – intergenic. Partially created with BioRender.com. **B** Cumulative proportion of gene biotypes expressed in the kidney, stratified by chromosome. Red – protein-coding genes, yellow – long non-coding RNAs, purple – pseudogenes, orange – miRNAs. **C** Cumulative proportion of the sum of miRNA expression ($\log_2(TPM+1)$) and gene expression ($\log_2(TPM+1)$) against the total number of miRNAs and genes ranked by their expression in descending order. Red – protein-coding genes, yellow – long non-coding RNAs, purple – pseudogenes, orange – miRNAs. **D** Tissue enrichment profiles of 52 miRNAs with highest level of expression enrichment/enhancement in the kidney – data across 35 tissues. Blue – tissue enhanced (expression 4x higher than the cross-tissue mean), purple – group enriched (group of 2–5 tissues with expression 4x higher than any other tissue), grey – no enrichment. **E** Predictive miRNA expression model workflow used in TWAS. Grey – genotype, blue – gene expression, yellow – predictive models. **F** Correlation between the predicted GReX and observed expression of miR-196-3p. The best-fitting line with 95% confidence interval (highlighted in grey) is represented. Blue –

HKTR ($n = 339$), red – NIH kidney validation resource ($n = 150$). P – $P$-value is calculated from two-sided Pearson correlation. **G** Overview of association between 80 imputable, validated kidney miRNAs and blood pressure traits, ordered by chromosome. Red – significantly associated with SBP, blue – significantly associated with DBP, purple – significantly associated with PP, grey – non-significant association. **H** Selected characteristics of miRNAs significantly associated with at least one blood pressure trait. 'Genetic origin' represents the location of each miRNA relative to other genes. Red – exonic, green – intronic, yellow – intergenic. 'Imputability' represents the degree of imputability of each miRNA as determined by the predictive model $r^2$. Blue – higher imputability, white – weaker imputability. 'Expression' represents the expression fold-change of each miRNA in comparison to the most highly expressed kidney miRNA. Purple – strongly expressed, white – weakly expressed. 'Specificity' represents the expression fold-change of each miRNA in comparison to the cross-tissue mean. Green – most highly expressed in the kidney, white – similar expression in the kidney to other tissues, red – most highly expressed in tissues other than the kidney.

other biotypes accounted for the same proportion of cumulative gene expression in the kidney (Fig. 5C).

We then examined the degree of miRNA specificity to kidney tissue by integrating our catalogue of renal miRNAs with expression profiles of 35 human tissues curated by miRNATissueAtlas2[86]. Of 1431 miRNAs overlapping between our kidney catalogue and miRNA-TissueAtlas2, 49 and 3 fulfilled the criteria of kidney enhanced and kidney group enriched[87], respectively (Fig. 5D). MiR-30a-3p, miR-30a-5p and miR-188-5p showed strong enrichment in the kidney and some of them have prior evidence of contributions to kidney disease (Fig. 5D)[88,89].

Using 339 kidneys from our discovery resource we then generated and cross-validated GReX prediction models for 201 kidney miRNAs (Fig. 5E). Of these, 143 were available for validation and 80 of these were validated in an independent resource of 150 National Institutes of Health (NIH) kidneys (TCGA and CPTAC) with matching genotype and small RNAseq-derived expression profiles (Fig. 5E, F and

Supplementary Data 13). miR-196a-3p is an example of a miRNA whose strong genetic regulatory component renders it an excellent target for TWAS (Fig. 5F).

We then used a computational pipeline with reciprocal replications in two cohorts established at earlier stages to conduct BP kidney microRNA-TWAS (Fig. S7). We identified 11 kidney miRNAs whose genetically imputed expression was associated with BP across both cohorts (Fig. 5G and Supplementary Data 14). They mapped to nine independent BP GWAS loci and represented different spectra of kidney abundance, tissue specificity and genetic origin (Fig. 5H and Supplementary Data 15).

Given the intragenic (exonic/intronic) DNA origin for a majority of BP kidney miRNAs, we then examined whether their host genes could act as BP TWAS genes and whether they may account for the detected associations with BP. Out of ten intragenic miRNAs, only one – hsa-miR 1908-5p – had a kidney BP TWAS gene as the host gene (*FADS1*) (Supplementary Data 16). However, further analyses showed that the

association between the hsa-miR 1908-5p and BP was not mediated by *FADS1* – the only BP TWAS gene in the locus (Supplementary Data 16). We then extended the mediation analysis to all BP kidney TWAS genes mapping onto the proximity of nine remaining intragenic kidney miRNAs. For eight of the kidney miRNAs with at least one BP TWAS gene in the locus, we found only one gene (Homeobox C6, *HOXC6*) accounting for the mediation signals between kidney miRNAs (hsa-miR-196a-3p) and BP (Supplementary Data 16).

Taken together, our data provided insights into the landscape of miRNAs expressed in the kidney. We uncovered new associations between BP and kidney miRNAs and showed that the associations between BP and kidney miRNAs are not necessarily mediated via their host genes. Finally, we demonstrated that in some cases other BP-associated genes in proximity to the miRNA may act as the mediators of their associations with BP.

## Kidney proteome-wide association studies uncover new proteins associated with blood pressure and provide an orthogonal validation for blood pressure associations uncovered at the transcriptome level

Proteins are the key effector molecules translating biological signals inherited in DNA and/or transcribed in mRNA into phenotypic differences between individuals including their health and susceptibility to disease. It is not clear whether genetic determinants of BP are associated with changes of protein expression in the human kidney and whether the associations between BP genetic variants and the renal mRNA expression have a functional impact at the protein level. Using 72 NIH-CPTAC human kidneys with tissue proteome characterised by liquid chromatography mass spectrometry, we identified 7,291 proteins; with the majority classified as predicted intracellular proteins (89%) (Supplementary Data 17). The proportions of soluble, membrane-bound and secreted proteins (Fig. 6A) were consistent with the data from tissue-based map of the global human proteome[90]. Of 452 HPA genes with elevated expression in the kidney (when compared to other tissues), 223 were identified in the NIH-CPTAC dataset (Supplementary Data 18). As expected, we found the strongest enrichment for proteins encoded by genes with highest level of specificity to the kidney (Fig. 6B).

We then examined how the abundance of proteins known for their relevance to BP regulation and hypertension correlates with the expression of their respective genes. A total of 7036 genes quantified by RNA-sequencing had their abundance measured at the protein level in the NIH-CPTAC dataset. Genes known for Mendelian hypertension/hypotension syndromes[91] and kidney targets for BP-lowering medications were significantly enriched amongst genes with the highest positive correlation with their respective proteins ($P$-value = $2.5 \times 10^{-2}$ and $P$-value = $1.5 \times 10^{-3}$, respectively) and the magnitude of enrichment was comparable to HPA kidney-enriched/enhanced genes (Fig. 6C). For example, *SLC12A1* (encoding bumetanide-sensitive sodium-(potassium)-chloride cotransporter 2 – a target for loop diuretics) and *SLC12A3* (encoding a thiazide-sensitive sodium-chloride cotransporter – a target for thiazides) were within the top 25 kidney genes showing very strong mRNA-protein correlations (Supplementary Data 19 and Fig. 6D). Kidney genes with evidence of causal association with BP (Supplementary Data 20) were also enriched (although to a lesser degree) for significant positive correlations with the respective proteins (Fig. 6C).

Of 815 proteins, whose kidney expression was genetically imputable (Supplementary Data 21), 97 showed reciprocal association with at least one BP trait in two independent cohorts (UK Biobank and ICBP) (Supplementary Data 22 and Figs. 6E and S8). Of these, 57 had no evidence of association with BP at mRNA level because either their GReX prediction model did not converge ($n = 34$) or there was no or only weak BP signal in our kidney TWAS analysis (Supplementary Data 23). This may indicate either the existence of differences in the

genetic regulation of their mRNA and protein expressions contributing to BP/hypertension or/and reflect a false-positive finding in proteome-wide association studies (PWAS).

For 46 BP-associated kidney proteins we identified additional evidence for relevance to BP/hypertension at other molecular levels (Supplementary Data 23 and Fig. 6F). Indeed, parent genes of 40 proteins showed association with BP in our kidney BP TWAS (Supplementary Data 23 and Fig. 6F) and six others had an additional layer of prior evidence for association with BP in our previous kidney QTL studies[25] (Supplementary Data 23 and Fig. 6F).

Overall, BP PWAS proteins were enriched for being therapeutically tractable (Fig. 6G). Indeed, when compared to a random set of kidney proteins, our set of 97 BP PWAS proteins showed a strong enrichment for proteins with discovery potential (i.e., not currently targeted) by small molecule and proteolysis targeting chimera (PROTAC) modalities (Fig. 6G).

Our pathway enrichment analysis further revealed enrichment of BP-associated proteins for hypertensive crisis (Fig. 6H and Supplementary Data 24) as well as enrichment for RAS and proteins involved in innate and adaptive immunity (Fig. 6H and Supplementary Data 24). This is consistent with a rarely appreciated role of the kidney as a tissue contributor to immune activation[38] and antimicrobial defence[92]. We also observed an enrichment of BP-associated proteins within metabolic pathways including those involved in detoxification of reactive oxygen species, arachidonic acid metabolism, tyrosine metabolism and fatty acid metabolism; in particular mitochondrial fatty acid beta-oxidation (Fig. 6H, Supplementary Data 24). Indeed, genetically determined reduction in three proteins of key importance to mitochondrial beta oxidation of fatty acids including long-chain specific acyl-CoA dehydrogenase, mitochondrial (ACADL), medium-chain specific acyl-CoA dehydrogenase, mitochondrial (ACADM) and Acyl-CoA Dehydrogenase Family Member 10 (ACAD10) were associated with increased BP (Supplementary Data 22).

Collectively, we show that kidney proteins showing positive correlations with their parent mRNAs are enriched for genes of relevance to BP, genetically mediated hypertension/hypotension and antihypertensive treatment. This substantiates the evidence for informativeness of kidney transcriptome as a source of information on biological underpinnings of hypertension in the absence of larger datasets on kidney proteome. Through triangulation with other omics, we further corroborate the evidence behind relevance of kidney proteins to hypertension. Such proteins (i.e., with evidence of genetic contribution to their abundance at multiple molecular levels) are of utmost clinical interest, e.g. for druggability studies.

## Transcriptome profiling of cells harvested from urine yields non-invasive insights into expression of kidney genes including those of relevance to blood pressure

Human kidneys shed their epithelial cells (both of glomerular and tubular origin) into urine[93–96] and RNA-based analysis of urinary cells has been proposed as a new strategy with potential to inform kidney diagnoses[97–99].

We have generated 33 RNA-sequencing derived transcriptomic profiles of urinary cell pellets from individuals recruited into our HKTR (Supplementary Data 25). We confirmed that a set of 12 RNA-sequencing quality control metrics (generated by RNASeQC[100] produce either similar (e.g. "Expression profiling efficiency") or superior (e.g. "Mapping rate") values in urine compared to saliva (Fig. S9). We detected expression of 21,981 genes in urinary cells and similar numbers of the major gene biotypes expressed in urine compared to human kidney tissue (Fig. 7A). Transcriptome complexity was broadly similar between urinary cells and kidney tissue (Fig. 7B). We confirmed there was no obvious transcriptomic differences due to different donor sources of urine in our study (i.e., nephrectomy and kidney

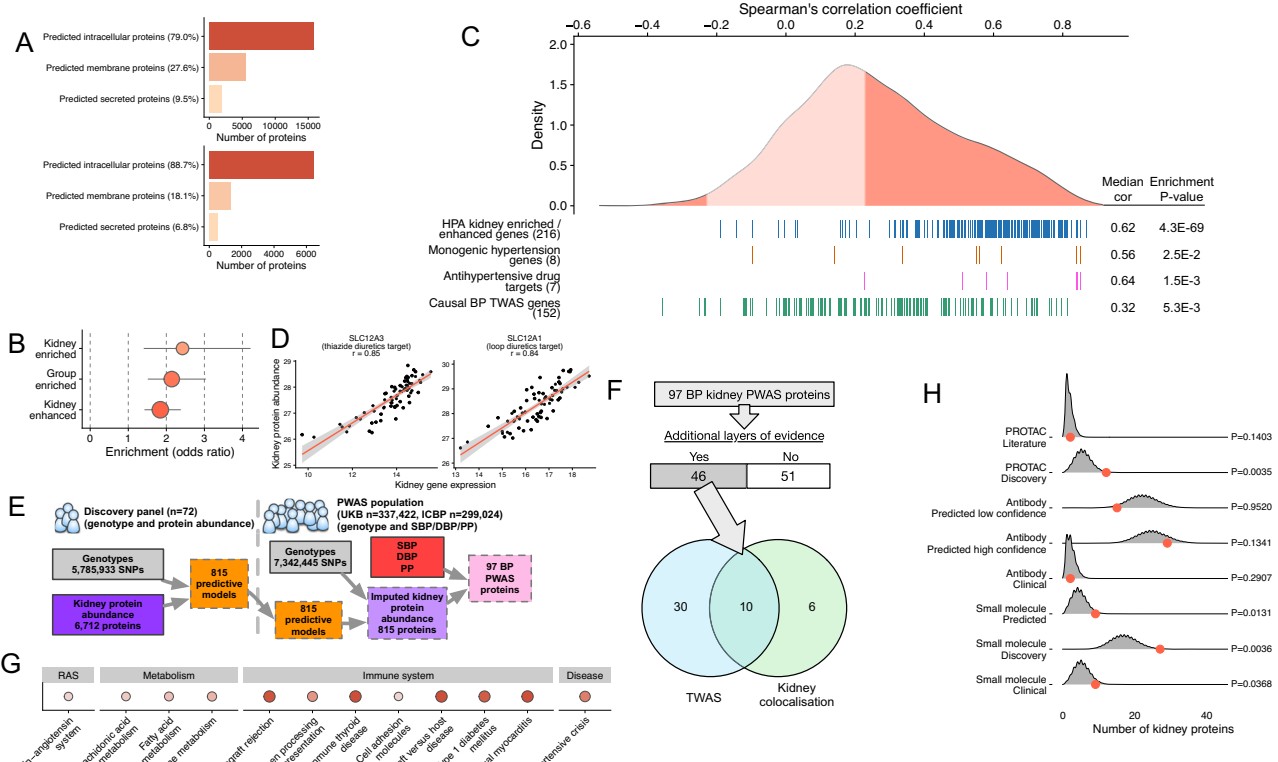

**Fig. 6 | Kidney tissue proteomics and blood pressure PWAS. A** Classification of 20,082 proteins (from the Human Protein Atlas (HPA)) and 7,291 measurable proteins in human kidney (from CPTAC) by predicted localisation. Percentage of all measurable proteins in each class is shown, a single protein may be predicted to belong to more than one category. **B** Enrichment for tissue specificity (kidney enriched, kidney enhanced, and group enriched genes) in measurable kidney proteins (6608 proteins) compared to all the HPA proteins (20,082 proteins). Tissue enrichment and enhancement status is taken from the HPA. Enrichment is calculated by two-sided Fisher's exact test, coloured circles represent statistically significant enrichment results, increasing size and red colour intensity denotes a larger negative $\log_{10}$ *P*-value. Circles are positioned by enrichment odds ratio with horizontal error bars showing 95% confidence intervals on the odds ratio. **C** Density of Spearman's correlation coefficients between protein abundance and gene expression for 6712 protein/gene pairs, with associated enrichment estimates for HPA kidney enriched genes, monogenic hypertension/hypotension genes, anti-hypertensive drug targets and kidney TWAS genes showing causal association with BP (derived from this study). Enrichment *P*-value was calculated by a one-sided two-sample Kolmogorov Smirnov test. **D** Correlation between kidney tissue gene expression and tissue protein abundance for two blood pressure lowering therapeutic targets (*SLC12A3* and *SLC12A1*). Error bands represent 95% confidence intervals. **E** Overview of human kidney tissue PWAS workflow. Objects are coloured

by their data type: genotype – grey, protein abundance – purple, predictive model – orange, phenotype – red, BP PWAS proteins – pink. **F** Of 97 BP PWAS proteins, 46 have at least one additional level of evidence for association with BP. Number of genes is shown in each area of the Venn diagram. TWAS (blue circle) – PWAS proteins with evidence from BP kidney TWAS analysis, Kidney colocalisation (green circle) – PWAS proteins with evidence from previous colocalisation analyses with BP (Eales et al.[25]). **G** Overrepresented KEGG pathways and Human Phenotype Ontology diseases in 97 BP PWAS proteins (one-sided hypergeometric test). All pathways significant at 5% FDR are shown as coloured circles, circles are sized by nominal *P*-value (large – most significant, small least significant) and coloured by magnitude of overrepresentation (light pink – least positive overrepresentation, dark red – most positive overrepresentation). Pathways are grouped by shared overlapping genes (>65%) and have been manually classified into themes based on the known function of overlapping genes. **H** Enrichment for drug tractability in 97 BP PWAS proteins (permutation test) across three therapeutic modalities. Distributions shown are smoothed density estimates of category counts across 100,000 randomly permuted gene sets of equal size to BP PWAS proteins (*n* = 97). Red point denotes observed count for each tractability category in the 97 BP PWAS proteins. *P*-values are calculated by one-sided permutation test. PROTAC proteolysis-targeting chimera.

biopsy, Fig. 7C). This is in line with our previous observations on transcriptomic profiles of kidney tissue specimens[84].

We hypothesised that functional annotations of genes with the strongest expression in cells harvested from urine and the kidney will show a strong overlap. Our analysis of the 100 mostly highly expressed genes in urine and kidney tissue showed an over-representation for 33 and 35 pathways, respectively (KEGG pathways and Gene Ontology Biological Process terms, Supplementary Data 26), 24 (73%) of which were common to both urinary cells and the kidney (Fig. 7D) and there were 44 distinct overrepresented pathways (Fig. 7D). These pathways were then manually grouped into 6 biological themes by overlap of shared genes. Most of the common pathways mapped onto immunity theme are consistent with the key role of the urinary tract in immune activation and defence against infections[38,92]. The top 100 most highly expressed

genes in urinary cells also showed enrichment for pathways reflective of the renal contributions to ion exchange (Fig. 7D). We also noted that the genes in urinary cells showed enrichment for glucose metabolism, glycolysis and gluconeogenesis (Fig. 7D). The latter is consistent with the renal origin of these cells given that apart from the liver, kidney (mainly proximal tubule) is the only other organ capable of de novo glucose production[101,102].

We then examined the extent of transcriptomic similarity between urinary cells from our dataset and 54 human tissue types from GTEx. Kidney cortex and kidney medulla showed the highest degree of correlation in expression of 19,273 protein-coding genes with urinary cells (Fig. 7E and Supplementary Data 27) at *r* = 0.81 and *r* = 0.80, respectively (Fig. 7F). We confirmed the magnitude of the correlation between urinary cells and the kidney using an independent (to GTEx) set of 430 kidney tissue samples from HKTR (Fig. 7F).

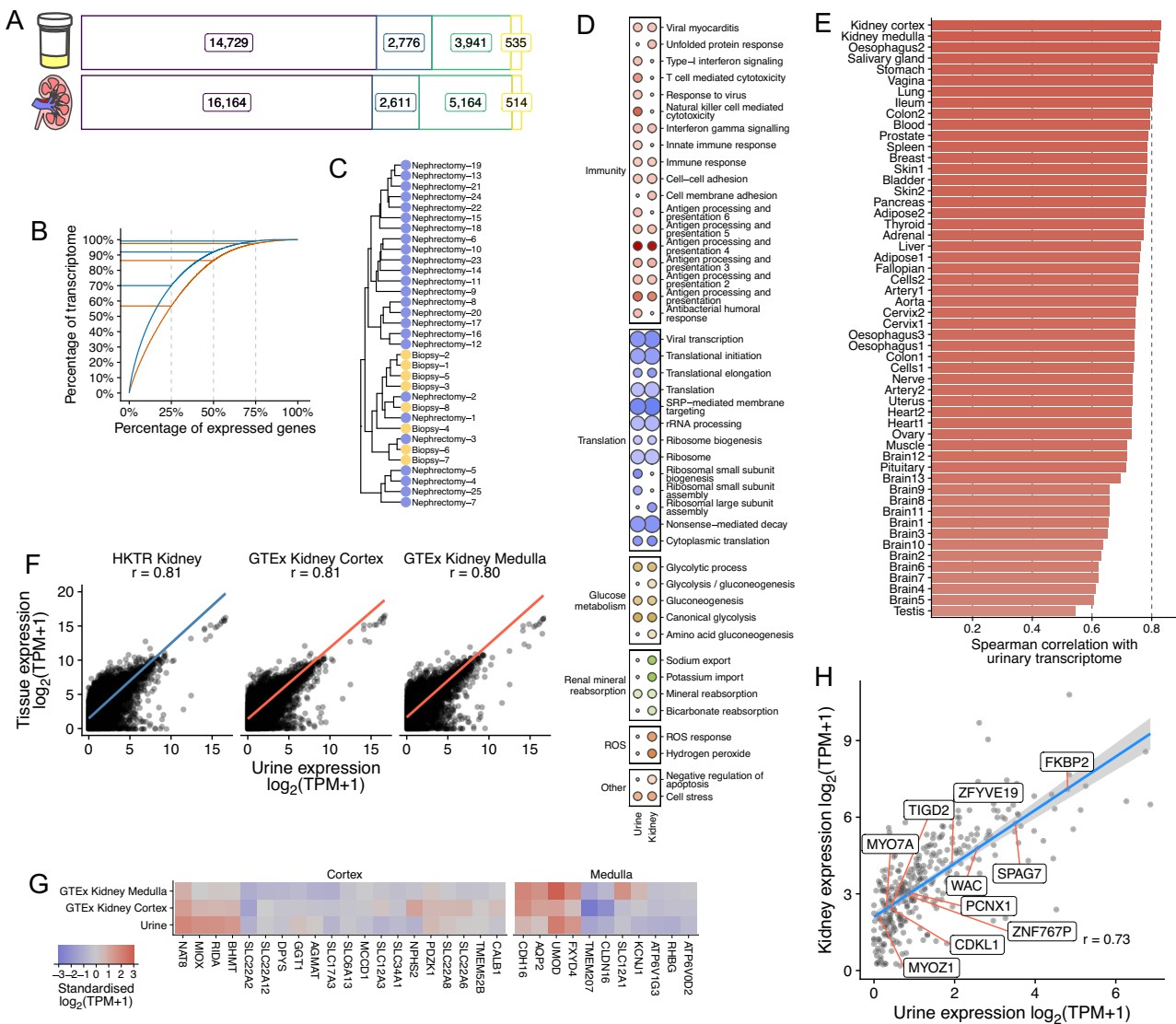

**Fig. 7 | Transcriptomic profiling of urinary cells by RNA-sequencing. A** Number of expressed genes from urinary cell pellets ($n = 33$) and kidneys ($n = 430$) by major gene biotypes. Purple – protein coding, blue – long non-coding, green – pseudogenes, yellow – short non-coding genes. **B** Transcriptome complexity for urinary cells and kidneys. Expressed genes are ordered from most to least highly expressed on the x-axis and the cumulative proportion of the overall transcriptome attributable to those genes is shown on the y-axis. Horizontal lines identify the proportion of overall transcriptome expression attributable to the top 25, 50 and 75% of genes in kidney (orange) and urine (blue) RNA-sequencing samples. **C** Hierarchical clustering of 33 urinary cell transcriptomes from nephrectomy and biopsy samples. Samples are hierarchically clustered using Pearson's correlation distances calculated from $\log_2(TPM+1)$ gene expression values of all expressed genes in urine. Nephrectomy samples are coloured blue and biopsy yellow. **D** Overrepresented KEGG pathways and GO biological processes in the 100 most highly expressed genes in urinary cells and kidneys characterised by RNA-sequencing. Over-representation analysis (one-sided Fisher's exact test) results significant at 5% false discovery rate. Results are grouped by biological themes determined by manual grouping of genes present in each pathway. Fold enrichment is represented by colour from pale to dark red (least to greatest enrichment). Statistical significance (negative $\log_{10}$ $P$-value) is represented as the size of each circle. Non-significant results are represented by small grey circles. Some pathway names have been abbreviated or simplified for brevity. ROS – reactive oxygen species. SRP – signal

recognition particle. **E** Transcriptome similarity between the urinary transcriptome and 54 GTEx tissues. Similarity is calculated as the Spearman's correlation coefficient between median $\log_2(TPM + 1)$ expression of 19,273 expressed protein-coding genes. Tissue names have been abbreviated using alphabetic ordering. Colour denotes the magnitude of similarity from weak (grey) to strong (red). Tissues are ordered by the magnitude of similarity from strong (top) to weak (bottom). **F** Correlation in expression of 19,273 protein-coding genes between urinary cells and the kidney. Points represent median gene expression ($\log_2(TPM + 1)$) in urine (*x*-axis) and a comparative tissue (*y*-axis). r - Spearman's correlation coefficient. The linear trend line is derived from linear regression between urinary cell expression and the comparative tissue. Trend lines are coloured by data source – HKTR is blue and Genotype-Tissue Expression project is red. **G** Expression profile for genes specific to kidney cortex and medulla. Standardised median $\log_2(TPM + 1)$ median expression values from urinary cells, cortical and medullary renal tissue datasets for curated renal single-cell marker genes from HPA. Genes are ordered by hierarchical clustering of the expression values. Expression is represented by colour from three standard deviations below mean expression (blue) through grey (mean expression) to three standard deviations above the mean expression (red). **H** Correlation between urinary cell and kidney median expression ($\log_2(TPM + 1)$) for 339 causal BP TWAS genes. The ten genes with smallest deviation from the linear regression line are labelled. Error bands represent 95% confidence intervals. r – Pearson correlation coefficient.

We also found that the magnitude of expression of genes known as markers of renal cortex (20 genes) and those from medulla (11 genes) was correlated with that in urinary cells (Fig. 7G). There was a particularly strong correlation between urinary cell expression of medullary markers (Fig. 7G). *UMOD* in urinary cells was comparable to that in the medulla and higher than in cortex, consistent with the production of *UMOD* within the ascending loop of Henle (Fig. 7G). For cortical markers, *NAT8*, *MIOX*, *RIDA*, *BHMT* all had relative urinary expression levels comparable with that of cortex samples.

Finally, we examined the urinary cell-kidney correlation in expression of 399 kidney genes showing a causal association with BP in our analyses. We noted that 339 of these genes had detectable expression in urinary cells. There was a strong positive correlation in expression of these genes between urinary cells and the kidney ($r = 0.73$, $P$-value $= 2.85 \times 10^{-57}$, Fig. 7H).

Collectively, we generated robust profiles of urinary cell transcriptome and showed its excellent correlation with the kidney transcriptome. We further demonstrate that several histological and functional annotations of gene expression profiles harvested from urine highlight specific kidney regions as the likely origin of these cells. Finally, we determine that profiling of urinary cells transcriptome offers a non-invasive insight into expression of genes of relevance to BP in the kidney. This highlights a potential diagnostic applicability of urinary cell transcriptomics, e.g., in development of non-invasive tests to determine kidney health and predict renal damage through analysis of spot urine samples.

### Genetically programmed reduction in kidney abundance of glutamyl aminopeptidase gene (*ENPEP*) and protein is associated with increased blood pressure and the risk of

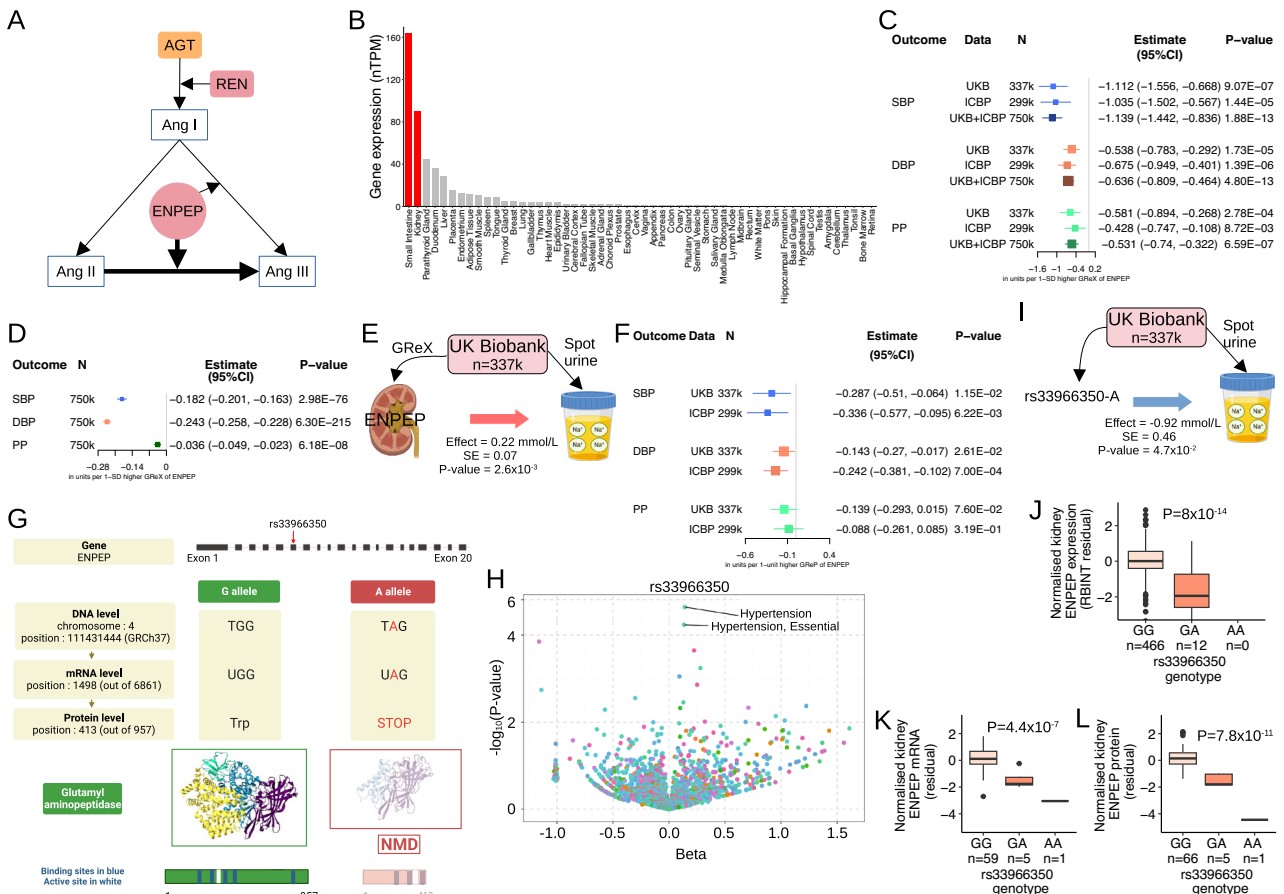

**Fig. 8 | Kidney glutamyl aminopeptidase gene (*ENPEP*) and blood pressure.**
**A** Simplified renin-angiotensin system. *AGT* – angiotensinogen gene. **B** Normalised *ENPEP* expression in GTEx. Red – Tissues enhanced, nTPM – consensus normalised expression. **C** Effects of genetically regulated expression (GReX) of *ENPEP* on systolic blood pressure (SBP), diastolic blood pressure (DBP) and pulse pressure (PP) from kidney TWAS. Nominal *P*-value is calculated from two-sided *Z*-score test. UKB – UK Biobank, ICBP – International Consortium for Blood Pressure, Estimate – change of SBP/DBP/PP in units per one-unit higher GReX of *ENPEP*, CI – confidence interval. **D**. Causal effects of GReX of kidney *ENPEP* on SBP/DBP/PP in UKB and ICBP. Estimate – change of outcome in units per one-unit higher GReX of *ENPEP*. Nominal *P*-value is calculated from one-sided chi-squared test. **E** Effect of GReX of *ENPEP* (per one standard deviation higher) on urinary sodium in UKB. Nominal *P*-value is calculated from linear regression (two-sided test). Red arrow – positive association. SE – standard error. Partially created with BioRender.com. **F** Effects of genetically regulated protein (GReP) of glutamyl aminopeptidase on SBP/DBP/PP from kidney PWAS. Nominal *P*-value is calculated from two-sided Z-score test. Estimate – change of outcome in units per one-unit higher GReP of glutamyl aminopeptidase. **G** rs33966350 of *ENPEP* [A – minor allele, G – major allele] leading to a premature stop codon at position 413 (out of 957) in exon 6. A shortened glutamyl aminopeptidase protein (red) compared to the normal variant (green). White – active site (catalysis residues), blue – binding site (protein-chemical interactions), Trp – Tryptophan, NMD – nonsense-mediated mRNA decay. Structural domains are coloured within each protein structure. Created with BioRender.com. **H** Effects of rs33966350-A on human diseases from FinnGen. Nominal *P*-value is calculated from Firth regression (two-sided test). Dots are coloured by categories of diseases. Beta – log of Odds ratio. **I** Effect of rs33966350-A on urinary sodium in UKB. Blue arrow – negative association. Partially created with BioRender.com. **J** Effect of rs33966350 on *ENPEP* expression in HKTR. Nominal *P*-value is calculated from linear regression (two-sided test). **K, L** Effects of rs33966350 on *ENPEP* expression and protein abundance in CPTAC. Nominal *P*-value is calculated from linear regression (two-sided test). In 8J-8L, whiskers denote extent of 1.5x interquartile range. Upper, middle and lower box lines denote 75th, 50th and 25th percentiles, respectively. In **C**, **D** and **F**, squares are positioned by the estimated effects with horizontal error bars illustrating 95% confidence intervals.

**hypertension – integration of evidence from genome-level, kidney transcriptome and proteome**

*ENPEP* encodes glutamyl aminopeptidase – an enzyme responsible for cleaving an aspartate from N-terminal of angiotensin II in the process of its conversion to angiotensin III (Fig. 8A). It is an attractive druggable target for development of new antihypertensive medications. However, the recent clinical trial (Firibastat in Treatment-resistant Hypertension - FRESH) has failed to demonstrate a BP lowering effect through pharmacological inhibition of this enzyme - an oral inhibitor of brain glutamyl aminopeptidase (Firibastat) did not lead to a significant reduction in BP in patients with difficult-to-treat and resistant hypertension when compared to the placebo.

*ENPEP* is one of kidney-enriched genes – it shows approximately 36-fold higher expression in the kidney than cerebral cortex (Supplementary Data 28 and Fig. 8B).

Our kidney BP TWAS study showed a consistently strong association between genetically determined increase in renal expression of *ENPEP* and a drop in all BP traits across two independent datasets as well as in their joint analysis bringing together approximately 750,000 individuals (Fig. 8C) and further MR demonstrated a causal effect (Fig. 8D). Genetically determined renal expression of *ENPEP* showed an association with increased urinary sodium in 337,350 individuals from UK Biobank (Fig. 8E); this may suggest renal Ang III-mediated effect on natriuresis as the relevant mechanism[103,104]. Through our BP kidney PWAS we also demonstrated an association between glutamyl aminopeptidase protein and BP in the consistent direction – reduced abundance of the protein was associated with higher values of DBP (Fig. 8F). Finally, we determined a signal of multi-trait colocalisation between DBP, *ENPEP* mRNA and protein expression in the kidney on chromosome 4 (Supplementary Data 29).

We then sought to provide an orthogonal replication of these findings using a rare *ENPEP* variant (rs33966350) associated with hypertension in previous GWAS[21]. We confirmed that rs33966350 has not been included in the models used to generate GReX for *ENPEP* in TWAS and PWAS and is not in strong LD with the genetic variants used in these models (Supplementary Data 30). The rare allelic variant (A) of rs33966350 leads to a premature stop codon, has a scaled Combined Annotation Dependent Depletion (CADD) score of 43 (consistent with top ~0.01 % of 8.6 billion variants) and is one of the high-confidence loss of function variants (Fig. 8G).

Our in silico analysis mapped this nonsense variant to exon 6 of *ENPEP* and confirmed that the truncated mRNA is a strong candidate for nonsense-mediated decay (NMD) and that the truncated protein (lacking two binding sites and a residue responsible for transition-state stability) is unlikely to be functional (Fig. 8G).

In the phenome-wide association of 2269 binary traits in 377,277 individuals from the FinnGen consortium (r9.finngen.fi) (Fig. 8H) we identified hypertension as the top association signal for rs33966350. We further uncovered that the carriers of AA genotype of this variant had approximately 33% increase in odds of hypertension compared to those with a wild-type genotype ($P$-value = $1.6 \times 10^{-6}$). Our studies in UK Biobank confirmed that rs33966350 was also associated with urinary sodium ($P$-value = $4.7 \times 10^{-2}$, Fig. 8I). We then showed that when compared to those with a wild-type genotype (GG), carriers of one copy of A-allele of rs33966350 have approximately 4.9-fold lower expression of *ENPEP* mRNA in the analysis of 478 kidneys ($P$-value = $8 \times 10^{-14}$, Fig. 8J). We replicated this observation in the analysis of 65 kidneys from CPTAC – carriers of the rare homozygous genotype had significantly lower levels of *ENPEP* than those with GG genotype ($P$-value = $4.4 \times 10^{-7}$, Fig. 8K). We further validated these observations at the protein level (Fig. 8L).

Collectively, our data show that genetically determined reduction of kidney abundance of *ENPEP* mimicking the effects of its rare loss-of-function genetic variant on gene and protein expression leads to increase in BP and the risk of hypertension possibly via effects on renal

excretion of sodium. These results provide a persuasive case for the development of pharmaceuticals that increase levels of *ENPEP* in the kidney (rather than inhibit its activity in the brain, e.g. as in FRESH trial).

## Discussion

Through analysis of effects of subtle changes in genetically determined portion of gene expression on phenotypes at a population scale, TWAS has emerged as a powerful approach to gene discovery[36,47,105–108]. The attractiveness of TWAS lies in its scaled-up detection power[109,110], immunity to confounding from pharmacological treatment, lifestyle, environment[111], and reverse causality[105] as well as a capacity to uncover disease/trait-associated genes within chromosomal regions "missed by GWAS"[47]. Indeed, application of TWAS-based approach in our project has increased the discovery of kidney genes showing robust associations with BP by 2.2-fold when compared to our previous *cis*-eQTL-driven analysis of BP GWAS[25] and mapped many of them outside the BP GWAS loci. This enhanced discovery power stems partly from enhanced genetic input into the TWAS prediction models (aggregation of numerous SNPs for each gene at the same time rather than single "top" e-variant) and a reduced burden for multiple testing (fewer genes in TWAS than SNPs in GWAS are tested[109,110]). We further augmented the rate of gene discovery through integrating input from kidney 3D genomic and epigenomic data in our GReX models[41] developed for the purpose of TWAS. With larger tissue reference panels, cross-ethnic analyses[112,113], utilisation of input from rare and in-*trans* variants[114], future BP TWAS studies should be able to generate even more precise estimates for expression of more human genes enhancing the discovery of new and fine mapping of the existing BP GWAS and TWAS loci.

The genotype-based imputations of molecular layers other than transcriptome are also gaining traction in post-GWAS searches of the effector genes for complex disorders[115–117]. Our study is the first analysis applying the genotype-based prediction models to impute the kidney microRNAome and proteome. This has helped us not only to uncover new BP-associated molecules amongst kidney miRNAs and proteins but also to substantiate the evidence for the robustness of our key targets emerging from BP kidney TWAS (e.g., through multi-omic overlaps or/and mediation analyses).

We also illustrate the benefit of triangulation of outputs from kidney omics with data from other molecular layers including plasma proteomics and metabolomics to uncover the consequences of BP-related changes in gene/protein expression in clinically accessible materials. Such analyses are a critical step in translating the findings from post-GWAS omics into diagnostically actionable targets with a potential to be tested e.g., as bio-markers of hypertension-mediated organ damage. We further provide a proof-of-principle for the informativeness of urinary cells to track the expression of the kidney transcriptome and the expression of genes associated with BP in GWAS and TWAS. Urinary mRNA signatures have emerged as non-invasive predictors of kidney allograft status[97,99]; our data corroborate the evidence behind potential diagnostic applicability of cell harvested from human urine even in the absence of a powerful clinical driver stimulating the urinary excretion of leucocytes[97,99].

Apart from uncovering new potential repositioning opportunities for hypertension, and illuminating BP-related effects of drugs used in other conditions than hypertension, our analyses provide specific cues into targets already tested in clinical trials (i.e. glutamyl aminopeptidase encoded by *ENPEP*) or those emerging as novel therapeutic strategies for hypertension [(e.g. suppression of *AGT* mRNA expression using antisense oligonucleotides[118]]. Numerous kidney genes causally associated with BP in our studies show a high level of therapeutic tractability. Indeed, *SLC5A11* and *AQP4* belong to druggable families whose other members (e.g. *SLC5A2* and *AQP2*) are already targeted (either directly or indirectly) by existing cardiovascular/

nephroprotective medications (i.e. SGLT2 inhibitors and vasopressin receptor 2 antagonists)[119–121].

Our study has several limitations. First, it should be noted that TWAS, MR and FOCUS are not fully orthogonal approaches to discovery of genes associated with a phenotype of interest. However, integration of these three approaches in one analytical pipeline increases the level of robustness and confidence in the detected signals (genes) that received support at each stage of this computational strategy. Second, at the TWAS stage we chose an FDR-based correction for multiple testing rather than a more conservative Bonferroni correction. This means we could have discovered more genes with perhaps less stringent statistical evidence. However, we have carefully reduced the likelihood of false positive signals through a layered system of gene replications and refinements created by applying additional downstream filters on genes delivered by TWAS, i.e., independent replication, MR and FOCUS. Third, we accept that our "recapitulation analyses" tested a limited number of carefully selected biochemically active transporters and receptors in the kidney. Further studies inclusive of other genes, molecules, pathways, and phenotypes of relevance to kidney physiology will help to strengthen the evidence for biological robustness of the predicted kidney gene expression models. Finally, due to a very limited availability of kidney tissues samples from individuals of non-white European origin[24] we were restricted in our studies to the European ancestry group. More international efforts are required to overcome this barrier to cross-ancestry analyses of kidney-relevant diseases in the post-GWAS era.

Taken together, these studies demonstrate the value of kidney omics to provide new biological understanding of the genetic regulation of BP and to generate insights of therapeutic relevance to hypertension.

## Methods

### Prioritisation of human tissues of relevance to blood pressure

**Transcriptome-wide association studies across 49 human cell-types and tissues.** We used genotype information with matching transcriptome of 49 human cell-types and tissues included in the Genotype-Tissue Expression (GTEx, v8) for the purpose of this analysis. Given that the currently employed metrics cannot fully account for the dependence of TWAS discovery on the sample size, we opted for a numerically equal representation of samples from each selected tissue – 65 samples of European ancestry was the most optimal cost-performance trade-off between the power of discovery and the number of tissues included in the analysis[122]. Using an equally weighted sampling algorithm[123], we selected a random 65 samples from each of 49 tissues using publicly available GTEx v8. Genotype information derived from whole genome DNA sequencing was downloaded from dbGaP. We conducted the quality control on the genotype data using updated PredictDB Pipeline (http://predictdb.org/). In total, 6,821,602 single nucleotide polymorphisms (SNPs) were available for further analyses. Gene expression information was derived from quality-controlled and normalised RNA-sequenced tissue datasets from the GTEx (http://www.gtexportal.org/). In line with the recommendations of GTEx Consortium[122], 15 probabilistic estimation of expression residuals (PEER) were generated using the PEER framework[124] for each tissue. Gene expression residuals for the purpose of transcriptomics model prediction were generated by adjusting the gene expression for sex, RNA-sequencing protocol (PCR-based or PCR-free), sequencing platform (Illumina HiSeq 2000 or HiSeq X), top five genotyping principal components and 15 PEER factors. As an input into TWAS analysis, we also used meta-analysis summary of the associations between 7,088,121/7,160,657 SNPs and SBP/DBP from 750,000 individuals[22] (UK Biobank and ICBP data). The position of SNPs under GRCh37 in the data was converted to GRCh38 by LiftOver[125]. SNPs not present in the 1000 genome reference panel under GRCh38 were excluded from the analysis.

Following the updated PredictDB Pipeline, we trained predictors of gene expression by applying PrediXcan[126] to genotype and RNA-sequencing datasets across 49 tissues in GTEx v8. For each tissue, we kept genes with nested cross validated correlation between predicted and actual levels > 0.10 (or equivalently $R^2 > 1\%$) and $P$-value of the correlation test <0.05. We identified gene-SBP/DBP associations by analysing summary statistics from SBP/DBP and gene expression predictors using S-PrediXcan[127]. SNPs in the broad major histocompatibility complex (MHC) region (chromosome 6: 28–34 Mb) were excluded. The correction for multiple testing was calculated using Bonferroni-adjusted $P$-value < 0.05 after adjustment for the total number of genes in each tissue tested. The gene-SBP/DBP associations that remained after this filtering were considered significant.

To quantify the overall relevance of a tissue to BP, we adopted three independent metrics of the tissue-disease association. Firstly, a proportion of independent genes identified from TWAS was quantified in each tissue. Gene-gene independence was defined as a correlation between genes $R^2 < 0.05$[128] within a 2 Mb region or genes located larger than 2 Mb from each other. Second, the mean TWAS association statistics (mean of squared Z-score) using all independent genes identified from TWAS[110] were used to quantify average strength of their association with SBP and DBP – the tissues were ranked based on the average effect of the magnitude of significant genes' association with BP. Third, we counted the number of significant genes associated with SBP/DBP outside the previously identified BP GWAS loci, i.e. 49 tissues were ranked according to the number of TWAS genes showing no overlap with any BP GWAS loci (GWAS locus was defined as ±1 Mb from the sentinel BP GWAS SNP). The independence of three metrics was tested by Pearson's product-moment correlation test. The overall rank was derived by taking the sum of three ranks for each tissue. The relevance of each tissue to BP were then visualised by integrating DBP overall ranking score against SBP overall ranking score for each tissue (Fig. 2A).

### Populations, DNA and RNA processing

**Human Kidney Tissue Resource – populations and samples.** The Human Kidney Tissue Resource (HKTR) is the collection of human kidney tissue samples secured for the purpose of multi-omics analyses. The following studies have contributed to the resource: the TRANScriptome of renaL humAn TissuE study (TRANSLATE)[25,31–34,84], TRANScriptome of renaL humAn TissuE - Transplant study (TRANSLATE-T)[25,32,34], moleculAr analysis of human kiDney-Manchester renal tIssue pRojEct (ADMIRE)[25,34], Renal gEne expreSsion and PredispOsition to cardiovascular and kidNey Disease (RESPOND)[25,34] and moleculaR analysis of mEchanisms regulating gene exPression in post-ischAemic Injury to Renal allograft (REPAIR)[25,34].

As reported before[25], the specimens were taken directly from the healthy (unaffected by cancer) pole of the kidney immediately after elective nephrectomy or by needle biopsy of donor kidneys before the transplantation. Of the 478 specimens used in this study 296 were from nephrectomies and 182 were from kidney biopsies. All individuals were of white-European ancestry. Further information on the individuals recruited into each study are given in Supplementary Data 31.

**Human Kidney Tissue Resource – DNA genotyping data generation, quality control and analysis, genetic principal components.** As reported previously[25,31–33,84], tissue samples were first homogenised and DNA extracted using the Qiagen DNeasy Blood and Tissue kit. DNA was then hybridised to the Illumina HumanCoreExome-24 beadchip array. Genotype calls for each sample were made using Illumina GenomeStudio. DNA quality control consisted of excluding any sample displaying cryptic relatedness, low genotyping rate (<95%), heterozygosity outside ±3 standard deviations from the mean, genetic/phenotypic sex mismatch. Variant-level quality control excluded all variants with genotyping rate (<95%), genomic location on a sex-

chromosome or mitochondrial genome, ambiguous genomic location, Hardy-Weinberg equilibrium (HWE) *P*-value <$1 \times 10^{-3}$ and minor allele frequency <5%. Genotypes were imputed by the Michigan Imputation Server (MIS)[129] using the 1000 Genomes phase 3 reference panel. Post-imputation quality control excluded all variants duplicated genomic location, imputation score <0.4, minor allele frequency <1% or HWE *P*-value < $1 \times 10^{-6}$. 8,735,852 variants remained after all genotyping quality control steps. Genotype principal components were derived from genotyped autosomal variants that passed all genotyping quality control filters using EIGENSTRAT[130] and SNPWeights[131].

**Human Kidney Tissue Resource – RNA-sequencing data generation, quality control and analysis.** As reported previously[25,31–33,84], kidney tissue was first homogenised and then either the Qiagen RNeasy kit or the Qiagen miRNeasy kit was used to complete the extraction of RNA. RNA integrity and purity was checked and 1 μg of normalised RNA was used in either a New England Biosciences or Illumina TruSeq poly-A selection sequencing library preparation protocol. Libraries were then sequenced paired-end with either a 75 bp, 100 bp or 150 bp read length. An average of 35 million paired reads and 6 Gb of sequencing data were generated per sample. RNA-sequencing quality control metrics were generated by FastQC and RNASeQC. Reads were pseudoaligned to the Ensembl v83 GRCh38 human transcriptome reference using Kallisto v0.44.0. Gene expression was quantified at the transcript-level in units of Transcripts Per Million (TPM) and was summarised to the gene-level by summing expression of all transcripts produced by each gene in the Ensembl transcriptome reference. The criteria for a gene to be expressed were if at least 20% of kidney samples in each population had TPM > 0.1 and read count > 5. After application of these criteria 21,414 kidney genes remained available for analysis.

**Inference of cell-type proportions from RNA-sequencing data.** To adjust for between-sample cell-type heterogeneity in gene expression data from HKTR we used a computational deconvolution approach combined with a single-cell renal gene expression dataset. Firstly, we extracted expression profiles for the 3448 normal kidney cells present in the data generated by Young et al[132]. These data were generated from FACS-sorted cellular suspensions which were derived from ≈30mm³ kidney tissue samples. Sequencing cDNA libraries were created by the 10X Genomics Chromium platform and sequenced on an Illumina HiSeq 4000. Gene expression, at the single cell level, was then normalised and kidney cell-types were identified using the Seurat R package[133]. We identified seven key distinct cell-type clusters amongst these cells and then used non-negative least squares multivariate regression on the single-cell expression data to determine cell-type specific gene weightings using the "MuSiC" R package[134]. We applied these gene weightings to all bulk tissue sample profiles and deconvolved estimated cell-type proportions for each sample.

**Human Kidney Tissue Resource ethical compliance.** The studies adhered to the Declaration of Helsinki and were approved/ratified by the Bioethics Committee of the Medical University of Silesia (Katowice, Poland), Bioethics Committee of Karol Marcinkowski Medical University (Poznan, Poland), Ethics Committee of University of Leicester (Leicester, UK), University of Manchester Research Ethics Committee (Manchester, UK) and National Research Ethics Service Committee Northwest (Manchester, UK). Informed written consents were obtained from all individuals recruited (for the deceased donors, the consent was obtained in line with the local governance; e.g. from the family members).

**National Institutes of Health kidney collections – populations and samples.** We used kidney samples from The Cancer Genome Atlas (TCGA)[135], GTEx[122] and Clinical Proteomic Tumor Analysis Consortium (CPTAC)[136].

TCGA contains human tissue samples collected after elective surgical procedures for a variety of cancers. Normal adjacent tissue samples (NATs) were taken from companion normal tissue adjacent to the tumour[135]. We identified 91 kidney NATs with matching genotype and RNA-sequencing data in this resource.

Tissue samples in the GTEx project were collected post-mortem and immediately stored for DNA/RNA extraction and processing[122]. We identified and used 65 kidney cortex samples with matching whole genome sequencing and RNA-sequencing data.

CPTAC collects and generates proteomic, transcriptomic and genomic data from a variety of solid tissue tumours, along with same data from NATs[136]. We used 66 NAT kidney tissue samples with whole genome sequencing and RNA-sequencing profiles for the purpose of TWAS.

Collectively, we used 222 kidney samples from NIH cohorts for the purpose of prediction performance validation of the GReX prediction models. The characteristics of individuals from these cohorts are given in Supplementary Data 32.

**National Institutes of Health kidney collections – DNA extraction, genotyping data generation, quality control and analysis.** In TCGA, DNA was extracted from blood samples using QiAamp Blood Midi Kit (CGARN, 2016) and hybridised with probes on the Affymetrix SNP 6.0 array (composed of 906,600 probes); genotype calls were made using the Birdseed algorithm (https://www.broadinstitute.org/birdsuite/birdsuite-analysis). The TCGA genotype data were downloaded from the Genomic Data Commons (GDC) Portal's legacy archive. A total of 525 cases/files were initially identified using the following query criteria: "project name"–"TCGA", "primary site"–"kidney", "sample type"–"solid tissue normal", "race"–"white", "data category"–"simple nucleotide variation", "data type"–"genotypes", "experimental strategy"–"genotyping array" and "access"–"controlled". We downloaded the data for 110 individuals who had matching RNA-sequencing-derived information on the transcriptome of normal kidney tissue. Genotype quality control, imputation, post-imputation quality control and genotype principal component analyses were performed identically to the HKTR genotyping data set. After all steps were complete, 8,541,201 variants remained in the TCGA genotyping data set.

Genotyping in the GTEx project was performed by whole genome sequencing on an Illumina HiSeqX (to 15x coverage) on DNA extracted from blood samples. The full experimental protocol is reported in the following NCI SOPs: BBRB-PR-0004, BBRB-PR-0004-W1, BBRB-PR-0004-W1-G3 and in the original publication[122]. Complete methodological detail is provided elsewhere[122], but in brief, raw reads were mapped to the human GRCh38 reference sequence using BWA-MEM (http://bio-bwa.sourceforge.net). Autosomal variant calling was performed by SHAPEIT. Genotype data was downloaded in indexed VCF format from dbGAP. The final number of variants available for combination with other genotype datasets was 45,138,608.

DNA extraction in CPTAC was performed on blood samples for each sample according to the following SOPs: https://brd.nci.nih.gov/brd/sop-compendium/show/41. Briefly, DNA was extracted using the QIAsymphony DNA Mini Kit (Qiagen), acoustically sheared, indexed, multiplexed and then sequenced to 15x coverage on an Illumina HiSeqX. FASTQ reads were mapped to the National Cancer Institute (NCI) Genomic Data Commons (GDC) GRCh38.d1.vd1 reference sequence (https://api.gdc.cancer.gov/data/254f697d-310d-4d7d-a27b-27fbf767a834) following the standard GDC protocol (https://docs.gdc.cancer.gov/Data/Bioinformatics_Pipelines/DNA_Seq_Variant_Calling_Pipeline/#alignment-workflow) which involves read mapping with BWA-MEM[137] alignment sorting and merging using Picard (https://broadinstitute.github.io/picard), duplicate marking (also using Picard) and finally base quality score recalibration using the Genome Analysis

ToolKit (GATK). BAM files containing reads aligned using this workflow were then downloaded from the GDC portal using the following query parameters: cases.samples.sample_type = 'solid tissue normal', cases.primary_site = "kidney", cases.project.program.name = "CPTAC", files.data_format = "bam" and files.experimental_strategy = "WGS". Variant calling was performed by the GATK version 3.8 "Haplotype-Caller" variant caller in single sample GVCF mode with the following command line arguments "-dontUseSoftClippedBases -ERC GVCF" using 8 CPU cores with 32GB of RAM. All individual GVCF files were then genotyped jointly using the GATK tool "GenotypeGVCFs" using default arguments using 32 CPU cores with 128GB or RAM. BCFTools was then used to split any multiallelic variant calls into biallelic calls to produce a final variant call set which could be integrated with that from all other cohorts. The final number of variants available for combination with other genotype datasets was 19,847,739.

**National Institutes of Health kidney collections – RNA processing, RNA-sequencing data generation, quality control and analysis.** In TCGA kidney RNA was extracted using the Qiagen ALLPrep kit. Sequencing libraries were generated from poly-A selected mRNA and sequenced on an Illumina HiSeq2000 producing an average of 80.6 million paired reads per sample. All TCGA sequencing reads for kidney NAT samples were downloaded from the genomic data commons portal (https://portal.gdc.cancer.gov).

Tissue RNA extraction in GTEx was performed as reported before[138]; sequencing libraries were generated using the Illumina Tru-Seq poly-A selection protocol. The libraries were sequenced on either an Illumina HiSeq2000 or HiSeq2500 producing an average of 82 million paired 76 bp reads per sample. GTEx sequencing reads for all kidney cortex samples were downloaded from the GTEx v8 google cloud computing bucket (data accessed May 2021).

Extraction of RNA from CPTAC samples was performed using the study's SOPs: https://brd.nci.nih.gov/brd/sop-compendium/show/41. The libraries were constructed using the Illumina TruSeq total RNA protocol with rRNA depletion (RiboZero gold) and sequenced on an Illumina HiSeq4000 generating a minimum of 120 million paired 75 bp reads per sample. CPTAC aligned reads were downloaded from the genomic data commons data portal (https://portal.gdc.cancer.gov/).

All validation panel samples were downloaded, raw FASTQ reads extracted and pre-processed using the protocol identical to the discovery resource. Raw FASTQ reads were extracted from aligned BAM files using bamtofastq from biobambam (https://github.com/gt1/biobambam).

**Regulatory compliance.** NIH has granted us access to TCGA, GTEx and CPTAC data under the approved dbGAP project 13040.

**Generation of genetically regulated expression models for the purpose of kidney blood pressure transcriptome-wide association studies**
**Input into gene expression prediction models – Human Kidney Tissue Resource (discovery resource).** We conducted the quality control on the genotype data using updated PredictDB Pipeline (http://predictdb.org/). SNPs not present in the 1000 genome reference panel under GRCh37 and not available from the meta-analysis summary of the SNP-BP associations[22] were excluded from the analysis. In total, 6,571,172 SNPs remained for the analysis.

Gene expression, in TPM units, was normalised by logarithmic transformation, quantile normalisation (using the R package aroma.light) and rank-based inverse normal transformation, as described before[25]. We then used PEER[124] to infer 100 hidden factors that describe global sources of variation in the normalised data. Residuals of gene expression were then calculated from the normalised data by adjusting for age, sex, tissue source (nephrectomy/biopsy), the first three

genetic principal components, the 100 PEER hidden factors and seven cell-type proportions using linear regression.

**Gene annotation data.** Gene biotypes, start and end coordinates, strand information and chromosomal localisations were collected using the "biomaRt" R package using the Ensembl gene ID for all expressed genes in our data set. We queried v83 of the "hsapiens_gene_ensembl" dataset using the GRCh38 human genome build, data accessed February 2021. The positions of genes were also converted to GRCh37 by LiftOver[125] for downstream analyses.

**Prediction Using Models Informed by Chromatin conformations and Epigenomics model for imputing gene expression in the kidney.** PUMICE[41] generates GReX models utilising epigenetic information (i.e. epigenetic annotation tracks) to prioritise essential genetic variants that carry important functional roles and 3D genomic information to define windows that harbour *cis*-regulatory variants.

Variants that overlap these annotation tracks are deemed "essential" genetic variants, and variants that do not overlap the annotation tracks are deemed "non-essential variants". We use $X_1$ to represent the set of essential genetic variants, $X_2$ to denote "non-essential" genetic variants, and $E$ to represent the vector of gene expression levels across multiple individuals. $\phi$ is a tuning parameter that controls the penalty on the "essential" predictors relative to the non-essential predictors[139]. We assume $\phi \leq 1$ so that "essential" predictors are penalised no more than the non-essential predictors.

The GReX model seeks to estimate the weights $\beta_1$ and $\beta_2$ by minimising the following objective function:

$$L(\beta_1, \beta_2; \lambda, \phi) = ||E - X_1\beta_1 - X_2\beta_2||_2^2 + \frac{1}{2} \times \frac{\lambda}{2}(\phi||\beta_1||_2^2 + ||\beta_2||_2^2) + \frac{1}{2}\lambda(\phi||\beta_1||_1^1 + ||\beta_2||_1^1)$$

(1)

PUMICE also utilises different choices for windows that harbour *cis*-regulatory variants as another tuning parameter (denoted as $w$), which includes the ones defined by conventional linear windows surrounding gene start and end sites (i.e., ±250 kb and ±1 Mb) as well as by 3D genomics informed regions (i.e., loop and TAD). In total, nested cross-validation is utilised to select the optimal combinations of tuning parameters $\lambda$, $\phi$, and $w$.

For generating PUMICE-kidney-specific GReX models, we retrieved epigenomic annotation data for adult human kidney tissue from ENCODE[140,141] and kidney-specific chromatin conformations[42]. Specifically, we downloaded epigenomic annotations (in BED format) from ChIP-seq experiments for the following epigenetic marks: H3K27ac (ENCFF077LXK), H3K4me3 (ENCFF423PKK), DNase hypersensitivity sites (ENCFF416ORJ), and CTCF (ENCSR000DMC). For BED files based on GRCh38 human genome build, we used LiftOver implemented in rtracklayer[142] to convert genomic positions to GRCh37 version. Fastq files of two technical replicates for H3K27ac HiChIP from the proximal tubule-derived HK2 cell line were obtained from European Nucleotide Archive (Run Accession SRR11434878 and SRR11434879)[42]. HiChIP paired-end reads for each replicate were combined and aligned to hg19 reference genomes using the Juicer pipeline[143] with default parameters to assign reads to MboI restriction fragments and generate binned interaction matrices with aligned reads MAPQ ≥ 30. TADs were called using HiCtool[144] at the required 40 kb resolution. A 40 kb resolution interaction matrix was obtained by converting 10 kb matrix generated by Juicer to 40 kb resolution using HiCExplorer[145]. Loops were called using Peakachu[146] at 10 kb resolution using pre-trained CTCF ChIA-PET and H3K27ac HiChIP model for 30 million intra-reads. The final set of loops were combined from both models with predicted probability > 0.59. The probability threshold

was chosen to obtain approximately 10,000 loops combining results from both models.

We then used 478 samples from HKTR to generate GReX models for kidney genes. We trained predictors of gene expression by applying the above PUMICE-based algorithm to genotype and RNA-sequencing-derived transcriptome from HKTR. We kept genes with nested cross-validated Pearson correlation between predicted and actual levels > 0.10 (or equivalently $r^2 > 0.01$) and *P*-value of the correlation test <0.05[126].

**Prediction performance validation of PUMICE-derived genetically regulated expression models for kidney genes using the validation panel of The Genotype-Tissue Expression, The Cancer Genome Atlas and Clinical Proteomic Tumor Analysis Consortium data.** For the purpose of the prediction performance validation, genotype data from GTEx, TCGA and CPTAC ($n = 222$) were combined. TCGA was integrated with GTEx and CPTAC by only retaining variants common to all data sets. SNPs with missing rate > 5% were excluded. After further overlapping with the genotype data from HKTR, the final genotype dataset contained 6,305,298 autosomal variants that were used for prediction performance validation of the GReX models.

Gene expression from all NIH datasets (GTEx, TCGA, CPTAC, totalling 222 samples), in units of TPM, was normalised by logarithmic transformation, quantile normalisation (using "aroma.light"), and rank-based inverse normal transformation, as described before[25]. We then used PEER[124] to infer 30 hidden factors that describe global sources of variation in the normalised data. Residuals of gene expression were then calculated from the normalised data by adjusting for age, sex, study (GTEx, TCGA, CPTAC), top three genetic principal components and the 30 hidden factors from PEER using linear regression.

We obtained predicted expression for all imputable genes by applying the PUMICE-derived GReX models to our fully independent validation dataset of 222 human kidney tissue samples (from GTEx, TCGA and CPTAC). Predicted expression for each imputable gene was compared to residuals of gene expression across all samples using Pearson's correlation coefficient. A correlation coefficient of > 0.1 was used as the criterion of a validated model. Predictive performance of PUMICE-derived validated models between the discovery and validation kidney resources were tested using Pearson's correlation coefficient.

**Analysis of recapitulation of biological function for generated models of predicted gene expression.** We examined the association between predicted kidney expression of genes of key relevance to four biochemical blood phenotypes (i) reflective of the kidney's involvement in excretion and/or reabsorption of solutes and (ii) measured in UK Biobank[147] [serum glucose (mmol/L), serum urate (μmol/L), serum phosphate (mmol/L) and serum calcium (mmol/L)].

For each phenotype, we looked up the GWAS SNPs (ordered by the magnitude of statistical significance of association with the respective trait) from the list of GWAS Catalog[148] associations and selected the top genes with validated GReX models to which the GWAS SNPs were mapped (*SLC2A2* for glucose, *SLC2A9* and *SLC2A12* for urate, *IP6K3* for phosphate, *CASR* for calcium).

We also manually curated the diseases arising from/related to abnormalities in each of the selected blood biochemistry traits (Supplementary Data 5) from those that (i) had at least 100 cases present in UK Biobank and (ii) were assigned full ICD10 code (Supplementary Data 5). In brief, type 2 diabetes and gout were selected as the diseases related to glucose and urate, respectively. For phosphate we examined "polyarthrosis" and "other arthrosis" traits (7 diseases in total) (Supplementary Data 5). A total of 17 diseases were examined as related to renal metabolism of calcium under two umbrella terms ("urolithiasis" and "obstructive and reflux uropathy") (Supplementary Data 5). The

following four disorders were selected as relevant to renal handling of sodium/potassium "hyperosmolality and hypernatraemia", "hypo-osmolality and hyponatraemia", "hyperkalaemia" and "hypokalaemia" (Supplementary Data 5). Further details of UK Biobank fields, field codes and ICD codes are provided in Supplementary Data 5.

In the absence of serum sodium/potassium levels in UK Biobank we selected SCNN1B as the well-established contributor to renal homeostasis of sodium/potassium[24,149].

Associations between a genetically predicted expression of the relevant kidney gene (generated using a PUMICE algorithm in the discovery resource and validated in the validation resource) and a continuous blood biochemistry phenotype were conducted using the relevant genotype data from 337,350 unrelated individuals of white-European ancestry in UK Biobank[147] – we used linear regression with an adjustment for age, sex, genotyping array and the top ten genetic principal components.

Associations between a genetically predicted expression of the relevant kidney gene and a binary disease/disorder status were determined by Firth logistic regression[150] (to minimise a potential bias arising from an imbalanced case/control ratio). The logistic regression models were adjusted for the same set of covariates as those included in the analysis of continuous traits. For the scenarios with more than one trait examined we applied a correction for multiple testing (calculated using the Benjamini–Hochberg FDR); the threshold of corrected statistical significance was established at FDR < 0.05.

### Kidney transcriptome-wide association studies of blood pressure

**A catalogue of variants associated with blood pressure in genome-wide association studies.** As an input into TWAS analysis, we used summary statistics of the associations between (i) 19,267,390 SNPs and SBP/DBP/PP from 337,422 UK Biobank individuals, and (ii) 7,371,711/7,476,460/7,359,508 SNPs and SBP/DBP/PP from 299,024 ICBP individuals[22] (Supplementary Data 33). SNPs (i) not present in the 1000 genome reference panel under GRCh37 and (ii) not available in the BP GWAS analyses (UK Biobank and ICBP) were excluded from the analysis. A total of 6,305,298 variants common for HKTR, TCGA, GTEx and CPTAC were retained for the TWAS analysis.

**Existing blood pressure genome-wide association study loci and genomic loci outside blood pressure genome-wide association study loci.** A total of 885 GWAS variants were associated with at least one of the BP traits (SBP, DBP, PP) in previous BP GWASs[22]. A GWAS locus was defined by a GWAS variant and the 1 Mb region adjacent to the variant (on both sides). For the purpose of this study, overlapping loci were merged into single locus. In such cases, the most significant GWAS variant in each locus was defined as the sentinel GWAS variant for this locus. As a result, 429 independent BP GWAS loci were identified (Supplementary Data 34). Genomic regions not overlapping with these BP GWAS loci were mapped to pre-defined disjoint LD blocks[151].

**Blood pressure kidney transcriptome-wide association studies – reciprocal replication in UK Biobank and International Consortium for Blood Pressure.** We examined associations between kidney PUMICE-derived predicted gene expression and SBP, DBP and PP using S-PrediXcan[127]. We first examined associations between all 6490 kidney genes with validated expression models developed using kidney PUMICE with each of three BP-defining traits in each of the two resources, separately. The correction for multiple testing was based on the Benjamini–Hochberg FDR. Kidney genes whose predicted expression retained its significant association with BP in one of the resources were then taken for reciprocal replication (for analysis with the same BP trait) in the other resource. The correction for multiple testing was calculated using the Benjamini–Hochberg FDR (at the individual resource level) separately for each BP-defining trait. The

threshold of corrected statistical significance in both stages above was established at FDR < 0.05. Only genes showing directionally consistent associations with the same BP-defining trait in both cohorts were considered statistically significant (Fig. S3).

**Computational drug repurposing with Connectivity Map.** Using S-PrediXcan, we also generated summary statistics for gene-SBP/DBP/PP associations using meta-analysis GWAS summary statistics on SBP/DBP/PP from UK Biobank and ICBP[22]. For each BP-defining trait, we extracted its significant TWAS associations (FDR q-value < 0.05) from the pool of 6490 kidney genes and used this as a proxy for a trait signature. We then applied CMap algorithm[52] to identify drugs capable of inducing and reversing the disease signature. Specifically, we queried BP TWAS association signals against the reference profiles in the CMap database (from L1000 assay)[53], which recorded gene expression changes caused by perturbagens as the signature of the drug x gene pair. Only reference data from touchstone dataset were used, which comprised of reference signatures across nine cell lines treated with ~3000 small molecule drugs. To quantify the correlation between a query signature and reference profile, CMap calculated a $\tau$-score. A negative $\tau$-score indicates that the identified molecule can normalise trait signature and are capable of drug reposition, and vice versa. Only drug classes that survived a correction for multiple testing (calculated by the Benjamini–Hochberg FDR < 0.05) were considered as statistically significant.

### Causal inference and fine mapping analyses

**Mendelian randomisation with PMR-Egger.** We adopted a recent two-sample Mendelian randomisation method, PMR-Egger[62], developed specifically to control for horizontal pleiotropy and the presence of multiple correlated instruments through a maximum likelihood inference framework[62]. As an input to each MR analysis, we used: (i) a summary of the associations between the instrumental SNPs and SBP/DBP/PP, calculated from a meta-analysis of ~750,000 individuals[22] from UK Biobank and ICBP and (ii) a summary of the associations between the same SNPs and the selected BP TWAS genes generated in a kidney cis-eQTL analysis conducted using 478 samples with informative genotype and transcriptome information from HKTR. In brief, kidney cis-eQTL analysis brought together 8,735,852 genetic variants and 21,414 kidney genes. The normalised expression of each kidney gene was regressed against alternative allele dosage, age, sex, source of tissue indicator (nephrectomy/kidney biopsy), the top three principal components derived from genotyped autosomal variants, 100 hidden factors estimated using PEER and seven kidney cell-types. Only variants within 1Mb-regions from the transcription start site of a gene were considered and the analysis was carried out using FastQTL[152]. For each MR analysis, all cis-SNPs were LD-clumped with $r^2 < 0.5$ and selected as instruments. The correction for multiple testing was calculated using the Benjamini–Hochberg FDR, and the threshold of corrected statistical significance was established at FDR < 0.05. Genes with statistically significant causal effects and no evidence of pleiotropic effects estimated through PMR-Egger (i.e., FDR > 0.05) were selected for further analyses.

**Fine-mapping of transcriptome-wide association study associations.** We first mapped kidney genes prioritised through BP TWAS and PMR-Egger to pre-defined disjoint LD blocks[151]. Genes located within the MHC region were excluded. We then applied probabilistic fine-mapping to each of the independent genomic regions with more than one TWAS signal. Using Fine-mapping Of CaUsal gene Sets[63] (FOCUS) we prioritised kidney genes within each locus while controlling for pleiotropic effects. Within each locus we computed a 95% credible set of causal genes and the posterior inclusion probability (PIP) for each gene[63] (Fig. S4). We retained all genes present in the credible set and with a PIP > 0.5 for further analysis. As an input into FOCUS analysis, we

used (i) a weight database containing 6490 validated kidney gene expression prediction models generated by kidney PUMICE from HKTR, (ii) a summary of the associations between SNPs and SBP/DBP/PP from the meta-analysis GWAS[22] and (iii) 1000 genome reference panel.

**Mapping TWAS signals to independent genomic regions.** We mapped TWAS signals to existing BP GWAS loci and genomic loci outside BP GWAS loci (BP TWAS loci) and examined whether a genomic region contained a single or multiple TWAS signals.

### Integration of outputs from blood pressure kidney transcritome-wide association studies with plasma metabolomics and proteomics

**Association between kidney expression of solute carrier family 5 (sodium/inositol cotransporter), member 11 (*SLC5A11*) and serum glucose and HbA1c.** Using the validated kidney PUMICE-derived prediction model for *SLC5A11*, we generated its GReX in UK Biobank. We then examined its association with serum glucose (Data-Field 30740) and HbA1c (Data-Field 30750) values from 294,159 and 321,435 unrelated UK Biobank individuals of white-European ethnicity (respectively). In brief, included in these analyses were individuals who passed the sample level quality control filters[147]. Prior to the analysis, circulating glucose levels and HbA1c were normalised using rank-based inverse normal transformation. We then examined the associations between predicted kidney expression of *SLC5A11* and both glucose and HbA1c by regressing their normalised values on the *SLC5A11* GReX adjusting for age, sex, genotyping array and the top ten genetic principal components (PCs) in respective linear regression models.

***SLC5A11* kidney expression, plasma myo-inositol and pulse pressure – mediation analysis using two-step Mendelian randomisation.** We conducted mediation analysis using two-step MR[153,154] to investigate the extent to which the effect of predicted kidney expression of *SLC5A11* on PP may be mediated by plasma levels of myo-inositol. In brief, each step of the two-step MR is an independent univariable two-sample MR analysis. Penalised inverse-variance weighted (IVW) method[155] was used to estimate the potentially causal effect of the exposure (e.g. kidney mRNA expression of *SLC5A11*) on the mediator (e.g. plasma myo-inositol) and the causal effect of the mediator on the outcome (e.g. PP) in the first and second step (respectively), separately. MR-Egger regression was used to detect horizontal pleiotropy in each MR analysis[155]. A modified version of the IVW and MR-Egger estimators[156] that account for genetic correlations was used for any MR analysis with selected correlated instruments. The total effect of the exposure on the outcome was estimated using penalised IVW. The indirect effect (i.e. mediation effect) was calculated by multiplying the estimated effects from both steps. Mediation proportion (MP) was then calculated by dividing the indirect effect by the total effect. Standard errors of the estimated indirect effect and MP were derived using the delta method[157,158]. In each MR analysis, significant causal effect was identified if (i) the effect estimate was statistically significant ($P$-value < 0.05) and (ii) there was no horizontal pleiotropy ($P$-value > 0.05).

As an input, we used (i) summary statistics for SNPs associated with kidney expression of *SLC5A11* in cis-eQTL analysis conducted in 478 individuals from HKTR, (ii) GWAS summary statistics for plasma myo-inositol levels from INTERVAL and EPIC-Norfolk ($n = 14,296$)[75] and (iii) GWAS summary statistics for PP from UK Biobank and ICBP[22] ($n = 750,000$). Moderately correlated SNPs associated with *SLC5A11* expression ($P$-value < $1 \times 10^{-3}$ and LD $r^2 < 0.5$) were selected as instruments for estimating effects from SLC5A11 expression to myo-inositol and PP, respectively. Independent SNPs associated with myo-inositol ($P$-value < $5 \times 10^{-8}$ and LD $r^2 < 0.1$) were selected as instruments to estimate an effect from myo-inositol to PP.

**Expression of *AGMAT* – tissue and cell-type enrichment in human tissues.** Tissue enrichment was examined in a panel of human tissues from the Human Protein Atlas (HPA) based on RNA tissue specificity classification[159]. The kidney cell-type enrichment of *AGMAT* was based on the HPA cell-type specificity definition (https://www.proteinatlas.org/ENSG00000116771-AGMAT/single+cell+type).

**Analysis of association between kidney expression of *AGMAT* and blood urea nitrogen.** We first generated GReX based on the validated kidney PUMICE-derived prediction model for *AGMAT* using UK Biobank imputed genotype data[147]. BUN values (mmol/L) were derived from blood urea values (mmol/L) (Data-Field 30530) measured in 321,409 unrelated UK Biobank individuals of white-European ethnicity surviving sample level quality control[147] by multiplying the blood urea values by 0.1554[160]. The BUN values were further normalised using rank-based inverse normal transformation. We then estimated the effect of *AGMAT* GReX on BUN by regressing normalised values of the latter on *AGMAT* GReX adjusting for age, sex, genotyping array and the top ten genetic principal components (PCs) in linear regression.

We next estimated the effect of *AGMAT* expression on BUN in an independent cohort from CKDGen Cosnortium using two-sample MR. IVW was used to estimate the effect of *AGMAT* expression on BUN and MR-Egger regression was used to examine the presence of horizontal pleiotropy. For the purpose of the analysis, we used: (i) a summary of the associations between the instrumental SNPs and BUN, calculated from a meta-analysis of ~243,000 individuals[160] from CKDgen (ii) summary statistics of the associations between the same SNPs and kidney expression of *AGMAT* generated in the kidney *cis*-eQTL analysis in 478 samples from HKTR. The correction for multiple testing was conducted by Benjamini–Hochberg FDR on *cis*-eQTL associations on *AGMAT* expression. Independent SNPs associated with kidney expression of *AGMAT* (FDR < 0.05 and LD $r^2$ < 0.1) were selected as instruments for the MR analysis. Significant causal effect was identified if the effect estimate is significant (*P*-value < 0.05) with no horizontal pleiotropy (*P*-value > 0.05).

***AGMAT* kidney mRNA expression, plasma concentrations of 4-guanidinobutanoate and systolic blood pressure – mediation analysis using two-step Mendelian randomisation.** We conducted a mediation analysis under the above MR framework to estimate whether the effect of kidney expression of *AGMAT* on PP is mediated by plasma 4-guanidinobutanoate. For the purpose of this analysis, we used (i) summary statistics for *AGMAT* expression in the kidney (from *cis*-eQTL analysis carried out in 478 samples from HKTR), (ii) summary statistics for plasma 4-guanidinobutanoate from GWAS in INTERVAL and EPIC-Norfolk (*n* = 14,296)[75] and (iii) GWAS summary statistics for SBP from UK Biobank and ICBP[22] (*n* = 750,000). We selected independent SNPs associated with kidney expression of *AGMAT* (*P*-value < 1 × $10^{-2}$ and LD $r^2$ < 0.1) as instruments to estimate effects of *AGMAT* expression on SBP. Independent SNPs associated with kidney expression of *AGMAT* (*P*-value < 1 × $10^{-3}$ and LD $r^2$ < 0.1) were selected as instruments for the purpose of estimating effects from *AGMAT* expression to 4-guanidinobutanoate. Independent SNPs associated with myo-inositol (*P*-value < 5 × $10^{-8}$ and LD $r^2$ < 0.1) were selected as instruments in estimation of the effect of 4-guanidinobutanoate on SBP.

***AGT* kidney mRNA expression, angiotensinogen plasma protein levels and systolic blood pressure – mediation analysis using two-step Mendelian randomisation.** To determine whether plasma angiotensinogen protein levels mediate the effect of *AGT* kidney mRNA expression on SBP, we performed a mediation analysis under the MR framework described above. As an input, we used (i) summary statistics from *cis*-eQTL analysis of kidney *AGT* mRNA expression conducted in 478 samples from HKTR, (ii) GWAS summary statistics for angiotensinogen plasma protein levels from the Fenland study

(*n* = 10,708)[82] and (iii) GWAS summary statistics for PP from UK Biobank and ICBP[22] (*n* = 750,000). MR analyses of the effects of kidney *AGT* expression on circulating angiotensinogen protein levels and SBP (respectively) were based on a selected set of independent ($r^2$ < 0.1) instrumental SNPs showing a significant (*P*-value < 1 × $10^{-3}$) association with *AGT* expression in the *cis*-eQTL analysis. In estimating effect of circulating concentrations of angiotensinogen protein on SBP we used independent SNPs associated with plasma angiotensinogen (*P*-value < 5 × $10^{-8}$ and LD $r^2$ < 0.1) as instruments.

**Characterisation of kidney microRNAome and kidney blood pressure microRNAome-wide association studies (micro-RNA-TWAS)**

**Processing of miRNA-sequencing data and quality control.** A total of 379 samples from HKTR studies underwent small RNA-sequencing analysis to identify and quantify kidney miRNAs; all of these samples had matching genotype information. The libraries were generated using Illumina TruSeq and were sequenced using either 75 bp single-end reads from an Illumina NextSeq or 50 bp single-end reads from an Illumina HiSeq2500. In addition, 114 TCGA samples had kidney miRNA profiles available; all were sequenced using 30 bp single-end reads on an Illumina HiSeq2500[161]. Finally, 75 kidney miRNA profiles were secured from CPTAC resource[162]. These samples were processed using an Illumina HiSeq4000, as reported before[162]. The TCGA and CPTAC miRNA-sequencing reads were obtained from the GDC data portal and filtered using the following filters: (i) Program: TCGA or CPTAC, respectively, (ii) Primary site: kidney, (iii) Race: white, (iv) Sample type: solid tissue normal, (v) Experimental strategy: miRNA-sequencing and (vi) Data format: bam. In total, 114 TCGA and 75 CPTAC miRNA-sequencing datasets with corresponding genotyping data were downloaded.

In all studies, adaptors were trimmed with Trimmomatic using standard Illumina adapter sequences (sliding window size: 4 bases; minimum window length: 15 bases)[163]. Next, Spliced Transcripts Alignment to a Reference (STAR) software was used to align reads and to quantify miRNA mature products into reads per million (RPM)[164]. miRNA mature product annotations were taken from miRbase (release 22) using the GRCh38 reference genome[165]. ComBat-seq was later used for batch effect correction[166].

Samples with total read number <2 million and a D-statistic > 5[167] were excluded from further analysis. A total of 489 samples with matching genotype data survived quality control (339 – HKTR, 85 – TCGA and 65 – CPTAC, Supplementary Data 35) and were included into the downstream analyses.

A miRNA was retained for downstream analysis if its expression in RPM was >0.1 and read count ≥5 in at least 10% of the kidney samples in each population. miRNAs that did not fulfil these criteria, had an interquartile range of 0 or were located on sex chromosomes were excluded from further analysis. There were 1459 miRNAs that passed the quality control filters in HKTR, 967 in TCGA and 1459 in CPTAC. A total of 967 miRNAs common to all datasets were retained for further analyses.

Expression data were normalised by logarithmic transformation, quantile normalisation (using the R package aroma.light[168]), and rank-based inverse normal transformation, as described before[25]. miRNA expression residuals from linear regression models were then further adjusted for (i) age, (ii) sex, (iii) source of tissue sample indicator (nephrectomy/pre-transplantation biopsy), (iv) the first three auto-somal genetic principal components (calculated using the EIGENSTRAT[130] and SNPWeights[131] packages), and (v) 30 PEER factors[124].

**Mapping miRNA onto host genes and their classification.** miRNA annotations for the human (Homo sapiens) genome were obtained from miRbase (GRCh38; release 22)[165] and genomic coordinates were

used to map each miRNA onto gene positions curated from the Ensembl genome database (GRCh38; release 108)[169]. miRNAs located within the start and end coordinates of genes were defined as 'intragenic', and further divided into 'intronic' and 'exonic' based on their position within the host gene. Other miRNAs (not present within the body of any gene) were defined as 'intergenic'.

**Complexity of kidney microRNAome.** To determine relative transcriptomic complexity of miRNAs, we first divided all 21,414 kidney-expressed genes into simplified categories through manual mapping of Ensembl v83 "gene_biotype" values (Supplementary Data 36) and excluded 72 miRNAs and 23 other small non-coding RNAs identified using polyA RNA sequencing. In total, there were 15,205 protein-coding genes, 5063 long non-coding RNAs and 1051 pseudogenes. For each biotype, we calculated the proportion of the transcriptome/microRNAome attributable to each kidney-expressed gene and ranked them in descending order. We quantified the cumulative proportion of transcriptomes for each biotype referable to each quartile (25, 50 and 75%) of expression.

**miRNA tissue enrichment.** Human (Homo sapiens) small non-coding RNA expression information was curated from miRNATissueAtlas2[86] – we identified a total of 2656 mature miRNAs across 35 organs. Each miRNA was then assessed for enrichment using the HPA definitions[159]: miRNAs were categorised as 'enriched' if the expression in an organ was at least four times that of any other organ, 'group enriched' if the expression in a group of 2-5 organs was at least four times that of any other organ, 'enhanced' if the expression in an organ was at least four times the cross-tissue mean, 'low specificity' if the expression in at least one organ had $\log_2$ RPM >1 but did not belong to any previous category, and 'not detected' if the expression in all organs had $\log_2$ RPM < 1.

**Kidney blood pressure microRNAome-wide association studies.** We used elastic net regression under Predixcan framework[126] to develop prediction models for kidney miRNAs expression using (i) genotype data and (ii) the adjusted residuals of renal miRNA expression data derived from normalised small RNA-sequencing profiles, as reported above. Consistent with the computational pipeline developed for the purpose of BP kidney TWAS, all HKTR samples with informative kidney miRNA profiles and matching genotype information were used as a discovery resource ($n = 339$) while samples from NIH resources (TCGA and CPTAC, $n = 150$) were used for the purpose of validation. The prediction models of expression for 1459 kidney miRNAs in the discovery resource were produced using nested cross validated elastic net regression.

The prediction models with nested cross validated Pearson's correlation coefficient > 0.1 and *P*-value < 0.05 in the discovery resource were retained for further analysis. They were further examined in the validation resource; the residuals of miRNA expression in the discovery cohort were correlated with their respective GReX in the validation resource. The prediction models with Pearson's correlation coefficient >0.1 in the validation resource were retained for further downstream analysis.

Overall, there were 201 prediction models for kidney miRNA expression in the discovery resource. Of these, 80 were validated and retained for kidney microRNA-TWAS of BP. As an input into this analysis, we used the GWAS summary statistics for SBP, DBP and PP from UK Biobank ($n = 337,422$) and ICBP ($n = 299,024$), as reported above. We first examined associations between all 80 kidney miRNAs with fully validated GReX models and each of three BP-defining traits in each of the two resources, separately. The correction for multiple testing was based on the Benjamini–Hochberg FDR. Kidney miRNAs whose predicted expression retained its significant association with the same BP trait in one of the resources were then taken for a reciprocal replication in the other resource. The correction for multiple

testing was conducted using Benjamini–Hochberg FDR separately for each BP-defining trait. Only miRNAs showing directionally consistent associations with the same BP-defining trait in both resources were considered statistically significant (Fig. S7).

**Kidney miRNAs associated with blood pressure – biological annotations.** To provide functional context to the detected associations between kidney miRNAs and BP we have examined their (i) genetic origin, (ii) magnitude of their kidney expression, (iii) tissue specificity and (iv) genetic imputability. To determine the relative expression of BP-associated miRNAs in the kidney, we quantified mean expression values for 1459 renal miRNAs using 339 samples in the discovery resource and scaled the values between 0 and 1. To characterise kidney specificity, we retrieved miRNA expression data for multiple tissues from miRNATissueAtlas2[52]. We derived a mean value of expression for each miRNA in each organ and inferred specificity by calculating the fold-change of mean expression in the kidney against the cross-tissue mean. To quantify the imputability of miRNAs, we assessed the performance of their prediction models using nested cross-validated correlation between the predicted and measured levels of expression in the kidney.

**Blood pressure-associated kidney miRNAs and the neighbouring genes – conditional analyses.** We sought to examine the extent to which the identified signal of associations between BP and 11 kidney miRNAs may be mediated by (i) the host genes (i.e. genes whose sequences overlap with miRNAs), (ii) all BP kidney TWAS genes sharing the same chromosomal region with the miRNA. For each BP TWAS miRNA, we first identified all validated imputable protein-coding genes whose TSS map within 500 kb of the TSS of the BP TWAS miRNA. For further analyses we selected either the host gene (if acting as BP kidney TWAS gene) or each of the BP kidney TWAS genes within the 500 kb boundaries.

As an input into the conditional analysis, we used (i) a *Z*-score for the BP TWAS miRNA (obtained from the BP microRNA-TWAS showing the most significant association), denoted $Z_{miRNA}$, (ii) a *Z*-score for the corresponding selected gene (obtained from the BP TWAS), denoted $Z_{gene}$, and (iii) *cis*-regulated genetic correlation (r) of the BP TWAS miRNA and the corresponding selected gene.

We conditioned BP TWAS gene associations on the corresponding miRNA to obtain conditional *Z*-scores, i.e. $Z_{gene|miRNA}$ using the following formula[170]:

$$Z_{gene|miRNA} = \frac{Z_{gene} - r \times Z_{miRNA}}{\sqrt{1 - r^2}} \quad (2)$$

The *cis*-regulated genetic correlation was calculated from the Pearson's coefficients of correlation between imputed miRNA expression and imputed gene expressions for individuals from 1000 Genome by applying weights of their imputation models to the genotype data. A significant mediation effect was identified if (i), $|r| > 0.1$, (ii) $|Z_{gene}| > |Z_{gene|miRNA}| > 2$, and (iii), the direction of $Z_{gene|miRNA}$ was consistent with $Z_{gene}$[171].

## Characterisation of kidney proteome and blood pressure kidney proteome-wide association studies
**CPTAC human kidney tissue proteomics – data generation, quality control and quantification.** The abundance of human kidney tissue proteins were obtained from the CPTAC pre-processed protein-level assembly[136] and downloaded from the proteomic data commons (data accessed May 2022, https://pdc.cancer.gov/pdc/study/PDC000127). The details of the biochemical analyses are reported in full in the original publication[136]. In brief, we made use of the NAT samples (taken from regions of the kidney adjacent to renal clear cell tumours) and reviewed by a board-certified pathologist (to confirm the histological

status)[136]. Each kidney tissue sample was homogenised, lysed, digested and trypsinised. Samples were then multiplexed using tandem mass tag (TMT) and fractionated by basic reversed-phase liquid chromatography (bRPLC)[136]. Peptides were then separated by ultra-high-performance liquid chromatography (UHPLC) and analysed using the Thermo Fusion Lumos mass spectrometer[136]. The protein-level assembly of spectra and peptides into estimates of protein abundance was performed by a software workflow using the Philosopher pipeline[172] including spectral search by MSFragger[173] and refinement by PeptideProphet[174]. Peptide spectral match data for each sample were then normalised by log$_2$-transformed reference intensities determined for each TMT channel. A total of 83 NAT samples had raw protein-level abundance data; 72 of these had genotype information from whole genome sequencing available and 65 of these also had poly-A RNA-sequencing data. All samples used in this analysis were collected from patients of European ancestry. A total of 7291 proteins were quantifiable in all 72 samples and 7036 of these from 65 samples were available with measurable expression of their source gene in the RNA-sequencing data.

**General characteristics of kidney proteins.** We have first assigned each of the quantified kidney proteins into three predicted localisation categories (intracellular, membrane and secreted) provided by the HPA https://www.proteinatlas.org/about/download (data accessed May 2022). We then examined the extent to which proteins from the CPTAC dataset are enriched for varying degrees of kidney specific expression. Kidney enriched, group enriched and kidney enhanced genes were downloaded from the HPA (data accessed May 2022). We then used Fisher's exact test to calculate the enrichment odds ratio (with corresponding *P*-values) illustrative of the extent to which each of the kidney-specificity categories are over-represented amongst the proteins quantified in the CPTAC dataset compared to the whole HPA.

**Correlation between kidney gene expression and protein abundance.** Using 65 CPTAC samples with overlapping transcriptome/proteome information we examined Pearson's correlation between abundances of each of 7036 proteins (normalised intensity) and log$_2$-transformed estimated read counts of their parent genes (from the matching kidney RNA-sequencing data). These were then catalogued across the entire distribution of the observed correlations – from the most negative (r = -0.54, *P*-value = $9.9 \times 10^{-7}$) to the most positive (r = 0.92, *P*-value = $1.6 \times 10^{-29}$) and further examined using a density plot. An FDR threshold of 5% was used to adjust for multiple testing and determine the statistical significance of the identified correlations.

We examined whether genes of relevance to BP/hypertension are enriched for positive kidney gene-protein correlations within the CPTAC catalogue of 7036 gene-protein pairs. We selected four categories of relevance to BP/hypertension: (i) 216 genes with enriched or enhanced expression in kidney from HPA (ii) eight monogenic hypertension/hypotension genes collected from the manual review of Samani and Tomaszewski et al[91], (iii) seven antihypertensive drug targets manually collected from drugbank (https://go.drugbank.com/) and the OpenTargets platform (https://platform.opentargets.org/), (iv) 152 protein-coding BP kidney TWAS genes selected from the list of 399 BP TWAS genes showing causal association to BP in MR analyses. The enrichment *P*-value was calculated by a two-sample Kolmogorov–Smirnov test.

**Kidney proteome-wide association studies of blood pressure in Clinical Proteomic Tumor Analysis Consortium.** Using 72 kidney proteome profiles with matching genotype information from CPTAC as a reference we developed prediction models for protein abundance in the kidney tissue (Fig. S8). All individuals included in this analysis were of white-European ancestry.

Of 7291 proteins whose abundance in the kidney tissue were quantified by mass spectrometer based tissue proteomics, included

were those with matching expression data for their parent gene in the RNA-sequencing dataset from HKTR. A total of 6712 such kidney proteins were identified for the downstream analyses (Fig. 6E).

Prior to the analysis, we used linear regression to adjust kidney abundance of each included protein for age, sex, the top three principal components derived from genotyped autosomal variants and the top ten principal components derived from normalised kidney abundance. The generated residuals of 6712 proteins were then included as an input in the protein abundance model prediction together with genotype information on 5,785,933 variants derived from the whole genome sequencing data (Fig. 6E). Variant positions were lifted to GRCh37. We retained all variants present in the 1000 Genome reference panel (GRCh37) and that were also available in the BP GWAS analyses (UK Biobank and ICBP).

We trained prediction models of kidney protein abundance using the genotype and protein abundance data from CPTAC using the PrediXcan framework[126] (Fig. S8). We kept predictive models with nested cross validated correlation between predicted and actual levels > 0.1 and *P*-value of the Pearson's correlation test <0.05 (Fig. S8). Those that satisfied these criteria were included in kidney PWAS of BP. For the purpose of this analysis, we used the GWAS summary statistics on SBP, DBP and PP from UK Biobank and ICBP reported above. In brief, we included 337,422 individuals from UK Biobank and 299,024 from ICBP in this analysis (Figs. 6E and S8). We first examined association between all 815 cross-validated kidney protein abundance models with each of three BP defining traits in each of the two resources, separately. The correction for multiple testing was applied based on the Benjamini–Hochberg FDR. Kidney proteins whose predicted abundance retained its significant association with BP in one of the resources were then taken for a reciprocal replication in the other resource (Fig. S8). The correction for multiple testing was applied again at this stage and was based on the Benjamini–Hochberg FDR. The threshold of corrected statistical significance in both stages was established at FDR < 0.05.

**Annotating blood pressure-associated kidney proteins with additional levels of evidence for relevance to blood pressure.** We sought to identify additional evidence for each BP-associated kidney protein at other molecular levels, i.e.: (i) association with BP in BP kidney TWAS and (ii) prior evidence of colocalisation between BP and any other molecular phenotypes (i.e. mRNA expression, alternative splicing or DNA methylation) from previous kidney QTL studies[25]. We further partitioned BP-associated kidney proteins with no association with BP in kidney TWAS into two categories (i) those that yielded non-significant associations with BP in TWAS and (ii) those whose GReX model did not fully converge prior to TWAS.

**Kidney proteins associated with blood pressure in proteome-wide association studies –overrepresentation pathway analysis.** Overrepresentation analysis was performed by the "fora" function in the "fgsea" R package. All KEGG pathways from MSigDB subcategory "C2:KEGG" and all Human Phenotype Ontology (HPO) terms from MSigDB subcategory "HPO" with more than ten genes and fewer than 1000 genes were tested. The enrichment odds ratio and the respective *P*-value were calculated by a hypergeometric test. Nominal *P*-values were adjusted by the FDR method and the corrected significance threshold was set at 5% FDR. KEGG pathways were manually grouped and labelled into classes based on the known functions of genes overlapping with each pathway. HPO terms were grouped into a category distinct from KEGG pathways.

**Kidney proteins associated with blood pressure in proteome-wide association studies – drug tractability enrichment.** Drug tractability data was collected from the OpenTargets ftp server (http://ftp.ebi.ac.uk/pub/databases/opentargets/platform/latest/input/target-inputs/

tractability/tractability.tsv, data accessed 05/2023) and was a product of the OpenTargets tractability pipeline (https://github.com/chembl/tractability_pipeline_v2). Tractability categories for each modality were extracted from the columns "Category_sm", "Category_ab", "Category_PROTAC" and manually relabelled. The enrichment analysis was performed using 100,000 permutations for each of the eight categories. For each permutation a random sample of genes was selected (without replacement) of size equal to the number of genes present in the category being tested and then the number of genes assigned to that category was recorded. *P*-values were calculated by one-sided permutation test for positive enrichment which is calculated as the proportion of permuted values which exceed the test value (the number of BP PWAS protein belonging to the category in question) divided by the number of permutations performed. For visualisation, the distribution of all permuted values (per category) was smoothed using the function "stat_density_ridges" from the "ggridges" R package with a bandwidth of 0.4.

### Characerisation of urinary cell transcriptome

**Urinary cell collection, purification and RNA extraction.** Demographic information for all individuals included in this analysis is presented in Supplementary Data 25. In brief, 100 ml of fresh urine was collected and refrigerated (up to a maximum of 2 hours) until centrifugation at 2000 g for 30 minutes at 4°C. The supernatant was then discarded, and the pelletised material resuspended in 1 ml of PBS minus (Sigma-Aldrich). The sample was then spun a second time at 10,000 g for 4 minutes at room temperature. The supernatant was again removed and 1 ml of a mixture of 70% Hank's balanced salt solution (HBSS), 20% foetal calf serum (FCS) and 10% dimethyl sulfoxide (DMSO) before storage at -80°C. RNA was extracted using the Qiagen RNeasy kit, following the standard protocol. Urinary cell RNA concentration and purity was quantified by NanoDrop and RIN scores estimated by TapeStation (Agilent) with a high-sensitivity assay.

**Urinary cell RNA-sequencing and data processing.** Sequencing libraries were generated using the New England Biolabs (NEB) poly-A selection kit for Illumina sequencers. The gene expression quantification protocols was identical to that used by the GTEx for version 8 of their data release[122]. This protocol was selected so the urinary cell expression data could be compared with the GTEx gene expression profiles without introducing a potential bias introduced by the differences in the quantification protocol. Briefly, the read alignment to the genome was conducted using STAR[164], read deduplication and quantification – by RNASeQC (against the GTEx v8 transcriptome reference in units of TPM at the gene level). Our quality control process excluded samples with (i) less than ten million paired reads, (ii) less than 2 million mapped paired reads and (iii) those with a D-statistic greater than 5. A total of 33 urinary cell pellet samples passed all quality control criteria. Expressed genes were defined as those with greater than 6 mapped reads and a TPM > 0.1 in more than 20% of samples; 21,981 genes passed all quality control filters and were used for the downstream analyses.

**Salivary cell RNA-sequencing and RNA-sequencing metric calculation.** Salivary samples were obtained from the publicly available GEO series (GSE108664) of longitudinally collected saliva specimens sampled before administration of a pneumococcal vaccination[175]. We used all available 40 samples collected before administration of the vaccination ("pre-vaccination"). The NCBI SRA-toolkit was used to download and extract raw fastq data ("fasterq-dump") for all 40 samples. The raw fastq was then processed using the same pipeline applied to the urinary cell and GTEx tissue samples. Gene expression was also quantified in an identical manner. RNA-sequencing metrics were calculated by RNASeQC. RNASeQC metrics were manually grouped by the range or units of reported values into 3 groups: (1) Read and fragment

count metrics [0-100 million reads/fragments], (2) rates [0.0–1.0 frequency] and (3) small value metrics [0-2.0 units].

**Urinary cell transcriptome – overall characteristics.** We assigned one of four biotype categories to each expressed gene by grouping all Ensembl v83 "gene_biotype" values using a manual mapping (Supplementary Data 36). We then examined the complexity of the urinary cell and kidney tissue transcriptome by calculating the cumulative proportion of each transcriptome attributable to each expressed gene and ranking them from highest to lowest expression. We then quantified the cumulative proportion of each transcriptome which is attributable to the top 25, 50 and 75% of expressed genes.

To examine whether transcriptomic profiles of urinary cell pellets collected from patients undergoing cancer nephrectomies differ from those who did not have cancer we used hierarchical clustering of sample similarity values. We used Pearson distances to determine an inverse measure of urinary cell sample similarity. Pearson distances were calculated as (1 – Pearson correlation coefficient) between all sample pairs, using $\log_2$ TPM data from all genes expressed in urine. These distances were then clustered hierarchically using the complete linkage method for cluster agglomeration as implemented in the R function "hclust"; samples were annotated as "nephrectomy" (to indicate that the urinary cell sediment was secured from a patient undergoing surgical removal of the kidney due to cancer) and "biopsy" (to indicate that the urinary cell pellet was secured from a donor prior to kidney transplantation).

**Urinary cell overrepresented pathways.** The top 100 expressed genes, by median TPM, were identified in both (i) 33 urinary cell samples and (ii) 430 kidneys from HKTR[25]. The top 100 HGNC gene symbols from each were then examined by an overrepresentation analysis using the Database for Annotation, Visualization and Integrated Discovery (DAVID)[176] functional annotation tool with the KEGG pathways and Gene Ontology biological process term annotations. The results significant at 5% FDR were grouped by shared genes and were manually assigned to specific biological themes using known functionality of overlapping genes.

**Analysis of correlation between the transcriptomic footprint of urinary cells and different human tissues/cell-types.** We used the publicly available RNA-sequencing-derived median gene expression profiles for each of the 54 different GTEx tissues from the GTEx portal (accessed 09/2021). The gene expression data were processed using the standard GTEx computational pipeline[122]. In brief, gene expression was quantified by RNASeQC in TPM units at the sample level and then a median value was calculated across all samples from each tissue, for all genes. The urinary cell pellet expression profile from 33 individuals was summarised in an identical way to calculate a median gene expression profile. All 19,273 protein-coding genes from the GTEx v8 transcriptome reference were used in the correlation analysis. Overall transcriptomic correlation was calculated as the squared Spearman's correlation coefficient between the urinary cell pellet profile and the profile for each GTEx tissue. Tissues were then ordered according to the strength of their transcriptomic correlation with urinary cell pellets. To validate the findings on correlation between transcriptome of urinary cells and the kidney tissues from GTEx we then used RNA-sequencing data from our in-house collection of 430 kidney samples (HKTR)[25] quantified and pre-processed according to the same pipeline used for GTEx samples and urinary cell pellets. Overall transcriptome correlation was calculated between urinary cell pellets and the kidney profile from the HKTR resource using all 19,273 protein-coding genes from the GTEx v8 transcriptome reference.

**Expression of markers for specific segments of the nephron in urinary cells.** To investigate whether cellular components of different

kidney regions are present within the cells harvested from urine, we first selected marker genes by renal cell-type (either enriched or enhanced for any specific kidney cell-type), from the HPA[177]. Marker genes for cortex were selected from the following cell-types - "Proximal tubule", "Glomerulus", "Distal tubule". Medullary marker genes were selected from the "Loop of Henle", "Collecting duct" cell-types. Marker genes for cortex and medulla were then pruned by the following criteria: (i) median expression <than 1 $\log_2(TPM + 1)$ and (ii) any significant expression ( >10 $\log_2(TPM + 1)$) in a renal cell-type from the opposite region. Finally, marker genes were trimmed to the top 20 most highly expressed, per region. Gene expression for each tissue (in $\log_2(TPM + 1)$ units) was standardised before visualisation. Genes were ordered (within each kidney region) by hierarchical clustering of a Euclidean distance matrix calculated from the marker gene expression profile of all 3 tissues by the R function "hclust" using the complete linkage method.

**Analysis of correlation in expression of genes causally associated with blood pressure between the kidney and urinary cells.** We first identified that out of 399 BP TWAS genes, 339 had detectable expression in urinary cells. We used calculated median expression in $\log_2(TPM + 1)$ values for these genes from our urinary cell samples dataset ($n = 33$) and kidney tissue samples ($n = 430$) and then fitted a linear regression line (with kidney expression as the dependent variable and urinary cell expression as the independent variable). The ten genes with the smallest deviation (defined as smallest absolute residual) from the regression line were then identified and labelled.

**Glutamyl aminopeptidase gene (ENPEP) and blood pressure – blood pressure kidney transcriptome-wide association studies, blood pressure kidney proteome-wide association studies and rare variant analyses**
*ENPEP* expression across human tissues. Expression of ENPEP across 54 human tissues from the HPA was based on the reclassified HPA gene expression enrichment classification.

**Association between kidney expression of *ENPEP* and urinary sodium excretion.** Using UK Biobank imputed genotype data[147], we generated GReX based on the validated kidney PUMICE-derived prediction model for ENPEP. Urinary sodium excretion values (Data-Field 30530) from 326,986 unrelated UK Biobank individuals of white-European ethnicity with surviving sample level quality control[147] were normalised using rank-based inverse normal transformation. We then estimated the effect of ENPEP GReX on urinary sodium excretion by regressing normalised values of the latter on ENPEP GReX adjusting for age, sex, genotyping array and the top ten genetic principal components in linear regression.

*ENPEP* gene expression, protein abundance and blood pressure – multi-trait colocalisation. To assess shared genetic effects across *ENPEP* expression and protein abundance and DBP, we conducted multi-trait colocalisation using HyPrColoc[178] (v.1.0.0) with default settings. In brief, HyPrColoc implements an efficient deterministic Bayesian algorithm to detect colocalisation across multiple traits. The colocalisation analyses were performed using summary statistics from *cis*-eQTL (from HKTR, $n = 478$) and *cis*-protein quantitative trait loci (pQTL) (from CPTAC, $n = 72$) of glutamyl aminopeptidase and meta-analysis BP GWAS from UK Biobank and ICBP[22] ($n = \sim750,000$) in pre-defined disjoint LD blocks[151]. Only cases with > 50 overlapping variants were considered. Posterior probability > 0.8 was considered as a signal of colocalisation across traits consistent with all the traits sharing a causal variant within the tested region.

The summary statistics for *ENPEP* mRNA expression were collected from *cis*-eQTL analysis conducted using 478 HKTR kidney samples, as reported above. The summary statistics for protein abundance of

glutamyl aminopeptidase were obtained from *cis*-pQTL analysis of 72 CPTAC kidney samples with informative genotype and protein abundance. The log-transformed protein abundance was regressed against effect of allele dosage, age, sex, the top three genetic principal components derived from genotyped autosomal variants and top ten principal components derived from normalised protein abundance. Only genetic variants with minor allele count > 5, missingness rate <0.05 and within 500 kb from the transcription start site of ENPEP were considered. A total of 2478 genetic variants satisfied these criteria and were included in the *cis*-pQTL of glutamyl aminopeptidase.

**Homology modelling of truncated protein.** All PDB structures linked to the human *ENPEP* gene[179] in the UniProtKB database (release 2023_01)[180] (4kx7, 4kx8, 4kx9, 4kxa, 4kxb, 4kxc, 4kxd) were downloaded. All of them were resolved using X-ray crystallography, and have a resolution between 2.15 Å and 2.40 Å. The crystal structures were assessed using MolProbity[181–183]: (1) hydrogens were added, (2) Asn, Gln and His side-chain flips were allowed, and 3) the all-atom contacts and the geometry of the optimised structures were analysed. The structure with the fewer warnings in the different validation categories, and the best MolProbity score was selected to be the template for homology modelling: the optimised 4kx7 structure.

The protein sequences of the protein structure and the truncated protein were aligned using the Smith-Waterman algorithm[184] as implemented in EMBOSS[185]. The default parameters (BLOSUM62 substitution matrix, gap opening penalty = 10, and gap extension penalty = 0.5) were used.

Ten structural models were built using MODELLER (version 10.4)[186]. We used the option of slow (thorough) Variable Target Function Method and Molecular Dynamics optimisations. We repeated the whole optimisation cycle twice.

All ten structural models were assessed using MolProbity as previously mentioned; briefly, 1) hydrogens were added, 2) flips were allowed, and 3) the contacts and geometry were assessed. Again, the model with the fewer warnings, and the best MolProbity score was selected.

**Analysis and visualisation of the model of the truncated protein.** Canonical and truncated protein structures were visualised with ChimeraX (version 1.6)[187]. CATH (version 4.3)[188] annotations were used to identify the different structural domains within the canonical and truncated proteins. An assessment of the hydropathy of the surface of the protein structures was performed by scoring each residue with the Kyte and Doolittle scale[189]. The annotation on functional sites was collated from the UniProtKB database[180].

**Effect of rs33966350 on ENPEP mRNA expression, protein abundance, urinary sodium excretion and disease endpoints from FinnGen.** We first collected phenome-wide association results for variant rs33066350 from the FinnGen release R9 (r9.finngen.fi), which includes data from 377,277 individuals and 2269 disease endpoints. Effect of rs33066350 on each of the disease endpoint was estimated under the additive GWAS model containing age, sex, ten genetic PCs and genotyping batch as covariates. The correction for multiple testing was calculated using Bonferroni-adjusted $P$-value < 0.05 after adjustment for the total number of disease endpoints.

We used linear regression to assess the effect of rs33966350 on urinary excretion of sodium in 326,986 biologically unrelated white-European individuals who survived sample level quality control[147] in UK Biobank. Urinary sodium excretion (Data-Field 30530) underwent rank-based inverse normal transformation. We then regressed normalised urinary sodium excretion on genotype of rs33966350 (under additive model of inheritance) adjusting for age, sex, genotyping array and the top ten genetic PCs generated from autosomal genotype data.

We next estimated the effect of rare loss-of-function variant (rs33966350) on *ENPEP* mRNA expression in 478 individuals from HKTR. In the absence of carriers of AA genotype in this dataset, we regressed the RNA-sequencing-derived normalised renal *ENPEP* expression on rare allele dosage adjusting for age, sex, tissue source (nephrectomy/biopsy), the first three genetic principal components, the 100 PEER hidden factors and seven cell-type proportions. The normalised *ENPEP* expression values were derived by logarithmic transformation, quantile normalisation and rank-based inverse normal transformation. We also evaluated the effect of rs33966350 on *ENPEP* mRNA expression in an independent cohort (CPTAC, $n = 60$) by regressing the normalised renal *ENPEP* expression on genotype of rs33966350 (under additive model of inheritance) adjusting for age, sex, the top three genetic principal components derived from genotyped autosomal variants and top ten principal components derived from normalised gene expression. The normalised *ENPEP* expression values in CPTAC were obtained by logarithmic transformation.

We assessed the effect of rs33966350 on glutamyl aminopeptidase protein in the same cohort (CPTAC, $n = 67$) by regressing the normalised renal glutamyl aminopeptidase protein abundance on genotype of rs33966350 (under additive model of inheritance) adjusting for the same covariates as above (i.e. those used in the analysis of rs33966350 effect on *ENPEP* mRNA expression in CPTAC). The normalised glutamyl aminopeptidase protein abundance in CPTAC were derived by logarithmic transformation.

**UK Biobank ethical compliance.** UK Biobank has approval from the North West Multi-centre Research Ethics Committee (MREC) to obtain and disseminate data and samples from the participants, and these ethical regulations cover the work in this study. Written informed consent was obtained from all participants.

### Reporting summary
Further information on research design is available in the Nature Portfolio Reporting Summary linked to this article.

## Data availability
The PUMICE-derived TWAS, microRNA-TWAS and PWAS summary statistics generated in this study are available in the Supplementary Data 6, 14 and 22, respectively. Bulk RNA-seq data TCGA, CPTAC and GTEx can be accessed from Genomic Data Commons (GDC) Data Portal (https://portal.gdc.cancer.gov/) and GTEx portal (https://gtexportal.org/home/). The normalised kidney gene expression, miRNA expression data and urinary transcriptomic data from HKTR are archived at https://doi.org/10.48420/24871785. Full summary statistics of blood pressure GWAS using 337,422 unrelated white European individuals from UK Biobank are available at https://doi.org/10.48420/24851436. The sample-size-balanced GTEx v8 TWAS models are available at https://doi.org/10.48420/24871794. The kidney PUMICE model is available at https://github.com/ckhunsr1/PUMICE/tree/master/model_HKTR. Source data are provided with this paper.

## Code availability
Our studies make use of well-established computational and statistical analysis software, and these are fully referenced in the main text and Methods. All software used to perform these studies is publicly available.

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

## Acknowledgements

These studies were supported by British Heart Foundation grants (PG/22/10957, PG/19/16/34270 and PG/17/35/33001), Kidney Research UK grants (RP_017_20180302 and RP_013_20190305) to M.T., British Heart Foundation Accelerator Award (AA/18/4/34221) to B.K. and M.T., NIHR Manchester Biomedical Research Centre (NIHR203308) to B.K., A.P.M. and M.T. X.X. was supported by the University of Manchester Wellcome Institutional Strategic Support Funding, S.R. is supported by funding from Health Innovation Manchester, the French Society of Hypertension (SFHTA) and the French Hypertension Research Foundation (FRHTA). J.M.E. is supported by funding from Translation Manchester and Health Innovation Manchester. B.K. was supported by a British Heart Foundation Personal Chair. C.K. was supported by the National Institutes of Health grant T32GM118294 and T32LM012415. C.K. was also funded in part by support from Robert and Sevia Finkelstein. P.R.P is supported by the Victoria Fellowship, Australia. A.S.W. acknowledges funding from MRC-NIHR UK Rare Disease Research Platform MR/Y008340/1. D.J.L is partially supported by NIH grants R01AI174108, R01HG011035, R56HG012358, and R01ES036042. We are grateful to Isobel Stewart and Claudia Langenberg from Omicscience.org for sharing summary statistics data for integrative multiomics studies. Figure 1A, E were partially created with Bioicons (https://bioicons.com). Figures 2H, 4H, D, G, 5A, 8E, G, I and S6 were partially created with BioRender (https://www.biorender.com). The data/analyses presented in the current publication are based on the use of study data downloaded from the dbGaP web site, under dbGaP accession e.g. phs000178.v11.p8, phs000424.v9.p2, phs001287.v17.p6. Access to UK Biobank data has been through approved project (46114).

## Author contributions

X.X, C.K. and J.M.E. performed the main analytical and experimental tasks. S.R., D.S., S.S., D.T., H.M., L.W., M.Dr, A.M., A.C.L., P.R.P., J.Re, A.R.D., M.De, J.P.D., P.R.M.-G., M. Wald, A.S.W., B.K., A.H.J.D, N.J.S., T.J.G., A.P.M., D.J.L and F.J.C provided additional analyses and data. G.R., J.Ry, R.K., M.S., M.Walc, A.A., E.Z.-S., W.W., J.Z. and P.B. were involved in the collection of kidney resources. X.X, C.K, J.M.E., D.S. and M.T. contributed to drafting the manuscript. M.T. led the preparations of the manuscript and provided the overall supervision of the project. All authors reviewed and approved the accepted version of the manuscript.

## Competing interests

A.H.J.D. is supported by research grant from Alnylam Pharmaceuticals, Boston, USA to perform animal studies with angiotensinogen siRNA (money is paid to the university). The remaining authors declare no competing interests.

## Additional information

Xiaoguang Xu[1], Chachrit Khunsriraksakul[2], James M. Eales[1], Sebastien Rubin[1], David Scannali[1], Sushant Saluja[1], David Talavera[1], Havell Markus[2], Lida Wang[2], Maciej Drzal[1], Akhlaq Maan[1], Abigail C. Lay[1], Priscilla R. Prestes[3], Jeniece Regan[2], Avantika R. Diwadkar[2], Matthew Denniff[4], Grzegorz Rempega[5], Jakub Ryszawy[5], Robert Król[6], John P. Dormer[7], Monika Szulinska[8], Marta Walczak[9], Andrzej Antczak[10], Pamela R. Matías-García[11,12,13], Melanie Waldenberger[11,12,13], Adrian S. Woolf[14,15], Bernard Keavney[1,16], Ewa Zukowska-Szczechowska[17], Wojciech Wystrychowski[6], Joanna Zywiec[18], Pawel Bogdanski[8], A. H. Jan Danser[19], Nilesh J. Samani[4,20], Tomasz J. Guzik[21,22,23], Andrew P. Morris[24], Dajiang J. Liu[2], Fadi J. Charchar[3,4,25], Human Kidney Tissue Resource Study Group* & Maciej Tomaszewski[1,16]✉

[1]Division of Cardiovascular Sciences, Faculty of Medicine, Biology and Health, University of Manchester, Manchester, UK. [2]Department of Public Health Sciences, Penn State College of Medicine, Hershey, PA, USA. [3]Health Innovation and Transformation Centre, Federation University Australia, Ballarat, Australia. [4]Department of Cardiovascular Sciences, University of Leicester, Leicester, UK. [5]Department of Urology, Medical University of Silesia, Katowice, Poland. [6]Department of General, Vascular and Transplant Surgery, Faculty of Medical Sciences in Katowice, Medical University of Silesia, Katowice, Poland. [7]Department of Cellular Pathology, University Hospitals of Leicester, Leicester, UK. [8]Department of Obesity, Metabolic Disorders Treatment and Clinical Dietetics, Karol Marcinkowski University of Medical Sciences, Poznan, Poland. [9]Department of Internal Diseases, Metabolic Disorders and Arterial Hypertension, Poznan University of Medical Sciences, Poznan, Poland. [10]Department of Urology and Uro-oncology, Karol Marcinkowski University of Medical Sciences, Poznan, Poland. [11]Institute of Epidemiology, Helmholtz Center Munich, Neuherberg, Germany. [12]Research Unit Molecular Epidemiology, Helmholtz Center Munich, Neuherberg, Germany. [13]German Research Center for Cardiovascular Disease (DZHK), partner site Munich Heart Alliance, Munich, Germany. [14]Division of Cell Matrix Biology and Regenerative Medicine, Faculty of Biology, Medicine and Health, University of Manchester, Manchester, UK. [15]Royal Manchester Children's Hospital and Manchester Academic Health Science Centre, Manchester University NHS Foundation Trust, Manchester, UK. [16]Manchester Academic Health Science Centre, Manchester University NHS Foundation Trust Manchester, Manchester Royal Infirmary, Manchester, UK. [17]Department of Health Care, Silesian Medical College, Katowice, Poland. [18]Department of Internal Medicine, Diabetology and Nephrology, Zabrze, Medical University of Silesia, Katowice, Poland. [19]Department of Internal Medicine, Division of Pharmacology and Vascular Medicine, Erasmus Medical Centre, Rotterdam, The Netherlands. [20]NIHR Leicester Biomedical Research Centre, Glenfield Hospital, Leicester, UK. [21]Department of Internal Medicine, Jagiellonian University Medical College, Kraków, Poland. [22]Centre for Cardiovascular Sciences, Queen's Medical Research Institute, University of Edinburgh, Edinburgh, UK. [23]Center for Medical Genomics OMICRON, Jagiellonian University Medical College, Kraków, Poland. [24]Centre for Genetics and Genomics Versus Arthritis, Centre for Musculoskeletal Research, Division of Musculoskeletal & Dermatological Sciences, Faculty of Medicine, Biology and Health, University of Manchester, Manchester, UK. [25]Department of Physiology, University of Melbourne, Melbourne, Australia. *A list of authors and their affiliations appears at the end of the paper. ✉e-mail: maciej.tomaszewski@manchester.ac.uk

## Human Kidney Tissue Resource Study Group

Grzegorz Rempega[5], Jakub Ryszawy[5], Robert Król[6], Monika Szulinska[8], Marta Walczak[9], Andrzej Antczak[10], Bernard Keavney[1,16], Ewa Zukowska-Szczechowska[17], Wojciech Wystrychowski[6], Joanna Zywiec[18], Pawel Bogdanski[8], Fadi J. Charchar[3,4,25] & Maciej Tomaszewski[1,16]✉

A full list of members and their affiliations appears in the Supplementary Information.

