## [Peer Review File · Nature Communications]

Genetic imputation of kidney transcriptome, proteome and integrative multi-omics yields new targets for blood pressure and hypertensionREVIEWER COMMENTS

Reviewer #1 (Remarks to the Author):

Xu et al. present their analysis on the “Genetic imputation of kidney transcriptome, proteome and integrative multi-omics yields new molecular, diagnostic and therapeutic targets for blood pressure and hypertension”. In this comprehensive and pioneering study, the researchers delved into the intricate genetic mechanisms governing blood pressure regulation and human hypertension through the kidney.

The study combines genotype and gene expression data from 700 human kidneys, identifying 889 genes with an putatively causal effect on blood pressure. Through various analyses, 399 of these genes are prioritized as key contributors to blood pressure regulation.

The research also uncovers kidney-related proteins and microRNAs impacting blood pressure and highlights their diagnostic potential. It suggests the repurposing of certain medications for hypertension and identifies therapeutic opportunities.

Overall, this study provides important insights into blood pressure with potential clinical applications in hypertension treatment and will be of interest to the wider research community as an exemplary approach to integrate multiple omics layers in a causal-inference methodology.

Some suggestions for enhancing the manuscript further are below:

Major:

- Excellent choice to subsample GTEx to 65 samples for each tissue to support the comparability of tissues. An interesting comparison would be at what cost this subsampling comes. I.e. how many more hits would the application of the publicly available GTEx v8 TWAS models based on the full (EUR) sample size have yielded? <http://gusevlab.org/projects/fusion/#gtex-v8-multi-tissue-expression>

- Fig S3: While the UKB and the ICBP GWAS summary statistics were based on comparable sample sizes (approx. 300k) the discovery stage of the reciprocal replication proved to be quite different in power (1734 vs 701 hits). Do you have an indication why this might be?

- The TWAS approach as implemented here can be interpreted as a form of MR (PMID: 35433074). As the epigenetic information was implemented on an annotation basis and no individual patient epigenetic data was used in the genetic imputation of gene expression this should also hold for the extended kidney TWAS approach presented here. To account for a potential effects of linkage disequilibrium-driven correlations it is recommended to combine TWAS with some form of colocalization analysis or something like the FOCUS approach employed here.

Hence the reciprocal TWAS using UKB and ICBP and the two-sample MR + FOCUS based on the meta-analysis of UKB and ICBP are methodologically closely related. Furthermore, the downstream CMap analyses seem to be based on TWAS on the meta-analysis bringing approaches even closer. A reader less familiar with the topic might be under the impression of a more independent line of evidence presented by the different results sections. That said as with any MR it is very helpful to combine several approaches to get a better hold on the underlying assumptions and robustness of detected putatively causal links. Maybe a more prominent guidance of readers that establishes the link would address the broad audience of Nature Communications.

- An aspect that could potentially add to the manuscript is a more systematic comparison of the connection of kidney gene expression of gene X, blood pressure and kidney function. Given the methodological links a recent manuscript on TWAS (including kidney specific models) and PWAS of kidney function could provide a suitable list of implied genes (PMID: 37365616).

- Given the importance of the kidneys to many traits the public availability of the newly trained kidney TWAS models would be highly encouraged. Additionally, the sample-size-balanced GTEx v8 TWAS models would be a great resource for many researchers prioritizing the tissue(s) of relevance for their phenotype of interest.

Minor:

- Fig1D: The figure suggests to me that the kidney tissue resource is a direct input. Arent these already integrated into the TWAS models that were generated and validated in F1B&C?
- Fig2A: Is the color scale defined by the sum of SBP and DBP scores? Could be added to F1S?
- Fig2D&E: Consider including the individual dots in the graphs.

Reviewer #2 (Remarks to the Author):

Dr. Xu and colleagues present a compendium article on the role that genes expressed on kidney tissues have on blood pressure regulation. Even if the manuscript is well written, I cannot say it was a fun to read, because it is too dense with results. But I was pleasantly impressed by the series of well-organized analyses that were presented and by the illustration of all that can be done by integrating gene expression in kidney tissue from a large cohort of patients with the latest bioinformatics tools. I found no major flaws in this work, but I would like to raise some issues to the Authors' attention, in the hope they can help improving the manuscript a bit further.

1) Overall, the number of findings is impressive. Where possible, I would recommend using more stringent statistical significance thresholds rather than the FDR. Having a bit fewer but more specific results would make the paper stronger. Particularly in the reciprocal replication analysis: what is the rationale, here, to apply an FDR as opposed to more stringent Bonferroni criterion?

2) In the Results section "The predicted kidney gene expression of metabolic transporters/receptors recapitulates their biological functions and expected contributions to human disease", given only a few genes were selected as a proof-of-principle, are the Authors sure they can claim that "these examples illustrate the biological robustness of the models for genetically predicted expression of kidney genes developed for the purpose of TWAS"? In other words: is there any risk that these finding are gene-specific and not generalizable?

3) For the top drug classes on Table S8 (those with $FDR < 0.05$), for which the potential of drug repurposing is claimed, it would be informative to assess the specificity to being beneficial to BP control as opposed to damage other human body functions.

4) Evidence that 663 out of 889 genes had a causal effect on at least one BP trait is overwhelming. Given an FDR of 0.05 is quite liberal, I would recommend the Authors to revisit this analysis using a more stringent Bonferroni correction for the number of genes tested (not the number of traits as they are strongly correlated, I assume) – see also point #1

5) For any GWAS conducted (on traits and expression), please report in a supplementary table, the genomic inflation factor and if correction had to be applied, the genetic model, and the relevant filters applied to the genotypes (MAF, imputation quality).

6) In the correlation analysis between gene expressions in urinary cells and 54 tissues (Results, Fig 7F, Fig 7E), I don't understand why the Authors use the squared of the Spearman correlation coefficient as opposed to the simple Spearman correlation coefficient, given correlations could be, in principle, either positive or negative. In addition, with such large sample size, all P-values for correlation coefficients are irrelevant and can be omitted. Interestingly, Fig. 7F seems to suggest that, beyond the high correlation, there is a group of genes whose expression escapes the diagonal, creating a cloud to the lower right. Did the authors notice the existence of a subgroup of genes with varying expression in the urine by systematically not expressed in the opposing tissue?

7) The Authors leave it uncommented the fact that a previous trial on ENPEP was unsuccessful, whilst their causal analyses showed a clear causal effect of ENPEP modification on BP. Why this difference? Could the trial have been underpowered or suffered of design bias? Could the Mendelian randomization studies still subjected to uncontrolled confounding, invalidating their results? Any other possible explanation?

8) Please, include discussion on the limitations of this study.

9) "Hypertension" should be removed from the title, as the manuscript is almost fully focused on blood pressure.

10) What does sensational sentences like „we scale up the discovery of BP genes by at least two-fold“ and "DNA-informed imputation to other molecular layers of the kidney" mean? Could the Authors rephrase, making sentences more informative?

11) The final part of the Introduction reads like a repetition of the abstract. Could the Authors use the space to add information for the reader instead?

12) In the Results section "Prioritisation of human cell-types/tissues for relevance to blood pressure through transcriptome-wide association studies", 2nd paragraph, the sense of "combine" is a bit obscure. The Authors should revisit the entire paragraph making it less abstract and more clearly reflective of what was really done with the data.

13) In Figure 1A, symbols are nice but not easy to interpret for things like "cultured fibroblast". It would be beneficial to add labels next to the dots in the graph.

14) Please, define what "genome-wide contacts" are.

15) Please, justify the sample size of UK Biobank samples.

16) What are "100 hidden factors estimated using PEER"?

Responses to Reviewers' comments.

Manuscript: NCOMMS-23-34237-T

Corresponding author: Maciej Tomaszewski

Reviewer 1

Xu et al. present their analysis on the “Genetic imputation of kidney transcriptome, proteome and integrative multi-omics yields new molecular, diagnostic and therapeutic targets for blood pressure and hypertension”. In this comprehensive and pioneering study, the researchers delved into the intricate genetic mechanisms governing blood pressure regulation and human hypertension through the kidney. The study combines genotype and gene expression data from 700 human kidneys, identifying 889 genes with an putatively causal effect on blood pressure. Through various analyses, 399 of these genes are prioritized as key contributors to blood pressure regulation. The research also uncovers kidney-related proteins and microRNAs impacting blood pressure and highlights their diagnostic potential. It suggests the repurposing of certain medications for hypertension and identifies therapeutic opportunities. Overall, this study provides important insights into blood pressure with potential clinical applications in hypertension treatment and will be of interest to the wider research community as an exemplary approach to integrate multiple omics layers in a causal-inference methodology. Some suggestions for enhancing the manuscript further are below.

We are very grateful to Reviewer 1 for their positive feedback on our study, a very thorough assessment of our manuscript and all the constructive comments and suggestions. We have addressed all the points raised by Reviewer 1 in our responses listed below and made the corresponding changes in the revised version of the manuscript. We believe that this revision has further strengthened the overall value of our paper for which we are very grateful to Reviewer 1.

R1.1. Major issues. Excellent choice to subsample GTEx to 65 samples for each tissue to support the comparability of tissues. An interesting comparison would be at what cost this subsampling comes. I.e. how many more hits would the application of the publicly available GTEx v8 TWAS models based on the full (EUR) sample size have yielded? <http://gusevlab.org/projects/fusion/> [gusevlab.org]#gtex-v8-multi-tissue-expression

We appreciate Reviewer 1's positive comments on our experimental strategy. Indeed, we have carefully contemplated a potential drop in the discovery rate at the expense of optimising our normalisation strategy. We were aware that the number of TWAS signals would drop in “down-sampled” tissues given the effect the sample size will have on TWAS discovery. However, the primary goal of this experiment was to ensure the full comparability of BP TWAS discovery rates across different tissues rather than to uncover new genes of BP in each tissue.

To quantify the impact of equalising the sample size across the panel of 49 tissues on the discovery rate (i.e. number of independent TWAS hits), we have run SBP TWAS using GTEx v8 TWAS models for 49 tissues [available from PredictDB (<https://predictdb.org/post/2021/07/21/gtex-v8-models-on-eqtl-and-sqtl/>)] using (i) the full available sample size for each tissue and (ii) an equal size of 65 samples (as in our experiments). Please note that in the analyses conducted for the purpose of this revision, we used GTEx v8 models from PredictDB rather than from TWAS FUSION because: (i) we trained our models following the PredictDB model prediction pipeline which is different from the TWAS FUSION; and (ii) the TWAS results will not be comparable if the TWAS models are trained from different model prediction methods.

As expected, we observed: (i) a drop in SBP TWAS discovery rates (i.e. number of TWAS genes) ranging from 16.3% to 80.9% across 48 tissues with samples greater than 100 (Table R1 below); and (ii) a strong correlation between the drop in the number of independent TWAS genes associated with SBP and the reduction in tissue sample size (Figure S1 below).

Table R1. Correlation between reduction in sample size and loss of TWAS discovery rate across 48 GTEx tissues with sample size greater than 100.

GTEx tissue	Reduction in sample size	Percentage of TWAS gene dropped (loss of discovery)
Adipose_Subcutaneous	88.8%	75.9%
Adipose_Visceral_Omentum	86.1%	80.9%
Adrenal_Gland	72.1%	62.6%
Artery_Aorta	83.2%	77.5%
Artery_Coronary	69.5%	63.3%
Artery_Tibial	88.9%	79.4%
Brain_Amygdala	49.6%	43.3%
Brain_Anterior_cingulate_cortex_BA24	55.8%	51.4%
Brain_Caudate_basal_ganglia	66.5%	55.4%
Brain_Cerebellar_Hemisphere	62.9%	51.9%
Brain_Cerebellum	68.9%	62.4%
Brain_Cortex	68.3%	58.6%
Brain_Frontal_Cortex_BA9	62.9%	61.6%
Brain_Hippocampus	60.6%	48.4%
Brain_Hypothalamus	61.8%	54.9%
Brain_Nucleus_accumbens_basal_ganglia	67.8%	52.0%
Brain_Putamen_basal_ganglia	61.8%	63.5%
Brain_Spinal_cord_cervical_c-1	48.4%	45.2%
Brain_Substantia_nigra	43.0%	25.0%
Breast_Mammary_Tissue	83.6%	75.5%
Cells_Cultured_fibroblasts	86.5%	73.1%
Cells_EBV-transformed_lymphocytes	55.8%	39.1%
Colon_Sigmoid	79.6%	71.4%
Colon_Transverse	82.3%	75.7%
Esophagus_Gastroesophageal_Junction	80.3%	72.1%
Esophagus_Mucosa	86.9%	71.4%
Esophagus_Muscularis	86.0%	78.4%
Heart_Atrial_Appendage	82.5%	80.9%
Heart_Left_Ventricle	83.2%	73.3%
Liver	68.8%	53.3%
Lung	87.4%	80.2%
Minor_Salivary_Gland	54.9%	16.3%
Muscle_Skeletal	90.8%	80.8%
Nerve_Tibial	87.8%	77.8%
Ovary	61.1%	53.0%
Pancreas	78.7%	65.9%
Pituitary	72.6%	64.0%
Prostate	70.6%	63.8%
Skin_Not_Sun_Exposed_Suprapubic	87.4%	74.8%
Skin_Sun_Exposed_Lower_leg	89.3%	77.9%
Small_Intestine_Terminal_Ileum	62.6%	44.8%

Spleen	71.4%	59.1%
Stomach	79.9%	78.2%
Testis	79.8%	67.1%
Thyroid	88.7%	67.7%
Uterus	49.6%	46.7%
Vagina	53.9%	44.1%
Whole_Blood	90.3%	73.0%

Figure R1. Correlation between reduction in sample size and loss of TWAS discovery rate across 48 GTEx tissues with sample size greater than 100. r – Spearman correlation coefficient, P-value – level of statistical significance from Spearman correlation test.

We have further commented on it in the revised version of Results (Page 4).

“We used an equal number of samples (n=65) across the panel of 49 tissues to maximise the comparability of BP TWAS discovery rates across these tissues accepting that this would be at the expense of the individual discovery rates in tissues that have been down-sampled.”

R1.2. Fig S3: While the UKB and the ICBP GWAS summary statistics were based on comparable sample sizes (approx. 300k) the discovery stage of the reciprocal replication proved to be quite different in power (1734 vs 701 hits). Do you have an indication why this might be?

We agree. Further to Reviewer 1’s comment we investigated how UK Biobank and ICBP sample sizes contributed to generation of BP TWAS models based on the SNPs selected as model predictors (model SNPs). The ICBP GWAS summary statistics are based on a meta-analysis across a large number of studies. Consequently, the actual number of individuals contributing to the GWAS summary statistics for a given SNP will depend on the number of GWAS contributing to the ICBP meta-analysis for which the SNP is reported. SNPs might not be reported in a contributing GWAS because of low imputation quality, for example.

Indeed, for model SNPs selected using ICBP GWAS summary statistics, there was substantial variation in the number of individuals contributing to those summary statistics. This is a possible explanation for the difference in power to detect associations with BP UK Biobank and ICBP GWAS (please see Table R2 below).

Table R2. Numerical contributions of individuals from UK Biobank and ICBP GWAS to model SNPs.

Sample size on model SNPs	UKB-SBP	UKB-DBP	UKB-PP	ICBP-SBP	ICBP-DBP	ICBP-PP
Minimum	337,422	337,422	337,422	172,349	181,245	172,347
1st quantile	337,422	337,422	337,422	237,600	252,395	237,594
Median	337,422	337,422	337,422	265,238	281,435	265,235
3rd quantile	337,422	337,422	337,422	275,043	291,626	275,040
Maximum	337,422	337,422	337,422	287,074	298,942	287,094

Sample size – number of individuals available, model SNPs – SNPs selected as model predictors, UKB – UK Biobank, ICBP – International Consortium for Blood Pressure, SBP – systolic blood pressure, DBP – diastolic blood pressure, PP – pulse pressure.

R1.3. The TWAS approach as implemented here can be interpreted as a form of MR (PMID: 35433074). As the epigenetic information was implemented on an annotation basis and no individual patient epigenetic data was used in the genetic imputation of gene expression this should also hold for the extended kidney TWAS approach presented here. To account for a potential effects of linkage disequilibrium-driven correlations it is recommended to combine TWAS with some form of colocalization analysis or something like the FOCUS approach employed here. Hence the reciprocal TWAS using UKB and ICBP and the two-sample MR + FOCUS based on the meta-analysis of UKB and ICBP are methodologically closely related. Furthermore, the downstream CMap analyses seem to be based on TWAS on the meta-analysis bringing approaches even closer. A reader less familiar with the topic might be under the impression of a more independent line of evidence presented by the different results sections. That said as with any MR it is very helpful to combine several approaches to get a better hold on the underlying assumptions and robustness of detected putatively causal links. Maybe a more prominent guidance of readers that establishes the link would address the broad audience of Nature Communications.

We concur with Reviewer 1 that a combination of MR and FOCUS does not provide fully independent sources of information on the signals detected from TWAS. However, with MR and FOCUS testing the TWAS signals from somewhat different angles (i.e. controlling for horizontal pleiotropy and co-regulation), the pipeline combining TWAS, MR and FOCUS offers an increased robustness in the discovery of genes associated with BP.

To clarify this, we added the following sentences under the paragraph of study limitations in Discussion (Pages 14-15).

“It should be noted that TWAS, MR and FOCUS are not fully orthogonal approaches to discovery of genes associated with a phenotype of interest. However, integration of these three approaches in one analytical pipeline increases the level of robustness and confidence in the detected signals (genes) that received support at each stage of this computational strategy”.

R1.4. An aspect that could potentially add to the manuscript is a more systematic comparison of the connection of kidney gene expression of gene X, blood pressure and kidney function. Given the methodological links a recent manuscript on TWAS (including kidney specific models) and PWAS of kidney function could provide a suitable list of implied genes (PMID: 37365616).

To address Reviewer 1's suggestion, we overlapped the set of 889 kidney genes from our BP TWAS with genes associated with estimated glomerular filtration rate, blood urea nitrogen or albumin creatinine ratio in TWAS conducted by Schlosser et al. (*Genome Biol.* 2023;24:150). The extent of overlap between our list of genes and those associated with CKD-defining traits in TWAS is shown below (Table R3). In summary, 18% (n=160) of genes associated with at least one BP trait showed an association with at least one of the three CKD-defining traits in the recent TWAS by Schlosser et al. (*Genome Biol.* 2023;24:150). We have further listed the overlapping genes in a new Supplementary Table (Table S7) and commented on these data in the revised version of the manuscript (Page 7).

“There was a modest (18%) degree of overlap between kidney genes associated with BP in our TWAS and the genes associated with CKD-defining traits in TWAS conducted by Schlosser et al. (Genome Biol. 2023;24:150) (Table S7). Amongst those that overlap were several notable genes linked already to both BP and kidney health/disease including interferon regulatory factor 5 gene (IRF5) (Nat Genet. 2021;53:630-637, Kidney Int. 2022;102:492-505), N-Acetyltransferase 8B gene (NAT8B) (BMC Med Genet. 2008;9:25, Nat Commun. 2018;9:4800) and Dipeptidase 1 gene (DPEP1) (Nat Commun. 2018;9:4800, Nat Genet. 2023;55:995-1008).”

Table R3. Degree of overlap between BP kidney TWAS genes and kidney genes associated with CKD-defining traits in TWAS conducted by Schlosser et al. (*Genome Biol.* 2023;24:150).

CKD-defining trait	BP kidney TWAS genes overlapped with kidney TWAS genes associated with CKD-defining traits
eGFR	140 (15.7%)
BUN	31 (3.5%)
ACR	39 (4.4%)
All CKD-defining traits	160 (18.0%)

Data are count and percentage. eGFR – estimated glomerular filtration rate calculated by creatinine and cystatin C, BUN – blood urea nitrogen, ACR – albumin creatinine ratio.

R1.5. Given the importance of the kidneys to many traits the public availability of the newly trained kidney TWAS models would be highly encouraged. Additionally, the sample-size-balanced GTEx v8 TWAS models would be a great resource for many researchers prioritizing the tissue(s) of relevance for their phenotype of interest.

Further to Reviewer 1's suggestion, we have made the kidney TWAS models publicly available at https://github.com/ckhunsr1/PUMICE/tree/master/model_HKTR. The sample-size-balanced GTEx v8 TWAS models will be uploaded to the figshare repository when submitting the revised manuscript.

R1.6. Minor issues. Fig1D: The figure suggests to me that the kidney tissue resource is a direct input. Arent these already integrated into the TWAS models that were generated and validated in F1B&C?

We concur with Reviewer 1. Indeed, the kidney tissue resource has already been integrated into the TWAS models represented in Figure1B&C. Thus, we have updated Figure1D by removing the blue box representing the kidney tissue resource.

R1.7. Fig2A: Is the color scale defined by the sum of SBP and DBP scores? Could be added to F1S?

Yes. The colour scale is aligned with the ranking of the tissue derived by the sum of SBP and DBP scores (from shades of orange representing the highest-ranking tissues to shades of pink for the tissues with the lowest positions in the ranking). We provided this clarification in the legend to Figure 2.

“Representation of 49 human tissues ... with blood pressure. The Natural Colour System-based scale is aligned with the position in the tissue ranking (from shades of orange

representing the highest ranking tissues to shades of pink for the tissues with the lowest positions)."

We further added number-labels to each tissue reflecting its exact position in the ranking and updated information in the legend to Figure 2A and Figure 1S, accordingly.

"The numbers from 1 to 49 reflect the position of each tissue in the ranking."

We also updated the Figure 1S legend as follows.

"Names for tissue icons shown in Figure 2A. Tissues are numbered from 1 to 49 based on their position in the ranking (1 being the top-ranking tissue)."

R1.8. Fig2D&E: Consider including the individual dots in the graphs.

We appreciate Reviewer 1's comment. We have considered including the individual dots in the Figure 2D&E. This has not improved the overall visibility of the Figure. Therefore, we decided to retain the current visualisation.

Reviewer 2

Dr. Xu and colleagues present a compendium article on the role that genes expressed on kidney tissues have on blood pressure regulation. Even if the manuscript is well written, I cannot say it was a fun to read, because it is too dense with results. But I was pleasantly impressed by the series of well-organized analyses that were presented and by the illustration of all that can be done by integrating gene expression in kidney tissue from a large cohort of patients with the latest bioinformatics tools. I found no major flaws in this work, but I would like to raise some issues to the Authors' attention, in the hope they can help improving the manuscript a bit further.

We are very grateful to Reviewer 2 for a very thorough revision of our paper – all the thoughtful comments and suggestions are much appreciated. We addressed these comments below and made corresponding changes in the manuscript. We believe that the revision has strengthened the overall value of our manuscript.

R.2.1 and R2.4. Overall, the number of findings is impressive. Where possible, I would recommend using more stringent statistical significance thresholds rather than the FDR. Having a bit fewer but more specific results would make the paper stronger. Particularly in the reciprocal replication analysis: what is the rationale, here, to apply an FDR as opposed to more stringent Bonferroni criterion?

Evidence that 663 out of 889 genes had a causal effect on at least one BP trait is overwhelming. Given an FDR of 0.05 is quite liberal, I would recommend the Authors to revisit this analysis using a more stringent Bonferroni correction for the number of genes tested (not the number of traits as they are strongly correlated, I assume) – see also point #1

We concur with Reviewer 2's comment that a Bonferroni correction is much more stringent than our choice (false discovery rate - FDR) for the multiple testing correction at the TWAS stage. It is also true that applying the conservative Bonferroni correction would result in fewer genes identified by TWAS whilst possibly increasing the confidence in the ones that survived this correction. One may argue though that Bonferroni correction is not the ideal solution here as the findings of TWAS are not entirely independent. Furthermore, the presence of reciprocal replication for each gene associated with BP in TWAS across two independent populations may be seen as an important argument for the less (rather than more) conservative correction for multiple testing. We should highlight that identifying fewer but highly credible kidney genes has been an important objective of our project and we contemplated different strategies to achieve that early – at the study design stage. In the end, we chose to minimise the chances of reporting less credible kidney genes through applying additional filters, i.e.: (i) independent replication at the TWAS stage; (ii) Mendelian randomisation after TWAS; and (iii) FOCUS-based fine mapping at the final stage of our computational pipeline. We reasoned that creating

a layered system of gene replications and refinements through applying these additional downstream filters on genes delivered by TWAS would not only provide a stringent statistical correction but would also deliver a more robust confirmation of the credibility of the identified BP kidney genes when compared to applying a conservative correction for multiple correction (i.e. Bonferroni) at the TWAS stage. Indeed, we have narrowed down the number of kidney genes showing causal association with BP from 889 (at the TWAS stage) to 399 (post MR and FOCUS-based fine mapping).

We have added the following comment to the revised manuscript (under limitations of the study) on Page 15:

“Secondly, at the TWAS stage we chose an FDR-based correction for multiple testing rather than a more conservative Bonferroni correction. This means we could have discovered more genes with perhaps less stringent statistical evidence. However, we have carefully reduced the likelihood of false positive signals through a layered system of gene replications and refinements created by applying additional downstream filters on genes delivered by TWAS, i.e., independent replication, MR and FOCUS.”

R2.2 In the Results section “The predicted kidney gene expression of metabolic transporters/receptors recapitulates their biological functions and expected contributions to human disease”, given only a few genes were selected as a proof-of-principle, are the Authors sure they can claim that “these examples illustrate the biological robustness of the models for genetically predicted expression of kidney genes developed for the purpose of TWAS”? In other words: is there any risk that these finding are gene-specific and not generalizable?

We have carefully considered this important comment. We should first clarify that we did apply a set of computational criteria to identify “example kidney genes” for the purpose of this experiment. The following criteria were applied: (i) the gene must have been imputable in TWAS; (ii) the gene must have had a relevant biochemical blood phenotype reflective of the kidney’s involvement in excretion and/or reabsorption of solutes; (iii) the relevant blood phenotype must have been measured in UK Biobank; and (iv) the gene must have been mapped to the GWAS Catalog top SNP in GWAS of the corresponding blood phenotype. This has been highlighted in Methods section on Page 21 of the revised manuscript. While this means that we are confident that the genes selected for the purpose of this experiment have undergone a robust selection process, we accept that we examined only a limited number of genes in this experiment. Indeed, for some of the relevant kidney transporters and receptors, our predictive models did not converge, whilst for others we were limited by the lack of relevant biochemical blood traits in UK Biobank (e.g. serum sodium and potassium levels). We have fully acknowledged Reviewer 2’s comment under “study limitations” in Discussion on Page 15. *“We accept that our “recapitulation analyses” tested a limited number of carefully selected biochemically active transporters and receptors in the kidney. Further studies inclusive of other genes, molecules, pathways and phenotypes of relevance to kidney physiology will help to strengthen the evidence for biological robustness of the predicted kidney gene expression models.”*

R2.3 For the top drug classes on Table S8 (those with FDR<0.05), for which the potential of drug repurposing is claimed, it would be informative to assess the specificity to being beneficial to BP control as opposed to damage other human body functions.

These drug class(es) and the selected representatives should be viewed only as potential candidates identified for the purpose of further repositioning studies, rather than fully-fledged clinically-ready pharmaceuticals for treating hypertension. It is essential to acknowledge that the question into how these drugs might affect other organs, while undoubtedly important from the clinical perspective, is beyond the scope of our project. Indeed, the intricacies of these drugs’ effects on BP and all other physiological systems and organs, and the net benefit to BP control versus the undesired effects are important questions that will require further dedicated investigations. We would like to think that by taking advantage of unique access to human tissues and “big data”, our study has provided investigators working on discovery of new

pharmaceutical repositioning opportunities with several interesting candidates for their future projects. Indeed, the recent topical reviews in *Circulation* (*Circulation*. 2021;144:159–169) and *European Heart Journal* (*Eur Heart J*. 2021;42:1464–1475) identified resistant hypertension as a condition where new drug solutions are necessary and where “assessment of existing non-cardiovascular drugs, testing their potential for repurposing for cardiovascular conditions” is urgently needed.

We have rephrased the following sections of the revised manuscript to better acknowledge that these drug classes represent only potential drug repurposing opportunities.

In Results section on Page 7:

“We also identified a new potential pharmaceutical repositioning opportunity for hypertension...”

In Discussion on Page 14:

“Apart from uncovering new potential opportunities in drug repositioning for hypertension ...”

R2.5 For any GWAS conducted (on traits and expression), please report in a supplementary table, the genomic inflation factor and if correction had to be applied, the genetic model, and the relevant filters applied to the genotypes (MAF, imputation quality).

Further to Reviewer 2’s suggestions, we provided the requested information in an additional Supplementary table (Supplementary Table S33). In brief, we provided details of the genetic model, inflation factor and other quality control parameters applied on SNPs from GWAS of systolic and diastolic blood pressure (SBP and DBP) as well as pulse pressure (PP) conducted in UK Biobank and ICBP. We also updated the corresponding section of the revised manuscript on Page 22.

“As an input into TWAS analysis, we used summary statistics of the associations between (i) 19,267,390 SNPs and SBP/DBP/PP from 337,422 UK Biobank individuals, and (ii) 7,371,711/7,476,460/7,359,508 SNPs and SBP/DBP/PP from 299,024 ICBP individuals¹²⁵. Details of the genetic model and statistical parameters used in generation of these data are provided in Table S33.”

R2.6.1. In the correlation analysis between gene expressions in urinary cells and 54 tissues (Results, Fig 7F, Fig 7E), I don’t understand why the Authors use the squared of the Spearman correlation coefficient as opposed to the simple Spearman correlation coefficient, given correlations could be, in principle, either positive or negative. In addition, with such large sample size, all P-values for correlation coefficients are irrelevant and can be omitted.

Further to Reviewer 2’s suggestion, we replaced the squared of the Spearman correlation coefficient to Spearman correlation coefficient in both Figure 7E and Figure 7F and updated the corresponding plots and the legend. We removed the P-value in Figure 7H as per Reviewer 2’s suggestion and updated the legend of Figure 7H correspondingly.

We have also updated the corresponding section in the revised manuscript on Page 12.

“Kidney cortex and kidney medulla showed ... at $r = 0.81$ and $r = 0.80$, respectively ...”

R2.6.2. Interestingly, Fig. 7F seems to suggest that, beyond the high correlation, there is a group of genes whose expression escapes the diagonal, creating a cloud to the lower right. Did the authors notice the existence of a subgroup of genes with varying expression in the urine by systematically not expressed in the opposing tissue?

The transcriptome of cells harvested from urine is not a perfect proxy of kidney transcriptome. While there is a high level of correlation in expression of genes across the kidney and urinary cells, inevitably there will be genes showing deviation from the line of best fit and those that are selectively expressed only in the kidney or urinary cells. Indeed, urinary sediments contain a mixture of cells including those shed from the kidney and those of other origin (i.e.

leukocytes, urothelial cells, etc.). This may explain why certain genes show deviations from the correlation line on kidney-urinary cell transcriptome correlation plot.

R2.7. The Authors leave it uncommented the fact that a previous trial on ENPEP was unsuccessful, whilst their causal analyses showed a clear causal effect of ENPEP modification on BP. Why this difference? Could the trial have been underpowered or suffered of design bias? Could the Mendelian randomization studies still subjected to uncontrolled confounding, invalidating their results? Any other possible explanation?

It is correct that the FRESH trial failed to demonstrate a significant blood pressure difference between the intervention arm (treated with firibastat - a pharmaceutical targeting glutamyl aminopeptidase - a protein encoded by ENPEP) and the placebo-arm in patients with uncontrolled primary hypertension (systolic BP 140-179 mm Hg at screening) despite treatment with at least two classes of antihypertensive drugs.

<https://www.healio.com/news/cardiology/20221111/firibastat-fails-to-improve-bp-in-resistant-hypertension-fresh>). Amongst the potential explanations is the selection of the molecular intervention that is designed to suppress the brain expression of ENPEP protein. Our data suggest that while ENPEP is certainly a therapeutically relevant molecule for hypertension, it should be the kidney expression of ENPEP that requires pharmaceutical targeting. Furthermore, given the evidence for the inverse direction of causal association between kidney ENPEP and BP and hypertension, such pharmaceuticals should increase (not inhibit) the abundance of kidney ENPEP.

We have briefly commented on that in Results on Page 13.

“These results provide a persuasive case for the development of pharmaceuticals that increase levels of ENPEP in the kidney (rather than inhibit its activity in the brain, e.g. as in FRESH trial)”

We further updated the corresponding section of Discussion on Page 14:

“Our analyses provide specific cues into targets already tested in clinical trials (i.e. glutamyl aminopeptidase encoded by ENPEP) or those emerging as novel therapeutic strategies for hypertension [(e.g. suppression of AGT mRNA expression using antisense oligonucleotides (JACC Basic Transl. Sci. 2021;6:485–496)].”

R2.8. Please, include discussion on the limitations of this study.

Further to Reviewer 2’s suggestion, we have added a section on study limitations to the revised version of *Discussion* (Pages 14-15). In this section, we commented on: (i) our computational pipeline to prioritise BP kidney genes after TWAS, (ii) the correction for multiple testing strategy, (iii) selection of kidney genes for the biological recapitulation analysis, (iv) using only individuals of white-European ancestry.

“Our study has several limitations. First, it should be noted that TWAS, MR and FOCUS are not fully orthogonal approaches to discovery of genes associated with a phenotype of interest. However, integration of these three approaches in one analytical pipeline increases the level of robustness and confidence in the detected signals (genes) that received support at each stage of this computational strategy. Second, at the TWAS stage we chose an FDR-based correction for multiple testing rather than a more conservative Bonferroni correction. This means we could have discovered more genes with perhaps less stringent statistical evidence. However, we have carefully minimised the chances for over-inflation of statistical signals through a layered system of gene replications and refinements created by applying additional downstream filters on genes delivered by TWAS, i.e., independent replication, MR and FOCUS. Third, we accept that our “recapitulation analyses” tested a limited number of carefully selected biochemically active transporters and receptors in the kidney. Further studies inclusive of other genes, molecules, pathways, and phenotypes of relevance to kidney physiology will help to strengthen the evidence for biological robustness of the predicted kidney gene expression models. Finally, due to a very limited availability of kidney tissues samples from individuals of non-white European origin (Kidney Int. 2022;102:492-505) we

were restricted in our studies to the European ancestry group. More international efforts are required to overcome this barrier to cross-ancestry analyses of kidney-relevant diseases in the post-GWAS era.”

R2.9. “Hypertension” should be removed from the title, as the manuscript is almost fully focused on blood pressure.

We concur with Reviewer 2 that while hypertension was not directly included in several stages of the project including TWAS, we included it in several analyses, e.g. ENPEP investigations, computational drug-repurposing analysis, and the FinnGen rare variant analysis for characterisation of 399 BP TWAS genes. Most importantly, several findings of our project including our computational drug-repurposing studies are very relevant to management of hypertension. Therefore, we kindly propose to retain “hypertension” in our title.

R2.10 and R2.11. What does sensational sentences like „we scale up the discovery of BP genes by at least two-fold” and “DNA-informed imputation to other molecular layers of the kidney” mean? Could the Authors rephrase, making sentences more informative?

The final part of the Introduction reads like a repetition of the abstract. Could the Authors use the space to add information for the reader instead?

Further to Reviewer 2’s comment we rephrased the corresponding section of *Introduction* on Page 4 as follows:

“Here, using a collection of up to 700 human kidneys^{25,31-34} and new computational algorithms embedded in three-dimensional (3D) configuration of the genome and kidney epigenome, we uncover 6,490 kidney genes with genetically imputable expression (approximately 30.3% of kidney transcriptome). We perform BP TWAS, Mendelian randomisation and fine-mapping of causal gene sets (FOCUS) to prioritise kidney effector genes for BP regulation. Through genetic imputation of the kidney micro-RNA-ome and proteome we uncover the identity of renal microRNAs and proteins associated with BP. Our computational drug repositioning analysis demonstrates BP effects of the existing non-cardiovascular medications and highlight new drug repositioning opportunities for hypertension. We also analyse urinary cell transcriptome to provide insights into diagnostic tractability of BP kidney genes. Finally, we triangulate outputs from plasma proteomics and metabolomics with kidney TWAS to yield new insights into pathways of blood pressure regulation.”

R2.12. In the Results section “Prioritisation of human cell-types/tissues for relevance to blood pressure through transcriptome-wide association studies”, 2nd paragraph, the sense of “combine” is a bit obscure. The Authors should revisit the entire paragraph making it less abstract and more clearly reflective of what was really done with the data.

S-PrediXcan, an extension of PrediXcan that allows the use of summary statistics from GWAS, estimates the mediating effects of gene expression levels on phenotypes. The method uses known relationships between single-nucleotide polymorphisms and gene expression to impute expression into GWAS samples.

We amended the corresponding sentence in the revised version of *Results* section accordingly (Page 5).

“After quality control and a nested cross-validation we used S-PrediXcan to conduct TWAS to estimate the mediating effects of gene expression levels in various GTEx tissues on systolic BP (SBP) and diastolic BP (DBP). In brief, we used eQTL data for 25,332 human genes across 49 GTEx tissues and summary statistics for 7,088,121 and 7,160,657 SNPs from the GWAS meta-analysis of SBP and DBP conducted in UK Biobank and the International Consortium for Blood Pressure (ICBP)²² (approximately 750,000 individuals).…”

R2.13. In Figure 1A, symbols are nice but not easy to interpret for things like “cultured fibroblast”. It would be beneficial to add labels next to the dots in the graph.

We believe that Reviewer 2 is actually referring to Figure 2A. As per Reviewer 2's suggestion, we have added a numerical label to each tissue to enhance the readability. We have also updated the legends of Figure 2A and Figure 1S accordingly. Please find further details in response R1.7 to Reviewer 1.

R2.14. Please, define what “genome-wide contacts” are.

We re-phrased these to “genomic contacts” to define “two regions of chromatin which are in close physical proximity in the HK-2 cell line” in the revised manuscript on Page 5.

“In total, we obtained 69,036,737 genomic contacts and ...”

We added the definition of “genomic contacts” to the legend of the revised version of Figure 2B.

“genomic contact – two regions of chromatin in close physical proximity in the HK-2 cell line.”

R2.15. Please, justify the sample size of UK Biobank samples.

We included up to ~337k individuals from UK Biobank for the purpose of this project. We followed UK Biobank sample-based quality control criteria (*Nature* 2018;562:203-209); excluded were samples/individuals based on the following criteria: (i) outliers in heterozygosity and missingness, (ii) sample call rate (computed using probesets internal to Affymetrix) <97% or the resolution of the distributions of intensity 'contrast' values <0.82, (iii) carriers of sex chromosomal abnormalities (configurations other than XX or XY), (iv) subjects who had cryptic relatedness with other individuals, or (v) individuals of non-white British genetic ancestry.

R2.16. What are “100 hidden factors estimated using PEER”?

The 100 hidden factors are variables that explain global sources of variation in the normalised gene expression data as derived by the software PEER (*Nat. Protoc.* 2012;7:500–507). This information has been described in the revised *Methods* on Page 19.

Other changes

1. We updated Figure 8J to reflect the negative effect of rs33966350 (per one copy of allele A) on urinary sodium in UK Biobank. We further added “Blue arrow represents negative association.” to the legend of Figure 8J.

2. We added a supplementary table (i.e., Table S27) to link the abbreviated tissue names in Figure 7E with the original names of GTEx tissues. We also updated the corresponding section in the manuscript on Page 12.

“Kidney cortex and kidney medulla showed the highest degree of correlation in expression of 19,273 protein-coding genes with urinary cells (Figure 7E, Table S27)”

3. We updated the list of references.

4. We have shortened the abstract and made it more factual.

REVIEWERS' COMMENTS

Reviewer #1 (Remarks to the Author):

Thank you very much for the excellent rebuttal that addresses all my suggestions.

Reviewer #2 (Remarks to the Author):

I would like to thank the Authors for addressing all raised issues and congratulate for the high quality of the work.